# Implicit Regularization of Decentralized Gradient Descent for Decentralized Sparse Regression

**Tongle Wu**
The Pennsylvania State University
tfw5381@psu.edu

**Ying Sun**
The Pennsylvania State University
ybs5190@psu.edu

## Abstract

We consider learning a sparse model from linear measurements taken by a network of agents. Different from existing decentralized methods designed based on the LASSO regression with explicit $\ell_1$ norm regularization, we exploit the implicit regularization of the decentralized optimization method applied to an over-parameterized nonconvex least squares formulation without sparse penalization. Our first result shows that despite nonconvexity, if the network connectivity is good, the well-known decentralized gradient descent algorithm (DGD) with small initialization and early stopping can compute the statistically optimal solution. Sufficient conditions on the initialization scale, choice of step size, network connectivity, and stopping time are further provided to achieve convergence. Our result recovers the convergence rate of gradient descent in the centralized setting, showing its tightness. Based on the analysis of DGD, we further propose a communication-efficient version, termed T-DGD, by truncating the iterates before transmission. In the high signal-to-noise ratio (SNR) regime, we show that T-DGD achieves comparable statistical accuracy to DGD, while the communication cost is logarithmic in the number of parameters. Numerical results are provided to validate the effectiveness of DGD and T-DGD for sparse learning through implicit regularization.

## 1   Introduction

Modern deep learning is generally in the over-parameterized regime where the models have significantly more parameters than available training examples [6, 41]. Although deep learning models exhibit remarkable performance in multiple domains, the theoretical understanding of optimization and generalization for deep learning is still limited. Recent studies show that despite being over-parameterized, gradient-based methods applied to minimize the emperical loss exhibit the implicit regularization phenomenon. For example, a line of works [3, 5, 11, 16] shows that with certain initialization, networks trained with gradient descent (GD) land in the "kernel regime" and share similar behaviors to the kernel method. However, the literature [2, 9, 24] suggests that kernel regime analyses fall short in explaining the success of deep learning because neural networks analyzed in the kernel regime are almost linearized, thus hindering feature learning from data. Further, many works [35, 38, 40] start investigating the "rich regime", showing that GD with small initialization induces structures on the solution, such as sparsity and low-rankness, that better explains the generalization capability of NNs. However, all aforementioned results are limited to the centralized setting, where data are stored on a single machine. Practical constraints such as limited computing and storage resources, data privacy and security, and regulation rules make the centralized learning framework increasingly inadequate for contemporary applications. Although a variety of decentralized learning algorithms can be applied to NN training, the questions of which solution they can converge, along with its generalization performance, are largely unclear.

38th Conference on Neural Information Processing Systems (NeurIPS 2024).

In this paper, we study the sparse learning problem [34] in the overparameterized regime, which shares many key characteristics with deep learning models but is more tractable to analyze, as a prototype for understanding the computation and statistical guarantees of decentralized learning algorithms. Specifically, we consider learning a sparse model $\boldsymbol{w}^\star$ from its noisy linear measurements over $m$ agents. These agents communicate over an undirected connected mesh network without a central coordinator, and each agent can only communicate with its one-hop neighbors. The $i$-th agent has its own $n$ samples $\{(\boldsymbol{x}_{i,j}, y_{i,j})\}_{j=1}^{n}$. Each $j$-th data pair $(\boldsymbol{x}_{i,j}, y_{i,j})$ is generated according to the noisy linear model

$$y_{i,j} = \boldsymbol{x}_{i,j}^T \boldsymbol{w}^\star + \xi_{i,j}, \quad \forall i \in [m] \text{ and } \forall j \in [n], \tag{1}$$

where $\boldsymbol{x}_{i,j} \in \mathbb{R}^d$ and $y_{i,j} \in \mathbb{R}$ denotes respectively the $j$-th feature and its corresponding response at $i$-th agent, $\xi_{i,j}$ is the observation noise, and $\boldsymbol{w}^\star \in \mathbb{R}^d$ is the sparse model parameter to be learned, common to all agents, and has only $s$ ($s \ll d$) non-zero elements. We are interested in the high-dimensional setting where the ambient dimension $d$ is substantially larger than the total sample size $N := mn$, i.e., $d \gg N$. By re-parameterizing $\boldsymbol{w} = \boldsymbol{u} \odot \boldsymbol{u} - \boldsymbol{v} \odot \boldsymbol{v}$, the loss function can be formulated as minimizing the following regularization-free nonlinear least square problem:

$$F(\boldsymbol{u}, \boldsymbol{v}) := \frac{1}{m} \sum_{i=1}^{m} f_i(\boldsymbol{u}, \boldsymbol{v}), \quad \text{with} \quad f_i(\boldsymbol{u}, \boldsymbol{v}) := \frac{1}{n} \left\| \boldsymbol{X}_i(\boldsymbol{u} \odot \boldsymbol{u} - \boldsymbol{v} \odot \boldsymbol{v}) - \boldsymbol{y}_i \right\|^2; \ \forall i \in [m],$$
$$\tag{2}$$

where $f_i(\boldsymbol{u}, \boldsymbol{v})$ corresponds to the loss function of $i$-th agent. Problem (2) can also regarded as the supervised learning problem on the diagonal linear network of degree-2 [38].

Problem (2) is highly non-convex with $\boldsymbol{u}$ and $\boldsymbol{v}$, however, recent works [12, 36, 42] demonstrate that if the design matrix satisfies the restricted isometric property (RIP) condition, the centralized GD without any regularization can yield the statistically optimal estimator with properly chosen initialization scale, step size, and early stopping time. This intriguing phenomenon is derived from the implicit regularization of GD. Roughly speaking, with small initialization, the gap $|u_i^2 - v_i^2|$ would increase with iteration for coordinate $i$ such that $w_i^\star \neq 0$, where the remaining ones stay small enough before early stopping. As a result, GD identifies the support of $\boldsymbol{w}^\star$ as the algorithm processes. In the decentralized setting, general results from the pure optimization perspective can only certify convergence to the stationary points, implying neither global optimality nor generalizability. Given the encouraging result of GD achieved in the centralized setting, it is natural to ask if statistically optimal solutions are also computable by decentralized gradient-type algorithms, which algorithm can achieve the goal, and what are the regularity conditions.

This paper aims to analyse the renowned decentralized gradient algorithm (DGD) for minimizing (2) over the undirected mesh networks. The main contributions of this paper are detailed as follows.

- **Statistical guarantee.** It is well established that even for convex objectives, DGD cannot compute an exact minimizer. It only converges to the neighborhood of the solution whose radius depends on the step size. However, we show that under specific conditions—namely, if the global design matrix satisfies the RIP condition, the initialization scale is sufficiently small, and the network is sufficiently connected—the solution computed by DGD with early stopping is statistically optimal.
- **Computational complexity.** Our convergence analysis reveals that the early stopping time increases logarithmically with the ambient dimension $d$. While network connectivity does not affect statistical error when it satisfies mild conditions, it does influence the stopping time of DGD to find the optimal estimator. Networks with poor connectivity will delay the early stopping time, and thus increase the iteration complexity.
- **Technical analysis.** Compared to the techniques used for analyzing the centralized GD [33, 38], proving the convergence of DGD faces the following challenge. Because the consensus error terms induced from the mesh network result in a perturbed version of the multiplicative update. Compared with the exact multiplicative updates, the challenge is that the additional error term outside of multiplication prevents applying the centralized analysis directly. In addition, the error terms within the multiplication have more complicated consensus error terms than that of the centralized setting, which requires bounding the consensus error terms carefully to control these error terms that can achieve the same order statistical error. To achieve this goal, we separately control the consensus errors on support $\boldsymbol{S}$ and non-support $\boldsymbol{S}^c$ by the magnitudes of parameters on support $\boldsymbol{S}$ and non-support $\boldsymbol{S}^c$, respectively. Our fine-grained analysis for consensus errors is distinct

from existing decentralized optimization analyses that bound the consensus errors uniformly. The additional error term also complicates the transfer of proof from the simplified non-negative $w^\star$ case to the general $w^\star$ setting as the centralized setting, we conduct a comprehensive induction process to both $u$ and $v$ simultaneously for general $w^\star$.

- **Truncated DGD.** We propose a communication-efficient truncated DGD (T-DGD) method that at each iteration, vectors being transmitted are truncated, keeping only $s$ elements with the largest magnitudes nonzero. We prove that if each agent has sufficient samples and the signal-to-noise ratio is high enough, T-DGD can perform as well as the vanilla DGD while reducing communication complexity to logarithmic dependence on ambient dimension $d$.

## 2 Related works

We categorize the existing works most relevant to our study into three main groups.

- **Implicit regularizations for sparse regression.** The recent study in [15] reparameterized the model parameter through overparameterized Hadamard product and discovered encouraged empirical performance by the first-order optimization algorithms. The statistical and convergence guarantees for this phenomenon are established in [36, 42] under mild conditions. Woodworth et al. [38] studied the impact of initialization scale on solutions. Scott et al.[27] demonstrated the benefit of stochasticity of SGD in sparse regression and explored the impact of momentum in [25]. The more recent process in understanding the linear diagonal networks can be found in references [7, 10, 12, 19, 23, 26, 43]. To the best of our knowledge, existing works have only discussed implicit regularizations induced by centralized optimization methods in linear diagonal networks. However, the question of whether decentralized algorithms induce implicit regularizations, and what type of implicit regularization they may induce has not been studied so far.

- **Decentralized sparse regression with explicit regularization.** For estimating ground truth $w^\star$ in high-dimensional sparse linear regression under the decentralized setting, Ji et al. [18] proposed DGD-CTA algorithm for tackling LASSO objective with consensus penalty and proved linear convergence rate to the neighbor of the statistical optimal estimator, but the convergence rate has polynomial dependence on ambient dimension $d$. Further Sun et al. [31] proposed the NetLASSO based on the gradient tracking method and obtained $d$-independent convergence rate and optimal statistical accuracy. To complement work [18], Ji et al. proposed DGD-ATC by mixing the local gradients along iterations and achieved logarithmic dependence on $d$ in [17]. Maros et al. [21] proposed DGD$^2$ method based on a double mixing for solving decentralized LASSO and obtained similar theoretical guarantees in [17, 31]. To improve the computation efficiency, Maros et al. [20] integrated accelerated proximal gradient descent with gradient tracking to solve decentralized LASSO. Despite these developments, it remains unclear whether leveraging the unregularized overparameterization and implicit regularization of decentralized optimization methods can achieve the optimal statistical guarantee over mesh networks.

- **Implicit regularizations of decentralized optimization.** Implicit bias or regularizations of centralized optimization methods for overparameterized models have been extensively studied [13, 30], but only a few works have investigated the implicit regularization of decentralized optimization methods. Richards et al. [29] studied the implicit regularization for decentralized stochastic gradient descent for solving general unregularized convex problems. Zhu et al. [44] demonstrated that decentralized stochastic gradient descent implicitly executes the sharpness-aware minimization algorithm for general non-convex problems. Taheri et al. [33] studied the implicit regularization of DGD in overparameterized classification for separable data. Recent work [22] demonstrated the implicit regularization of the DGD$^2$ in solving the overparameterized matrix sensing problem. Different from these works, we establish the statistical and computational results for specific non-convex sparse regression problem.

## 3 Preliminaries

In this section, we will introduce the basic notations used in this paper, and then formulate the DGD for solving the problem (2). Finally, we provide the necessary assumptions and definitions for the decentralized sparse regression problem.

### 3.1 Notations

Throughout this paper, we use $[m]$ to denote set $\{1, \cdots, m\}$ for given positive integer $m$, $\mathbf{1}_d$ denotes $d$-dimensional vector that all elements are one and $\boldsymbol{I}_d$ denotes $d$-dimensional identity matrix. For ground truth parameter $\boldsymbol{w}^\star$, the relevant notations are support set $\mathcal{S} := \{j | w_j^\star \neq 0\}$, positive support set $\mathcal{S}^+ := \{j | w_j^\star > 0\}$, negative support set $\mathcal{S}^- := \{j | w_j^\star < 0\}$ and non-support set $\mathcal{S}^c := \{j | w_j^\star = 0\}$, $w_{\max}^\star := \max_{j \in \boldsymbol{\mathcal{S}}} |w_j^\star|$ and $w_{\min}^\star := \min_{j \in \boldsymbol{\mathcal{S}}} |w_j^\star|$. $\forall \boldsymbol{x} \in \mathbb{R}^d, \boldsymbol{x}_{\mathcal{S}} := \mathbf{1}_{\mathcal{S}} \odot \boldsymbol{x}$ where $\mathbf{1}_{\mathcal{S}}$ denotes a vector equal to one for all coordinates $j \in \mathcal{S}$ and equal to zero everywhere else. Symbol "$\odot$" denotes Hadamard product that $(\boldsymbol{a} \odot \boldsymbol{b})_j = a_j b_j, \forall \boldsymbol{a}, \boldsymbol{b} \in \mathbb{R}^d$. The averaged signal is defined as $\overline{\boldsymbol{w}}^t := \frac{1}{m} \sum_{i=1}^{m} \boldsymbol{w}^{t,i}$ and similar notations can be extended to $\overline{\boldsymbol{u}}^t, \overline{\boldsymbol{v}}^t$.

$\boldsymbol{X} := [\boldsymbol{X}_1; \cdots; \boldsymbol{X}_m]$ denotes the concatenated sample matrix, where each row of $\boldsymbol{X}_i$ represents one feature vector in agent $i$. $\|\cdot\|$ denotes the Frobenius norm for vector and the spectral norm (maximum singular value) for matrix. $\|\boldsymbol{A}\|_\infty := \max_{i,j} |A_{ij}|$ denotes infinity norm. We use $a = \mathcal{O}(b)$ to denote that inequality $a \leq Cb$ holds with some absolute constants $C$ that do not depend on any parameters of the problem. The notation $a \lesssim b$ shares the same meaning as $a = \mathcal{O}(b)$. Finally, we use $a \gtrsim b$ if there exists a universal constant $c$ such that $a \geq cb$.

### 3.2 Method and Assumptions

We focus on DGD solving problem (2) over mesh network, modeled as an undirected graph $\mathcal{G} = \{\mathcal{V}, \mathcal{E}\}$ where nodes $\mathcal{V} = \{1, \cdots, m\}$ represent the set of agents and edges $\mathcal{E} \subset \mathcal{V} \times \mathcal{V}$ represent the communication links. An unordered pair $\{i, j\}$ is included in $\mathcal{E}$ if and only if there is a bidirectional communication link between agent $i$ and $j$. The set of one-hop neighbors for agent $i$ is denoted by $\mathcal{N}_i := \{j \in \mathcal{V} | (i, j) \in \mathcal{E}\} \bigcup \{i\}$.

DGD allows each agent to independently update its parameters based on local gradient descent and then synchronize with neighboring agents by weighted averaging these updates. The recursive iteration of DGD for each agent is described as follows.

$$\boldsymbol{u}^{t+1,i} = \sum_{j=1}^{m} W_{ij} \left( \boldsymbol{u}^{t,j} - \eta \frac{4}{n} \boldsymbol{u}^{t,j} \odot \left( \boldsymbol{X}_j^T \left( \boldsymbol{X}_j \left( \boldsymbol{u}^{t,j} \odot \boldsymbol{u}^{t,j} - \boldsymbol{v}^{t,j} \odot \boldsymbol{v}^{t,j} \right) - \boldsymbol{y}_j \right) \right) \right), \ \forall i \in [m];$$

$$\tag{3}$$

$$\boldsymbol{v}^{t+1,i} = \sum_{j=1}^{m} W_{ij} \left( \boldsymbol{v}^{t,j} + \eta \frac{4}{n} \boldsymbol{v}^{t,j} \odot \left( \boldsymbol{X}_j^T \left( \boldsymbol{X}_j \left( \boldsymbol{u}^{t,j} \odot \boldsymbol{u}^{t,j} - \boldsymbol{v}^{t,j} \odot \boldsymbol{v}^{t,j} \right) - \boldsymbol{y}_j \right) \right) \right), \ \forall i \in [m],$$

$$\tag{4}$$

where the $\boldsymbol{w}^{t,i} := \boldsymbol{u}^{t,i} \odot \boldsymbol{u}^{t,i} - \boldsymbol{v}^{t,i} \odot \boldsymbol{v}^{t,i}$ denotes the local estimator in agent $i$ at $t^{th}$ iteration, the initialization is $\boldsymbol{u}^{0,i} = \boldsymbol{v}^{0,i} = \alpha \mathbf{1}_d, \forall i \in [m]$ and $\eta$ is constant step size. $\boldsymbol{W}$ is the nonnegative weight mixing matrix for the undirected mesh network, where $W_{ij} > 0$ if there is a link between agents $i$ and $j$, and $W_{ij} = 0$ otherwise. The mixing matrix $\boldsymbol{W}$ related to the undirected graph satisfies the following assumption.

**Assumption 1.** *The communication network $\mathcal{G}$ is connected. The weight matrix $\boldsymbol{W} = [w_{ij}]_{i,j=1}^{m}$ for this graph has the following properties: (i) $w_{ij} = 0$ for all pairs $(i, j)$ that are not in $\mathcal{E}$; (ii) it is double stochastic that $\mathbf{1}_m^T \boldsymbol{W} = \mathbf{1}_m^T$ and $\boldsymbol{W} \mathbf{1}_m = \mathbf{1}_m$; (iii) the spectral gap $\rho := \left\| \boldsymbol{W} - \frac{1}{m} \mathbf{1}_m \mathbf{1}_m^T \right\| \leq 1$.*

This assumption is common in decentralized optimization literature [17, 22]. We need the following RIP condition which is a key condition to obtain the optimal estimator for sparse regression.

**Definition 1.** *The global design matrix $\boldsymbol{X}/\sqrt{N} \in \mathbb{R}^{N \times d}$ satisfies the $(\delta, s)$-Restricted Isometry Property (RIP) if for any $s$-sparse vector $\boldsymbol{w} \in \mathbb{R}^d$, there is $(1 - \delta) \|\boldsymbol{w}\|^2 \leq \left\| \boldsymbol{X} \boldsymbol{w}/\sqrt{N} \right\|^2 \leq (1 + \delta) \|\boldsymbol{w}\|^2$.*

The RIP condition was first introduced in the compressed sensing literature in [8] which is a little more restrictive condition to achieve optimal statistical rate than the restricted eigenvalue condition in [1]. We inherit this assumption in the centralized setting [36, 42] to achieve optimal estimator error under the condition that parameter $\delta$ is upper bounded. Besides the global RIP condition, we have the following local RIP condition for local design matrices $\{\boldsymbol{X}_i\}_{i=1}^{m}$.

**Definition 2.** *The local design matrices $\boldsymbol{X}_1/\sqrt{n}, \cdots, \boldsymbol{X}_m/\sqrt{n} \in \mathbb{R}^{n \times d}$ satisfy the local $(\delta_{\max}, s)$-(RIP) condition, if for any $s$-sparse vector $\boldsymbol{w} \in \mathbb{R}^d$ and any local design matrix $\boldsymbol{X}_i/\sqrt{n}$, there is $(1 - \delta_{\max}) \|\boldsymbol{w}\|^2 \le \|\boldsymbol{X}_i \boldsymbol{w}/\sqrt{n}\|^2 \le (1 + \delta_{\max}) \|\boldsymbol{w}\|^2.$*

The definition of the local RIP condition is just for ease of proof presentation, as we do not necessitate any upper bound on the local RIP parameter $\delta_{\max}$.

## 4 Main Result

Based on the above method and assumptions, we now give theoretical guarantees of DGD in solving problem (2) for sparse regression problem (1) as follows.

**Theorem 1.** *Considering the sequence generated by (3) and (4) based on DGD for solving problem (2) and $\forall \epsilon > 0$, if the global design matrix $\boldsymbol{X}/\sqrt{N}$ satisfies $(\delta, s+1)$-RIP condition with bounded RIP parameter $\delta \lesssim \frac{1}{\sqrt{s}}$, the local design matrices $\{\boldsymbol{X}_i/\sqrt{n}\}_{i=1}^m$ satisfy local $(\delta_{\max}, s+1)$-RIP condition, and the mesh network satisfies assumption 1, the initialization satisfies $\alpha \lesssim \min\left\{1, \frac{\epsilon^2}{(12d+1)^2}, \frac{\epsilon}{w_{\max}^\star}, \frac{\zeta}{6(w_{\max}^\star)^2}, \frac{w_{\min}^\star}{4}\right\}$, the constant stepsize $\eta$ satisfies*

$$\eta \lesssim \min\left\{\frac{1 - \sqrt{\rho}}{64\sqrt{\rho}w_{\max}^\star}, \frac{\log \frac{1}{\alpha^4}\left(1 - \sqrt{\frac{1+\sqrt{\rho}}{2}}\right)}{4w_{\max}^\star}, \frac{1 - \left(\frac{1+\sqrt{\rho}}{2}\right)^{\frac{1}{4}}}{w_{\max}^\star}\right\}, \tag{5}$$

*and the spectral gap $\rho$ satisfies*

$$\rho^{\frac{1}{4}} \lesssim \min\left\{\frac{1}{\sqrt{s}\delta_{\max} + 1}, \frac{\delta}{8\delta_{\max}}, \frac{\left\|\frac{\boldsymbol{X}^T\boldsymbol{\xi}}{N}\right\|_\infty}{8\max_i\left\|\frac{\boldsymbol{X}_i^T\boldsymbol{\xi}_i}{n}\right\|_\infty}\right\}, \tag{6}$$

*then after running $t = \mathcal{O}\left(\frac{1}{\eta\zeta}\log\frac{1}{\alpha}\right)$ iterations. There would be*

$$\left|\overline{w}_j^t - w_j^\star\right| \le \begin{cases} \mathcal{O}(\varsigma) & \text{if } j \in \mathcal{S} \text{ and } w_{\min}^\star \le \mathcal{O}(\varsigma) \\ \mathcal{O}\left(\max\left\{\left|\left(\frac{\boldsymbol{X}^T\boldsymbol{\xi}}{N}\right)_j\right|, \delta\sqrt{s}\left\|\frac{\boldsymbol{X}^T\boldsymbol{\xi}}{N}\odot\mathbf{1}_\mathcal{S}\right\|_\infty, \epsilon\right\}\right) & \text{if } j \in \mathcal{S} \text{ and } w_{\min}^\star \ge \mathcal{O}(\varsigma) \\ \mathcal{O}(\sqrt{\alpha}) & \text{if } j \notin \mathcal{S}, \end{cases} \tag{7}$$

*where $\varsigma := \max\left\{\left\|\frac{\boldsymbol{X}^T\boldsymbol{\xi}}{N}\right\|_\infty, \epsilon\right\}, \zeta := \max\left\{\frac{w_{\min}^\star}{5}, 960\varsigma\right\}.$*

- *Mechanism to promote sparsity.* The consensus errors induced from decentralized network complicate the multiplicative updates, which becomes inexact multiplicative updates as $\overline{\boldsymbol{u}}^{t+1} = \overline{\boldsymbol{u}}^t \odot \left(1 - 4\eta\left(\overline{\boldsymbol{u}}^t \odot \overline{\boldsymbol{u}}^t - \boldsymbol{w}^\star + \hat{\boldsymbol{p}}^t + \hat{\boldsymbol{b}}^t\right)\right) + \boldsymbol{e}^t$. Compared with the exact multiplicative updates of GD in [36], the challenge is that the extra error term $\boldsymbol{e}^t$ outside of the multiplication prevents applying the centralized analysis trivially. In addition, the perturbation error terms $\hat{\boldsymbol{p}}^t, \hat{\boldsymbol{b}}^t$ within the multiplication are much more complicated than that of the centralized setting due to additional multiple consensus errors. This requires bounding the consensus error terms carefully, which should control the complicated perturbation errors $\hat{\boldsymbol{p}}^t, \hat{\boldsymbol{b}}^t, \boldsymbol{e}^t$ not to be large. Thus, we can use network connectivity to control the consensus errors to bound these three perturbation errors small enough to make the distance between two trajectories obtained by inexact and exact multiplicative updates within statistical accuracy, which can promote sparsity in the decentralized setting. The detailed theoretical mechanism of promoting sparsity has been demystified in Proposition 3.
- *Statistical Guarantee.* Based on the result in (7) and conditions in (5) and (6), we can observe that if the initialization $\alpha$ is small enough and network connectivity is sufficiently well, the DGD with early stopping can obtain the desired estimator for sparse ground truth parameter $\boldsymbol{w}^\star$ that achieves the same order of statistical error as the centralized setting in [36]. The formula in (7) not only

illustrates that we establish the network-independent estimator error bound but also inherits the benefit of implicit regularization, which indicates that if the signal-to-noise is high enough, the statistical error is independent of ambient dimension $d$. In contrast, existing results in decentralized LASSO methods [17, 18, 21, 31], have consistent dependence on $d$ in any case.

- *Computational Complexity.* The iteration complexity of early stopping is network-dependent that is because the $t = \mathcal{O}\left(\frac{1}{\eta\zeta}\log\frac{1}{\alpha}\right)$ has the dependence on stepsize, which should satisfy the condition in (5). This suggests that poorer network connectivity leads to higher computational complexity. Although the initialization has no dependence on network connectivity, $\frac{1}{\alpha}$ has polynomial dependence on $d$, and the dependence of complexity on $d$ is just logarithmic, which is similar to DGD in solving LASSO in [17, 21] and improves the polynomial dependence on $d$ in [18].

- *Dependence on network connectivity.* For accurate estimation, it is essential that the network should be well-connected, as specified in condition (6). When this condition is not satisfied, we can run multiple rounds of communication per iteration. It is observable that the smaller ratio between the global RIP parameter $\delta$ and the local RIP parameter $\delta_{\max}$, and smaller ratio between the local noise and the global noise magnitude, necessitate a higher degree of network connectivity. This can be understood from the perspective of heterogeneity, where smaller ratios indicate a significant disparity between local and global design matrices. Consequently, condition (6) is reasonable as it suggests that higher levels of heterogeneity necessitate improved network connectivity. In numerical experiments, we can observe that if $\rho$ does not satisfy the condition as (6), obtaining optimal statistical error is not achievable, which indicates the optimal statistical error undergoes a phase transition with the network connectivity.

Our results demonstrate the benefit of overparameterization for DGD. Theorem 1 shows that standard DGD is sufficient to provide a satisfied statistical estimator with efficient computation without gradient correction techniques. This finding challenges the widely held belief in decentralized optimization literature that extra techniques like gradient tracking and other gradient-correction-based methods are necessary for heterogeneous scenarios [39, 32]. The following corollary considers the well-known instance where the design matrix and noise are generated from sub-Gaussian distribution, which indicates that DGD with early stopping can achieve the minimax optimal statistical rate under the $\ell_2$ metric.

**Corollary 1.** *Suppose that entries of global design matrix $\boldsymbol{X}$ generated from i.i.d 1-sub-Gaussian distribution, and the total sample size satisfies $N \gtrsim s\left(s\log\frac{ed}{s} + \log\frac{dN}{m}\right)$. The noise vector $\boldsymbol{\xi}$ is generated from independent $\sigma^2$-sub-Gaussian entries, and the initialization is set as Theorem 1 with*

$$\epsilon = \mathcal{O}\left(\sigma\sqrt{\frac{\log d}{N}}\right). \text{ If the spectral gap satisfies } \rho \lesssim \frac{1}{m^4} \text{ and stepsize is set as } \eta = \mathcal{O}\left(\frac{1-\left(\frac{1+\sqrt{\rho}}{2}\right)^{\frac{1}{4}}}{w_{\max}^{\star}}\right),$$

*then after running* $t = \mathcal{O}\left(\dfrac{w_{\max}^{\star}\sqrt{N}}{\sigma\sqrt{\log d}\left(1-\left(\frac{1+\sqrt{\rho}}{2}\right)^{\frac{1}{4}}\right)}\log\frac{1}{\alpha}\right)$ *iterations, the sequence generated by* (3)

*and* (4) *based on DGD for solving problem* (2) *would obtain estimator that* $\left\|\overline{\boldsymbol{w}}^t - \boldsymbol{w}^{\star}\right\| \lesssim \sigma\sqrt{\frac{s\log d}{N}}$ *with probability at least* $1 - \frac{3}{8d^3}$.

Corollary 1 indicates that in the sub-Gaussian setting, network-independent statistical error obtained by DGD matches optimal rate $\mathcal{O}\left(\sigma\sqrt{\frac{s\log d}{N}}\right)$ under $\ell_2$ metric in the centralized setting [28]. In this context, the condition for network connectivity implies that the smaller $\rho$ is required as the number of agents $m$ increases. This is reasonable because when the total sample size $N$ is fixed, an increase in the number of agents results in fewer samples assigned to each agent. Consequently, better network connectivity is necessary to achieve optimal estimation.

## 5 Communication Efficient DGD via Truncation

It is apparent that iterations in (3) and (4) of DGD, each agent has to transmit two $d$-dimensional vectors $\boldsymbol{u}^{t,i}$ and $\boldsymbol{v}^{t,i}$ to its neighboring agents per iteration. Because we are considering the high-dimensional regime where the feature has ultra-high dimension, which leads to the $\mathcal{O}\left(d\cdot\frac{1}{\eta\zeta}\log\frac{1}{\alpha}\right)$ high communication complexity (in terms of the bits transmitted) for DGD. The primary idea is

whether it is possible to transmit fewer partial elements instead of the entire $d$-dimensional vectors for $\boldsymbol{u}^{t,i}, \boldsymbol{v}^{t,i}$ in all rounds of communication. Since all elements of $\boldsymbol{u}^{t,i}$ and $\boldsymbol{v}^{t,i}$ equal to $\alpha$ at initialization, we can utilize the one step of local gradient descent step in each agent to distinguish the support set and non-support based on changes of magnitudes for each element. The intuition is that the elements on the support would grow more rapidly than those on the non-support. Thus, we propose the **T**runcated **D**ecentralized **G**radient **D**escent (T-DGD) as

$$\boldsymbol{u}^{t+1,i} = \sum_{j=1}^{m} W_{ij} \cdot \mathrm{Trun}_s \left( \left( \boldsymbol{u}^{t,j} - \eta \frac{4}{n} \boldsymbol{u}^{t,j} \odot \left( \boldsymbol{X}_j^T \left( \boldsymbol{X}_j \left( \boldsymbol{u}^{t,j} \odot \boldsymbol{u}^{t,j} - \boldsymbol{v}^{t,j} \odot \boldsymbol{v}^{t,j} \right) - \boldsymbol{y}_j \right) \right) \right) \right);$$

$$\boldsymbol{v}^{t+1,i} = \sum_{j=1}^{m} W_{ij} \cdot \mathrm{Trun}_s \left( \left( \boldsymbol{v}^{t,j} + \eta \frac{4}{n} \boldsymbol{v}^{t,j} \odot \left( \boldsymbol{X}_j^T \left( \boldsymbol{X}_j \left( \boldsymbol{u}^{t,j} \odot \boldsymbol{u}^{t,j} - \boldsymbol{v}^{t,j} \odot \boldsymbol{v}^{t,j} \right) - \boldsymbol{y}_j \right) \right) \right) \right),$$
$$(8)$$

for $\forall i \in [m]$, where $\mathrm{Trun}_s(\boldsymbol{x})$ is the operator that preserves only the $s$ largest magnitude elements of the vector $\boldsymbol{x}$ while setting all other elements to zero. The following proposition shows the benefit of T-DGD in sparse regression under proper conditions.

**Proposition 1.** *With the same setup in Corollary 1, if the ground truth $\boldsymbol{w}^\star$ satisfies $\frac{w_{\min}^\star}{2} \gtrsim \sqrt{s}\delta_{\max} w_{\max}^\star + \sigma\sqrt{\frac{\log d}{n}}$, then the sequence generated by T-DGD as* (8) *for solving problem* (2) *would obtain estimator that $\left\| \overline{\boldsymbol{w}}^t - \boldsymbol{w}^\star \right\| \lesssim \sigma\sqrt{\frac{s \log d}{N}}$ with probability at least $1 - \frac{3}{8d^3}$. However, the communication complexity in terms of transmitted bits would be at most $\mathcal{O}\left( s \cdot \left( w_{\max}^\star \sqrt{N} \right) \middle/ \left( \sigma\sqrt{\log d} \left( 1 - \left( \frac{1+\sqrt{\rho}}{2} \right)^{\frac{1}{4}} \right) \right) \right) \log \frac{1}{\alpha}$.*

To ensure condition $\frac{w_{\min}^\star}{2} \gtrsim \sqrt{s}\delta_{\max} w_{\max}^\star + \sigma\sqrt{\frac{\log d}{n}}$ satisfied, it is necessary to require that each agent has sufficient samples and SNR is high enough such that $\delta_{\max} \lesssim \frac{w_{\min}^\star}{\sqrt{s}w_{\max}^\star}$ and $\sigma\sqrt{\frac{\log d}{n}} \lesssim w_{\min}^\star$. This proposition enables each agent to transmit only $s$ elements of $d$-dimensional vector per communication round, which can achieve optimal statistical rate and eliminate the $d$-linear increasing communication complexity. The result in Proposition 1 validates the usefulness of the Hadamard product over-parameterization in decentralized gradient-based optimization.

## 6 Numerical Results

This section conducts the experimental studies to evaluate the theoretical findings of DGD and T-DGD for solving problem (2) in Subsection 6.1, Subsection 6.2, respectively. In Subsection 6.3, we compare the effectiveness of implicit regularization of DGD with explicit regularization based decentralized methods. The communication networks $\mathcal{G}$ are generated from Erdős Rényi (ER) graphs with link activation under given probabilities. By default, unless stated otherwise, all the design matrices $\boldsymbol{X}$ have i.i.d. standard Gaussian elements, noise $\boldsymbol{\xi}$ follows i.i.d. $\mathcal{N}(0, 0.5^2)$ distribution, and the magnitudes of elements on support $\mathcal{S}$ are 1. All experiments are conducted on 12th Gen Intel(R) Core(TM) i7-12700@2.10GHz processor and 16.0GB RAM under Windows 11 system.

### 6.1 Simulations on DGD

We organize the experiments as follows: 1) We visualize the dynamics of averaged variables and consensus errors that allow us to evaluate the implicit regularization of DGD and the soundness of our technical analysis. 2) We check whether DGD can achieve optimal statistical error, the impacts of ambient dimension $d$ and initialization scale $\alpha$ on statistical and computational properties. 3) We evaluate the condition of (6) that reveals the relationship between network connectivity and network scale for achieving the statistical accuracy of centralized setting.

- **Dynamics of $\overline{\boldsymbol{w}}^t, \overline{\boldsymbol{u}}^t, \overline{\boldsymbol{v}}^t$ and $\boldsymbol{u}^{t,i} - \overline{\boldsymbol{u}}^t, \boldsymbol{v}^{t,i} - \overline{\boldsymbol{v}}^t$.** In this case, we set $d = 2000, s = 10, m = 10, N = 400, \rho = 0.1778, \alpha = 10^{-6}$. Fig. 1(a) demonstrates the convergence of averaged $\overline{\boldsymbol{w}}^t$ in DGD, showing successful convergence of elements on support $\mathcal{S}$ and maintenance of small

magnitudes for elements on non-support $\mathcal{S}^c$. Fig. 1(b) and Fig. 1(c) further illustrate how DGD utilizes $\overline{\boldsymbol{u}}^t$ and $\overline{\boldsymbol{v}}^t$ to fit parameters on positive and negative support, respectively. Additionally, the magnitudes of $\overline{\boldsymbol{u}}^t$ and $\overline{\boldsymbol{v}}^t$ on non-positive and non-negative support remain small enough as the initialization. Consensus errors $\boldsymbol{u}^{t,i} - \overline{\boldsymbol{u}}^t$ and $\boldsymbol{v}^{t,i} - \overline{\boldsymbol{v}}^t$ are depicted in Fig. 1(d) and Fig. 1(e), respectively. The trends in these curves correspond to the magnitudes of the model parameter, affirming the validity of our analysis.

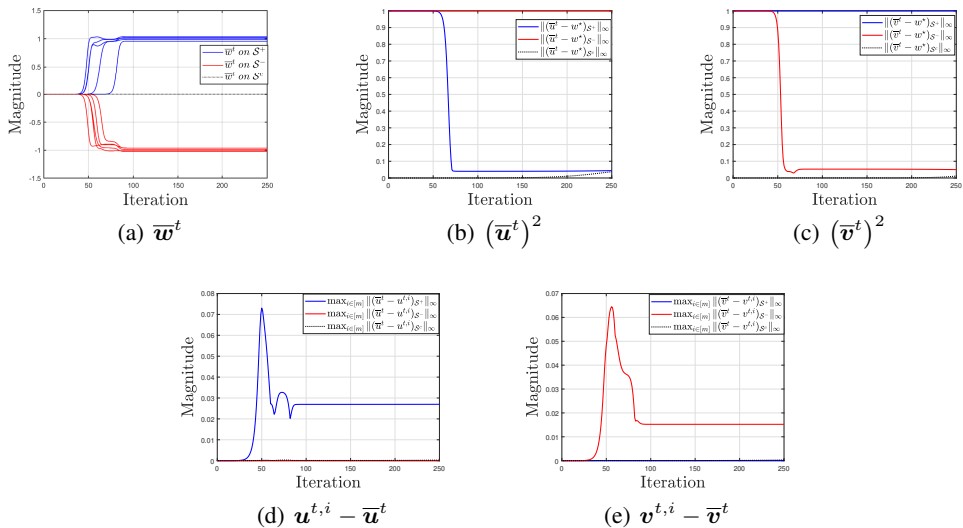

Figure 1: Dynamics of avergaed variables and consensus errors.

- **Impact of $d$ and $\alpha$ on optimal estimation.** We vary the dimension of $d$ $(4 \times 10^2, 4 \times 10^3, 4 \times 10^4)$ to access effect of $d$ on both statistical and computational properties. With $s = \lceil \log d \rceil$ and $N$ chosen to satisfy $s \log d/N \approx 0.25$, we aim to maintain the same order optimal statistical error $\mathcal{O}\left(\sigma\sqrt{\frac{s \log d}{N}}\right)$. Testing is conducted on two networks with $m = 20$ but different $\rho$. For each $d$, we select the maximum initialization $\alpha$ that achieves optimal statistical error, resulting in $\alpha = 10^{-8}$ for $d = 4 \times 10^2$, $\alpha = 10^{-8.5}$ for $d = 4 \times 10^3$ and $\alpha = 10^{-9}$ for $d = 4 \times 10^4$. The results for $\rho = 0.1778$ and $\rho = 0.7519$ are displayed in Fig. 2(a) and Fig. 2(b), respectively. It is observable that DGD obtains estimators with statistical error matching that of the centralized setting, with computational complexity remaining largely unaffected by ambient dimension $d$ across different network conditions. To assess the influence of $\alpha$, we set $d = 2000, s = 10, m = 20, N = 400$, and different values for $\alpha$ on network with $m = 20, \rho = 0.1778$. The results in Fig. 2(c) illustrate that it is necessary to use small enough initialization to obtain optimal estimator.

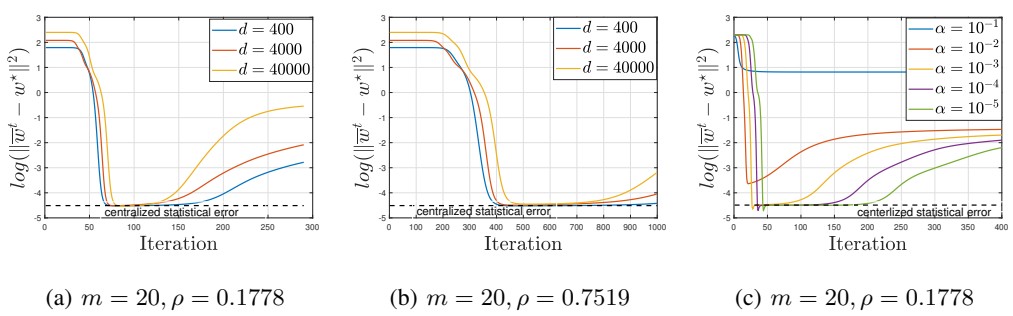

(a) $m = 20, \rho = 0.1778$      (b) $m = 20, \rho = 0.7519$      (c) $m = 20, \rho = 0.1778$

Figure 2: Impact of ambient dimension $d$ and initialization $\alpha$.

- **Dependence on $\rho$ and $m$.** We set $d = 2000, s = 10, N = 200, \alpha = 10^{-6}$ and test on networks with different numbers of agents. The results are shown in Fig. 3 where Fig. 3(a) and Fig.

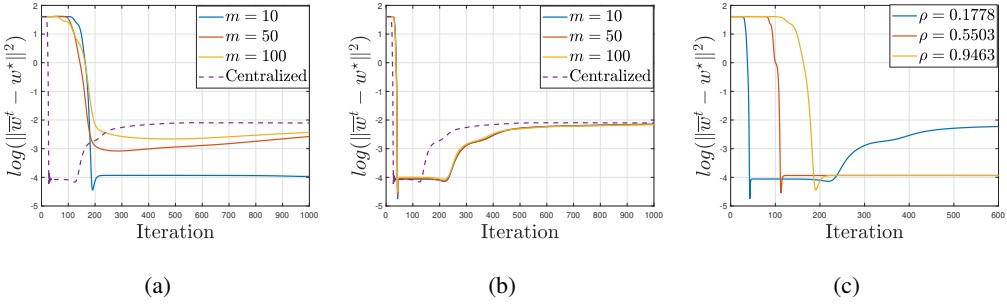

(a)                                        (b)                                        (c)

Figure 3: (a) $\rho = 0.9400$; (b) $\rho = 0.1778$; (c) $m = 10$.

3(b) display the performance with varied numbers of agents under fixed $\rho = 0.9400$ and fixed $\rho = 0.1778$, respectively. Fig. 3(a) indicates that DGD would not obtain the optimal estimator as the centralized setting when the number of agents is large which violates the condition in (6). When network connectivity is sufficiently connected as $\rho = 0.1778$, Fig. 3(b) conveys that this can allow a larger scale of agents to attain optimal statistical error. In Fig. 3(c), we fix $m = 10$ and observe the phenomenon under varied $\rho$ by choosing proper stepsizes to achieve the best statistical error. Fig. 3(c) illustrates that $\rho$ would influence the stopping time when DGD can obtain the optimal estimator. The worse the network, the more iterations it takes to find the optimal estimator.

## 6.2   Simulations on T-DGD

In this section, we evaluate the effectiveness of T-DGD. Initially, we vary the values of $N$, keeping the other parameters consistent with the simulations in Fig. 1. Fig. 4(a) illustrates that when each agent has inadequate samples ($N = 100, n = 10$), T-DGD would fail in achieving optimal estimation. However, with increasing local samples ($N = 400, n = 10$), T-DGD matches DGD in both statistical accuracy and convergence performance. Subsequently, we set the magnitudes of the ground truth on support as 100 and $N = 300$. The performance is depicted as dashed lines in Fig. 4(c), indicating failure of T-DGD under higher noise level ($\sigma = 0.5$). We further reduce the noise magnitude to $\sigma = 0.1$, and solid lines in Fig. 4(c) demonstrate the usefulness of T-DGD in sparse regression. These observations validate the statement in Proposition 1.

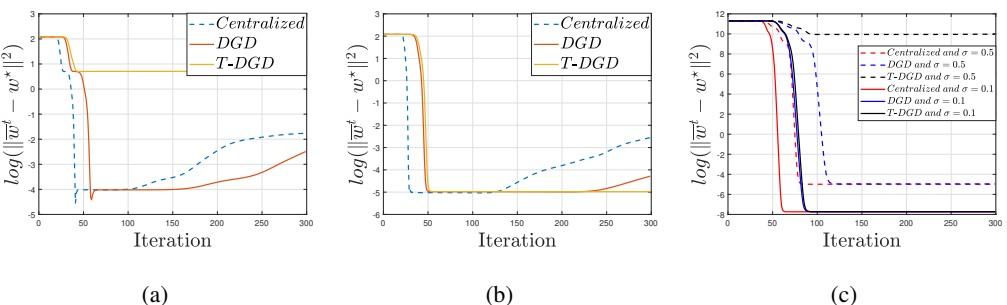

(a)                                        (b)                                        (c)

Figure 4:  (a) $N = 100$; (b) $N = 400$; (c) Different noise intensities.

## 6.3   Comparison with explicit regularization

We have compared our proposed method with three existing decentralized methods, namely: CTA-DGD (LASSO) [18], ATC-DGD (LASSO) [17], and DGT (NetLASSO) [31]. These methods are all derived based on the LASSO formulation with explicit regularization. The numerical results presented in Fig. 5 compare all four methods under three different network connectivity settings. For each method, we tuned the step size to achieve the best performance. Our proposed method demonstrated the best recovery performance in all network settings with minimal iterations.

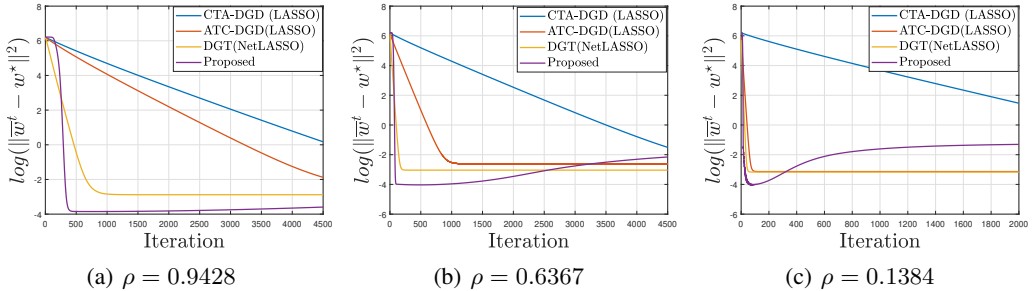

| (a) $\rho = 0.9428$ | (b) $\rho = 0.6367$ | (c) $\rho = 0.1384$ |

Figure 5: Comparison with decentralized sparse solvers under varying communication network. The setting is $d = 1000, k = 5, m = 50, N = 280, \sigma = 0.5$ and magnitude of sparse signal is 10.

We further compared T-DGD with existing methods with truncated versions of existing methods: Trun-CTA-DGD (LASSO), Trun-ATC-DGD (LASSO), and Trun-DGT (NetLASSO) which use the same Top-$s$ truncation operator. As shown in Fig. 6, our proposed method is the only one to achieve successful recovery, while all other truncated decentralized methods failed. The numerical evidence demonstrates that naively combining sparsification with decentralized algorithms is *not granted to converge*. This is precisely one of the motivations of this work: to provide communication-efficient algorithms with both provably statistical and computational guarantees. This result also demonstrates the unique benefit of overparameterization and implicit regularization for decentralized learning setting, which has not been explored in the literature of learning theory.

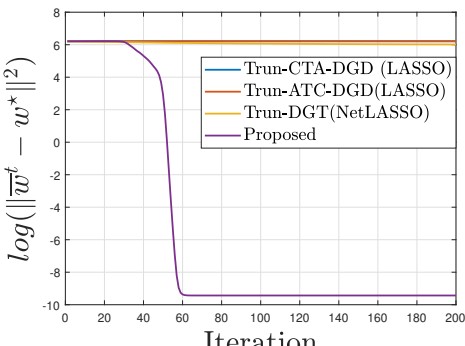

Figure 6: Truncated version: comparison with truncated decentralized sparse solvers. The setting is $d = 1000, s = 5, m = 50, N = 550, \sigma = 0.1, \rho = 0.2458$ and magnitude of sparse signal is 10.

## 7    Conclusion

In this paper, we study the implicit regularization of decentralized gradient descent for decentralized sparse regression in the unpenalized and overparameterized regimes. We establish both statistical and computational guarantees for the decentralized estimator under mild conditions of network connectivity, underscoring the utility of DGD in addressing overparameterized models. Furthermore, the proposed truncated DGD (T-DGD) offers a promising idea to reduce communication complexity while maintaining performance. In future work, exploring the possibility of relaxing the RIP condition in our assumption and leveraging the restricted eigenvalue condition to achieve optimal estimator in the decentralized setting is an interesting topic. Additionally, investigating alternative forms of implicit regularizations in decentralized optimization algorithms for more complicated overparameterized models is another intriguing direction.

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

# A Appendix.A

Section A.1, A.2 give the additional notations and useful basic lemmas, respectively. Section A.3 provides the properties for the simplified setting where all the elements of ground truth parameter $w^\star$ are non-negative. Section A.4 gives the key proposition for the general case where $w^\star$ contains both positive and negative elements on support $\mathcal{S}$. Final Section A.5 concludes the proofs for the Theorem 1, Corollary 1, and Proposition 1 in the main paper based on Proposition 3 in Section A.4.

## A.1 Full Notations

In addition to introduced notations from Section 3.1 in the main paper, we need additional notations to present the proof. The consensus error is denoted as $\boldsymbol{\Delta}_u^{t,i} := \boldsymbol{u}^{t,i} - \overline{\boldsymbol{u}}^t$. $\boldsymbol{\Delta}_{u,\mathcal{S}^+}^{t,i} = \boldsymbol{\Delta}_u^{t,i} \odot \mathbf{1}_{\mathcal{S}^+}$ and notations $\boldsymbol{\Delta}_{u,\mathcal{S}^-}^{t,i}, \boldsymbol{\Delta}_{u,\mathcal{S}^c}^{t,i}$ can be defined similarly. The additional notations are defined as

$$
\begin{aligned}
\boldsymbol{U}^t &:= \left[\boldsymbol{u}^{t,i}, \cdots, \boldsymbol{u}^{t,m}\right], \quad \boldsymbol{\Delta}_u^t := \left[\boldsymbol{\Delta}_u^{t,1}, \cdots, \boldsymbol{\Delta}_u^{t,m}\right], \quad \overline{\boldsymbol{U}}^t := \overline{\boldsymbol{u}}^t \mathbf{1}_m^T; \\
\nabla_u \boldsymbol{f}\left(\overline{\boldsymbol{U}}^t, \overline{\boldsymbol{V}}^t\right) &:= \left[\nabla_u f_1(\overline{\boldsymbol{u}}^t, \overline{\boldsymbol{v}}^t), \cdots, \nabla_u f_m(\overline{\boldsymbol{u}}^t, \overline{\boldsymbol{v}}^t)\right], \quad \nabla_u \boldsymbol{F}(\overline{\boldsymbol{U}}^t, \overline{\boldsymbol{V}}^t) := \nabla_u F(\overline{\boldsymbol{u}}^t, \overline{\boldsymbol{v}}^t) \mathbf{1}_m; \\
\nabla_u \boldsymbol{f}\left(\boldsymbol{U}^t, \boldsymbol{V}^t\right) &:= \left[\nabla_u f_1(\boldsymbol{u}^{t,1}, \boldsymbol{v}^{t,1}), \cdots, \nabla_u f_m(\boldsymbol{u}^{t,m}, \boldsymbol{v}^{t,m})\right]; \\
\nabla_u \boldsymbol{f}\left(\overline{\boldsymbol{U}}^t, \boldsymbol{V}^t\right) &:= \left[\nabla_u f_1(\overline{\boldsymbol{u}}^t, \boldsymbol{v}^{t,1}), \cdots, \nabla_u f_m(\overline{\boldsymbol{u}}^t, \boldsymbol{v}^{t,m})\right].
\end{aligned}
\tag{9}
$$

Above definitions can be also extended to variable $v$ similarly.

## A.2 Premilary Lemmas

**Lemma 1.** *(Theorem 6.5 in [37]) Consider the random feature vector $\boldsymbol{x} \in \mathbb{R}^d$ that all entries obey i.i.d. 1-sub-Gaussian distribution, if the sample size satisfies $n \gtrsim \delta^{-2}\left(s \log \frac{ed}{s} + \log \frac{2}{\epsilon}\right)$, then the sample covariance matrix $\hat{\boldsymbol{X}} = \frac{1}{n}\sum_{i=1}^n \boldsymbol{x}_i \boldsymbol{x}_i^T$ satisfies $(\delta, s)$-RIP condition with probability $1 - \epsilon$.*

**Lemma 2.** *(Lemma A.3 in [36]) Suppose that $\frac{\boldsymbol{X}}{\sqrt{n}} \in \mathbb{R}^{n \times d}$ satisfies the $(\delta, s+1)$-RIP, if $\boldsymbol{w} \in \mathbb{R}^d$ is a s-sparse vector, then $\left\|\left(\frac{\boldsymbol{X}^T \boldsymbol{X}}{n} - \boldsymbol{I}\right)\boldsymbol{w}\right\|_\infty \leq \sqrt{s}\delta\|\boldsymbol{w}\|_\infty$.*

**Lemma 3.** *(Lemma A.4 in [36]) Suppose that $\frac{1}{\sqrt{n}}\boldsymbol{X} \in \mathbb{R}^{n \times d}$ that satisfies $(\delta, 1)$-RIP with $0 \leq \delta \leq 1$, then we have $\left\|\frac{\boldsymbol{X}^T \boldsymbol{X}}{n}\boldsymbol{w}\right\|_\infty \leq 2d\|\boldsymbol{w}\|_\infty, \forall \boldsymbol{w} \in \mathbb{R}^d$.*

**Lemma 4.** *(Lemma B.5 in [42]) Let $\boldsymbol{\xi} \in \mathcal{R}^n$ is a vector of independent $\sigma$-sub-Gaussian random variables and all $\ell_2$ norm of column vectors of $\boldsymbol{X} \in \mathbb{R}^{n \times d}$ are bounded, then with high probability $1 - \frac{1}{8d^3}$ such that $\left\|\frac{\boldsymbol{X}^T \boldsymbol{\xi}}{n}\right\|_\infty \lesssim \sigma\sqrt{\frac{\log d}{n}}$.*

## A.3 Non-negative Case

We consider the simplified setting where all the elements on support are positive for ground truth $w^\star$. The following lemma shows the recursion of average variable $\overline{u}^t$ on support $\mathcal{S}$ and non-support $\mathcal{S}^c$.

**Lemma 5.** *Consider the sequence $\{\boldsymbol{u}^{t,i}\}$ generated according to (3) and (4) by DGD for solving loss function in (2), the average signal $\overline{\boldsymbol{u}}^t$ on support $\mathcal{S}$ and non-support $\mathcal{S}^c$ are updated according to the following formulas*

$$
\begin{aligned}
\overline{\boldsymbol{u}}_{\mathcal{S}}^{t+1} = \overline{\boldsymbol{u}}_{\mathcal{S}}^t \odot &\left(\mathbf{1}_d - 4\eta\left(\overline{\boldsymbol{u}}_{\mathcal{S}}^t \odot \overline{\boldsymbol{u}}_{\mathcal{S}}^t - \boldsymbol{w}^\star\right) - 4\eta\frac{\boldsymbol{X}^T \boldsymbol{X}}{N}\left(\overline{\boldsymbol{u}}_{\mathcal{S}^c}^t \odot \overline{\boldsymbol{u}}_{\mathcal{S}^c}^t\right) + 4\eta\frac{\boldsymbol{X}^T \boldsymbol{\xi}}{N}\right. \\
&\left. - 4\eta\left(\frac{\boldsymbol{X}\boldsymbol{X}^T}{N} - \boldsymbol{I}\right)\left(\overline{\boldsymbol{u}}_{\mathcal{S}}^t \odot \overline{\boldsymbol{u}}_{\mathcal{S}}^t - \boldsymbol{w}^\star\right) - 4\eta\boldsymbol{p}^t\right) - 4\eta\boldsymbol{q}^t;
\end{aligned}
\tag{10}
$$

$$
\overline{\boldsymbol{u}}_{\mathcal{S}^c}^{t+1} = \overline{\boldsymbol{u}}_{\mathcal{S}^c}^t \odot \left(\mathbf{1}_d - 4\eta\left(\frac{\boldsymbol{X}^T \boldsymbol{X}}{N}\left(\overline{\boldsymbol{u}}_{\mathcal{S}^c}^t \odot \overline{\boldsymbol{u}}_{\mathcal{S}^c}^t\right) - \frac{\boldsymbol{X}^T \boldsymbol{\xi}}{N} + \left(\frac{\boldsymbol{X}\boldsymbol{X}^T}{N} - \boldsymbol{I}\right)\left(\overline{\boldsymbol{u}}_{\mathcal{S}}^t \odot \overline{\boldsymbol{u}}_{\mathcal{S}}^t - \boldsymbol{w}^\star\right)\right)\right)
$$

$$-4\eta\boldsymbol{g}^t) - 4\eta\boldsymbol{f}^t, \tag{11}$$

*where the perturbed error terms* $\boldsymbol{p}^t, \boldsymbol{q}^t, \boldsymbol{g}^t, \boldsymbol{f}^t$ *induced from decentralized network are defined as*

$$\boldsymbol{p}^t = \frac{1}{m}\sum_{i=1}^{m}\left(\frac{\boldsymbol{X}_i^T\boldsymbol{X}_i}{n} - \boldsymbol{I}\right)\left(2\overline{\boldsymbol{u}}_{\mathcal{S}}^t \odot \boldsymbol{\Delta}_{\mathcal{S}}^{t,i} + \boldsymbol{\Delta}_{\mathcal{S}}^{t,i}\odot\boldsymbol{\Delta}_{\mathcal{S}}^{t,i}\right) + 3\boldsymbol{\Delta}_{\mathcal{S}}^{t,i}\odot\boldsymbol{\Delta}_{\mathcal{S}}^{t,i}$$

$$+ \frac{\boldsymbol{X}_i^T\boldsymbol{X}_i}{n}\left(2\overline{\boldsymbol{u}}_{\mathcal{S}^c}^t\odot\boldsymbol{\Delta}_{\mathcal{S}^c}^{t,i} + \boldsymbol{\Delta}_{\mathcal{S}^c}^{t,i}\odot\boldsymbol{\Delta}_{\mathcal{S}^c}^{t,i}\right); \tag{12}$$

$$\boldsymbol{q}^t = \frac{1}{m}\sum_{i=1}^{m}\boldsymbol{\Delta}_{\mathcal{S}}^{t,i}\odot\left(\left(\frac{\boldsymbol{X}_i^T\boldsymbol{X}_i}{n} - \boldsymbol{I}\right)\left(\overline{\boldsymbol{u}}_{\mathcal{S}}^t\odot\overline{\boldsymbol{u}}_{\mathcal{S}}^t - \boldsymbol{w}^\star + 2\overline{\boldsymbol{u}}_{\mathcal{S}}^t\odot\boldsymbol{\Delta}_{\mathcal{S}}^{t,i} + \boldsymbol{\Delta}_{\mathcal{S}}^{t,i}\odot\boldsymbol{\Delta}_{\mathcal{S}}^{t,i}\right)\right.$$

$$\left.+\boldsymbol{\Delta}_{\mathcal{S}}^{t,i}\odot\boldsymbol{\Delta}_{\mathcal{S}}^{t,i} - \frac{\boldsymbol{X}_i^T\boldsymbol{\xi}_i}{n} + \frac{\boldsymbol{X}_i^T\boldsymbol{X}_i}{n}\left(\overline{\boldsymbol{u}}_{\mathcal{S}^c}^t + \boldsymbol{\Delta}_{\mathcal{S}^c}^{t,i}\right)^2\right); \tag{13}$$

$$\boldsymbol{g}^t = \frac{1}{m}\sum_{i=1}^{m}\left(\frac{\boldsymbol{X}_i^T\boldsymbol{X}_i}{n} - \boldsymbol{I}\right)\left(2\overline{\boldsymbol{u}}_{\mathcal{S}}^t\odot\boldsymbol{\Delta}_{\mathcal{S}}^{t,i} + \boldsymbol{\Delta}_{\mathcal{S}}^{t,i}\odot\boldsymbol{\Delta}_{\mathcal{S}}^{t,i}\right) + \frac{\boldsymbol{X}_i^T\boldsymbol{X}_i}{n}\left(2\overline{\boldsymbol{u}}_{\mathcal{S}^c}^t\odot\boldsymbol{\Delta}_{\mathcal{S}^c}^{t,i} + \boldsymbol{\Delta}_{\mathcal{S}^c}^{t,i}\odot\boldsymbol{\Delta}_{\mathcal{S}^c}^{t,i}\right);$$

$$\tag{14}$$

$$\boldsymbol{f}^t = \frac{1}{m}\sum_{i=1}^{m}\boldsymbol{\Delta}_{\mathcal{S}^c}^{t,i}\odot\left(\left(\frac{\boldsymbol{X}_i^T\boldsymbol{X}_i}{n} - \boldsymbol{I}\right)\left(\overline{\boldsymbol{u}}_{\mathcal{S}}^t\odot\overline{\boldsymbol{u}}_{\mathcal{S}}^t - \boldsymbol{w}^\star + 2\overline{\boldsymbol{u}}_{\mathcal{S}}^t\odot\boldsymbol{\Delta}_{\mathcal{S}}^{t,i} + \boldsymbol{\Delta}_{\mathcal{S}}^{t,i}\odot\boldsymbol{\Delta}_{\mathcal{S}}^{t,i}\right)\right.$$

$$\left.+ \frac{\boldsymbol{X}_i^T\boldsymbol{X}_i}{n}\left(\overline{\boldsymbol{u}}_{\mathcal{S}^c}^t + \boldsymbol{\Delta}_{\mathcal{S}^c}^{t,i}\right)^2 - \frac{\boldsymbol{X}_i^T\boldsymbol{\xi}_i}{n}\right). \tag{15}$$

*Proof.* Based on the updating of DGD, one-step iteration of the averaged parameter is

$$\overline{\boldsymbol{u}}^{t+1} = \overline{\boldsymbol{u}}^t - \eta\nabla F(\overline{\boldsymbol{u}}^t) + \frac{\eta}{m}\sum_{i=1}^{m}\left(\nabla f_i(\overline{\boldsymbol{u}}^t) - \nabla f_i(\boldsymbol{u}^{t,i})\right). \tag{16}$$

The gradient difference has the formula as

$$\nabla f_i(\overline{\boldsymbol{u}}^t) - \nabla f_i(\boldsymbol{u}^{t,i}) = \overline{\boldsymbol{u}}^t\odot\left(\frac{4}{n}\boldsymbol{X}_i^T\boldsymbol{X}_i\left(\overline{\boldsymbol{u}}^t\odot\overline{\boldsymbol{u}}^t - \boldsymbol{w}^\star\right) - \frac{4}{n}\boldsymbol{X}_i^T\boldsymbol{\xi}_i\right)$$

$$-\boldsymbol{u}^{t,i}\odot\left(\frac{4}{n}\boldsymbol{X}_i^T\boldsymbol{X}_i\left(\boldsymbol{u}^{t,i}\odot\boldsymbol{u}^{t,i} - \boldsymbol{w}^\star\right) - \frac{4}{n}\boldsymbol{X}_i^T\boldsymbol{\xi}_i\right)$$

$$=\left(\overline{\boldsymbol{u}}^t - \boldsymbol{u}^{t,i}\right)\odot\left(\frac{4}{n}\boldsymbol{X}_i^T\boldsymbol{X}_i\left(\boldsymbol{u}^{t,i}\odot\boldsymbol{u}^{t,i} - \boldsymbol{w}^\star\right) - \frac{4}{n}\boldsymbol{X}_i^T\boldsymbol{\xi}_i\right)$$

$$+\overline{\boldsymbol{u}}^t\odot\left(\frac{4}{n}\boldsymbol{X}_i^T\boldsymbol{X}_i\left(\overline{\boldsymbol{u}}^t\odot\overline{\boldsymbol{u}}^t - \boldsymbol{u}^{t,i}\odot\boldsymbol{u}^{t,i}\right)\right)$$

$$=-\boldsymbol{\Delta}^{t,i}\odot\left(\frac{4}{n}\boldsymbol{X}_i^T\boldsymbol{X}_i\left(\overline{\boldsymbol{u}}^t\odot\overline{\boldsymbol{u}}^t - \boldsymbol{w}^\star + 2\overline{\boldsymbol{u}}^t\odot\boldsymbol{\Delta}^{t,i} + \boldsymbol{\Delta}^{t,i}\odot\boldsymbol{\Delta}^{t,i}\right) - \frac{4}{n}\boldsymbol{X}_i^T\boldsymbol{\xi}_i\right)$$

$$-\overline{\boldsymbol{u}}^t\odot\left(\frac{4}{n}\boldsymbol{X}_i^T\boldsymbol{X}_i\left(2\overline{\boldsymbol{u}}^t\odot\boldsymbol{\Delta}^{t,i} + \boldsymbol{\Delta}^{t,i}\odot\boldsymbol{\Delta}^{t,i}\right)\right), \tag{17}$$

where the last inequality is due to the definition of $\boldsymbol{\Delta}^{t,i}$. Substituting the above equality into (16) would have

$$\overline{\boldsymbol{u}}^{t+1} = \overline{\boldsymbol{u}}^t - 4\eta\overline{\boldsymbol{u}}^t\odot\left(\overline{\boldsymbol{u}}_{\mathcal{S}}^t\odot\overline{\boldsymbol{u}}_{\mathcal{S}}^t - \boldsymbol{w}^\star + \frac{\boldsymbol{X}^T\boldsymbol{X}}{N}\left(\overline{\boldsymbol{u}}_{\mathcal{S}^c}^t\odot\overline{\boldsymbol{u}}_{\mathcal{S}^c}^t\right) - \frac{\boldsymbol{X}^T\boldsymbol{\xi}}{N}\right.$$

$$+\left.\left(\frac{\boldsymbol{X}\boldsymbol{X}^T}{N} - \boldsymbol{I}\right)\left(\overline{\boldsymbol{u}}_{\mathcal{S}}^t\odot\overline{\boldsymbol{u}}_{\mathcal{S}}^t - \boldsymbol{w}^\star\right)\right)$$

$$-4\eta\overline{\boldsymbol{u}}^t\odot\frac{1}{m}\sum_{i=1}^{m}\left(2\left(\frac{\boldsymbol{X}_i^T\boldsymbol{X}_i}{n} - \boldsymbol{I}\right)\left(\overline{\boldsymbol{u}}_{\mathcal{S}}^t\odot\boldsymbol{\Delta}_{\mathcal{S}}^{t,i}\right) + 2\overline{\boldsymbol{u}}_{\mathcal{S}}^t\odot\boldsymbol{\Delta}_{\mathcal{S}}^{t,i}\right.$$

$$+2\frac{\boldsymbol{X}_i^T \boldsymbol{X}_i}{n}\left(\overline{\boldsymbol{u}}_{\mathcal{S}^c}^t \odot \boldsymbol{\Delta}_{\mathcal{S}^c}^{t,i}\right) + \left(\frac{\boldsymbol{X}_i^T \boldsymbol{X}_i}{n} - \boldsymbol{I}\right)\left(\boldsymbol{\Delta}_{\mathcal{S}}^{t,i} \odot \boldsymbol{\Delta}_{\mathcal{S}}^{t,i}\right) + \boldsymbol{\Delta}_{\mathcal{S}}^{t,i} \odot \boldsymbol{\Delta}_{\mathcal{S}}^{t,i}$$

$$+\frac{\boldsymbol{X}_i^T \boldsymbol{X}_i}{n}\left(\boldsymbol{\Delta}_{\mathcal{S}^c}^{t,i} \odot \boldsymbol{\Delta}_{\mathcal{S}^c}^{t,i}\right)\Bigg)$$

$$-4\eta\frac{1}{m}\sum_{i=1}^{m}\boldsymbol{\Delta}^{t,i}\odot\left(\overline{\boldsymbol{u}}_{\mathcal{S}}^t \odot \overline{\boldsymbol{u}}_{\mathcal{S}}^t - \boldsymbol{w}^\star + \left(\frac{\boldsymbol{X}_i^T \boldsymbol{X}_i}{n} - \boldsymbol{I}\right)\left(\overline{\boldsymbol{u}}_{\mathcal{S}}^t \odot \overline{\boldsymbol{u}}_{\mathcal{S}}^t - \boldsymbol{w}^\star\right)\right.$$

$$\left.+\frac{\boldsymbol{X}_i^T \boldsymbol{X}_i}{n}\left(\overline{\boldsymbol{u}}_{\mathcal{S}^c}^t \odot \overline{\boldsymbol{u}}_{\mathcal{S}^c}^t\right) - \frac{\boldsymbol{X}_i^T \boldsymbol{\xi}_i}{n}\right)$$

$$-4\eta\frac{1}{m}\sum_{i=1}^{m}\boldsymbol{\Delta}^{t,i}\odot\left(2\left(\frac{\boldsymbol{X}_i^T \boldsymbol{X}_i}{n} - \boldsymbol{I}\right)\left(\overline{\boldsymbol{u}}_{\mathcal{S}}^t \odot \boldsymbol{\Delta}_{\mathcal{S}}^{t,i}\right) + 2\overline{\boldsymbol{u}}_{\mathcal{S}}^t \odot \boldsymbol{\Delta}_{\mathcal{S}}^{t,i}\right.$$

$$+2\frac{\boldsymbol{X}_i^T \boldsymbol{X}_i}{n}\left(\overline{\boldsymbol{u}}_{\mathcal{S}^c}^t \odot \boldsymbol{\Delta}_{\mathcal{S}^c}^{t,i}\right) + \left(\frac{\boldsymbol{X}_i^T \boldsymbol{X}_i}{n} - \boldsymbol{I}\right)\left(\boldsymbol{\Delta}_{\mathcal{S}}^{t,i} \odot \boldsymbol{\Delta}_{\mathcal{S}}^{t,i}\right) + \boldsymbol{\Delta}_{\mathcal{S}}^{t,i} \odot \boldsymbol{\Delta}_{\mathcal{S}}^{t,i}$$

$$\left.+\frac{\boldsymbol{X}_i^T \boldsymbol{X}_i}{n}\left(\boldsymbol{\Delta}_{\mathcal{S}^c}^{t,i} \odot \boldsymbol{\Delta}_{\mathcal{S}^c}^{t,i}\right)\right) \tag{18}$$

Because there are $\frac{1}{m}\sum_{i=1}^{m}\overline{\boldsymbol{u}}_{\mathcal{S}}^t \odot \boldsymbol{\Delta}_{\mathcal{S}}^{t,i} = \boldsymbol{0}$ and $\frac{1}{m}\sum_{i=1}^{m}\boldsymbol{\Delta}_{\mathcal{S}}^{t,i} \odot \left(\overline{\boldsymbol{u}}_{\mathcal{S}}^t \odot \overline{\boldsymbol{u}}_{\mathcal{S}}^t - \boldsymbol{w}^\star\right) = \boldsymbol{0}$, the above formula can be simplified as

$$\overline{\boldsymbol{u}}^{t+1} = \overline{\boldsymbol{u}}^t - 4\eta\overline{\boldsymbol{u}}^t \odot\left(\overline{\boldsymbol{u}}_{\mathcal{S}}^t \odot \overline{\boldsymbol{u}}_{\mathcal{S}}^t - \boldsymbol{w}^\star + \frac{\boldsymbol{X}^T \boldsymbol{X}}{N}\left(\overline{\boldsymbol{u}}_{\mathcal{S}^c}^t \odot \overline{\boldsymbol{u}}_{\mathcal{S}^c}^t\right) - \frac{\boldsymbol{X}^T \boldsymbol{\xi}}{N}\right.$$

$$\left.+\left(\frac{\boldsymbol{X}\boldsymbol{X}^T}{N} - \boldsymbol{I}\right)\left(\overline{\boldsymbol{u}}_{\mathcal{S}}^t \odot \overline{\boldsymbol{u}}_{\mathcal{S}}^t - \boldsymbol{w}^\star\right)\right)$$

$$-4\eta\overline{\boldsymbol{u}}^t \odot \frac{1}{m}\sum_{i=1}^{m}\left(2\left(\frac{\boldsymbol{X}_i^T \boldsymbol{X}_i}{n} - \boldsymbol{I}\right)\left(\overline{\boldsymbol{u}}_{\mathcal{S}}^t \odot \boldsymbol{\Delta}_{\mathcal{S}}^{t,i}\right)\right.$$

$$+2\frac{\boldsymbol{X}_i^T \boldsymbol{X}_i}{n}\left(\overline{\boldsymbol{u}}_{\mathcal{S}^c}^t \odot \boldsymbol{\Delta}_{\mathcal{S}^c}^{t,i}\right) + \left(\frac{\boldsymbol{X}_i^T \boldsymbol{X}_i}{n} - \boldsymbol{I}\right)\left(\boldsymbol{\Delta}_{\mathcal{S}}^{t,i} \odot \boldsymbol{\Delta}_{\mathcal{S}}^{t,i}\right) + \boldsymbol{\Delta}_{\mathcal{S}}^{t,i} \odot \boldsymbol{\Delta}_{\mathcal{S}}^{t,i}$$

$$\left.+\frac{\boldsymbol{X}_i^T \boldsymbol{X}_i}{n}\left(\boldsymbol{\Delta}_{\mathcal{S}^c}^{t,i} \odot \boldsymbol{\Delta}_{\mathcal{S}^c}^{t,i}\right)\right)$$

$$-4\eta\frac{1}{m}\sum_{i=1}^{m}\boldsymbol{\Delta}^{t,i}\odot\left(\left(\frac{\boldsymbol{X}_i^T \boldsymbol{X}_i}{n} - \boldsymbol{I}\right)\left(\overline{\boldsymbol{u}}_{\mathcal{S}}^t \odot \overline{\boldsymbol{u}}_{\mathcal{S}}^t - \boldsymbol{w}^\star\right)\right.$$

$$\left.+\frac{\boldsymbol{X}_i^T \boldsymbol{X}_i}{n}\left(\overline{\boldsymbol{u}}_{\mathcal{S}^c}^t \odot \overline{\boldsymbol{u}}_{\mathcal{S}^c}^t\right) - \frac{\boldsymbol{X}_i^T \boldsymbol{\xi}_i}{n}\right)$$

$$-4\eta\frac{1}{m}\sum_{i=1}^{m}\boldsymbol{\Delta}^{t,i}\odot\left(2\left(\frac{\boldsymbol{X}_i^T \boldsymbol{X}_i}{n} - \boldsymbol{I}\right)\left(\overline{\boldsymbol{u}}_{\mathcal{S}}^t \odot \boldsymbol{\Delta}_{\mathcal{S}}^{t,i}\right) + 2\overline{\boldsymbol{u}}_{\mathcal{S}}^t \odot \boldsymbol{\Delta}_{\mathcal{S}}^{t,i}\right.$$

$$+2\frac{\boldsymbol{X}_i^T \boldsymbol{X}_i}{n}\left(\overline{\boldsymbol{u}}_{\mathcal{S}^c}^t \odot \boldsymbol{\Delta}_{\mathcal{S}^c}^{t,i}\right) + \left(\frac{\boldsymbol{X}_i^T \boldsymbol{X}_i}{n} - \boldsymbol{I}\right)\left(\boldsymbol{\Delta}_{\mathcal{S}}^{t,i} \odot \boldsymbol{\Delta}_{\mathcal{S}}^{t,i}\right) + \boldsymbol{\Delta}_{\mathcal{S}}^{t,i} \odot \boldsymbol{\Delta}_{\mathcal{S}}^{t,i}$$

$$\left.+\frac{\boldsymbol{X}_i^T \boldsymbol{X}_i}{n}\left(\boldsymbol{\Delta}_{\mathcal{S}^c}^{t,i} \odot \boldsymbol{\Delta}_{\mathcal{S}^c}^{t,i}\right)\right). \tag{19}$$

Thus, this would obtain the recursion (10) for support averaged signal.

The recursion of optimization error on non-support $\mathcal{S}^c$ becomes

$$
\overline{\boldsymbol{u}}_{\mathcal{S}^c}^{t+1} = \overline{\boldsymbol{u}}_{\mathcal{S}^c}^t - 4\eta\overline{\boldsymbol{u}}_{\mathcal{S}^c}^t \odot \left( \frac{\boldsymbol{X}^T\boldsymbol{X}}{N} \left( \overline{\boldsymbol{u}}_{\mathcal{S}^c}^t \odot \overline{\boldsymbol{u}}_{\mathcal{S}^c}^t \right) - \frac{\boldsymbol{X}^T\boldsymbol{\xi}}{N} + \left( \frac{\boldsymbol{X}\boldsymbol{X}^T}{N} - \boldsymbol{I} \right) \left( \overline{\boldsymbol{u}}_{\mathcal{S}}^t \odot \overline{\boldsymbol{u}}_{\mathcal{S}}^t - \boldsymbol{w}^\star \right) \right)
$$

$$
- 4\eta\overline{\boldsymbol{u}}_{\mathcal{S}^c}^t \odot \frac{1}{m}\sum_{i=1}^m \left( 2\left( \frac{\boldsymbol{X}_i^T\boldsymbol{X}_i}{n} - \boldsymbol{I} \right)\left( \overline{\boldsymbol{u}}_{\mathcal{S}}^t \odot \boldsymbol{\Delta}_{\mathcal{S}}^{t,i} \right) \right.
$$

$$
+ 2\frac{\boldsymbol{X}_i^T\boldsymbol{X}_i}{n}\left( \overline{\boldsymbol{u}}_{\mathcal{S}^c}^t \odot \boldsymbol{\Delta}_{\mathcal{S}^c}^{t,i} \right) + \left( \frac{\boldsymbol{X}_i^T\boldsymbol{X}_i}{n} - \boldsymbol{I} \right)\left( \boldsymbol{\Delta}_{\mathcal{S}}^{t,i} \odot \boldsymbol{\Delta}_{\mathcal{S}}^{t,i} \right)
$$

$$
\left. + \frac{\boldsymbol{X}_i^T\boldsymbol{X}_i}{n}\left( \boldsymbol{\Delta}_{\mathcal{S}^c}^{t,i} \odot \boldsymbol{\Delta}_{\mathcal{S}^c}^{t,i} \right) \right)
$$

$$
- 4\eta\frac{1}{m}\sum_{i=1}^m \boldsymbol{\Delta}_{\mathcal{S}^c}^{t,i} \odot \left( \left( \frac{\boldsymbol{X}_i^T\boldsymbol{X}_i}{n} - \boldsymbol{I} \right)\left( \overline{\boldsymbol{u}}_{\mathcal{S}}^t \odot \overline{\boldsymbol{u}}_{\mathcal{S}}^t - \boldsymbol{w}^\star \right) \right.
$$

$$
\left. + \frac{\boldsymbol{X}_i^T\boldsymbol{X}_i}{n}\left( \overline{\boldsymbol{u}}_{\mathcal{S}^c}^t \odot \overline{\boldsymbol{u}}_{\mathcal{S}^c}^t \right) - \frac{\boldsymbol{X}_i^T\boldsymbol{\xi}_i}{n} \right)
$$

$$
- 4\eta\frac{1}{m}\sum_{i=1}^m \boldsymbol{\Delta}_{\mathcal{S}^c}^{t,i} \odot \left( 2\left( \frac{\boldsymbol{X}_i^T\boldsymbol{X}_i}{n} - \boldsymbol{I} \right)\left( \overline{\boldsymbol{u}}_{\mathcal{S}}^t \odot \boldsymbol{\Delta}_{\mathcal{S}}^{t,i} \right) \right.
$$

$$
+ 2\frac{\boldsymbol{X}_i^T\boldsymbol{X}_i}{n}\left( \overline{\boldsymbol{u}}_{\mathcal{S}^c}^t \odot \boldsymbol{\Delta}_{\mathcal{S}^c}^{t,i} \right) + \left( \frac{\boldsymbol{X}_i^T\boldsymbol{X}_i}{n} - \boldsymbol{I} \right)\left( \boldsymbol{\Delta}_{\mathcal{S}}^{t,i} \odot \boldsymbol{\Delta}_{\mathcal{S}}^{t,i} \right)
$$

$$
\left. + \frac{\boldsymbol{X}_i^T\boldsymbol{X}_i}{n}\left( \boldsymbol{\Delta}_{\mathcal{S}^c}^{t,i} \odot \boldsymbol{\Delta}_{\mathcal{S}^c}^{t,i} \right) \right). \tag{20}
$$

Rearranging the above equality would obtain the (11). □

The following lemma shows the recursion of consensus error on support $\mathcal{S}$ and non-support $\mathcal{S}^c$.

**Lemma 6.** *Consider the sequence $\{\boldsymbol{u}^{t,i}\}$ generated according to (3) and (4) by DGD for solving loss function in (2), the consensus error $\boldsymbol{\Delta}^t$ on support $\mathcal{S}$ and non-support $\mathcal{S}^c$ have following recursion*

$$
\left\| \boldsymbol{\Delta}_{\mathcal{S}}^{t+1} \right\|_\infty \leq \rho\left\| \boldsymbol{\Delta}_{\mathcal{S}}^t \right\|_\infty \left( 1 + 4\eta\left( \left( \sqrt{s}\delta_{\max} + 1 \right)\left\| \overline{\boldsymbol{u}}_{\mathcal{S}}^t \odot \overline{\boldsymbol{u}}_{\mathcal{S}}^t - \boldsymbol{w}^\star \right\|_\infty + 2d\left( \overline{\boldsymbol{u}}_{\mathcal{S}^c}^t + \boldsymbol{\Delta}_{\mathcal{S}^c}^t \right)^2 \right.\right.
$$

$$
+ 2\left( \sqrt{s}\delta_{\max} + 1 \right)\left( \left\| \overline{\boldsymbol{u}}_{\mathcal{S}}^t \right\|_\infty + \left\| \boldsymbol{\Delta}_{\mathcal{S}}^t \right\|_\infty \right)^2 + \max_i\left\| \frac{\boldsymbol{X}_i^T\boldsymbol{\xi}_i}{n} \right\|_\infty \bigg)\bigg)
$$

$$
+ 4\rho\eta\left\| \overline{\boldsymbol{u}}_{\mathcal{S}}^t \right\|_\infty \cdot \left( \sqrt{s}\left( \delta_{\max} + \delta \right)\left\| \overline{\boldsymbol{u}}_{\mathcal{S}}^t \odot \overline{\boldsymbol{u}}_{\mathcal{S}}^t - \boldsymbol{w}^\star \right\|_\infty + \max_i\left\| \frac{\boldsymbol{X}_i^T\boldsymbol{\xi}_i}{n} \right\|_\infty \right.
$$

$$
+ \left\| \frac{\boldsymbol{X}^T\boldsymbol{\xi}}{N} \right\|_\infty + 2d\left( \overline{\boldsymbol{u}}_{\mathcal{S}^c}^t + \boldsymbol{\Delta}_{\mathcal{S}^c}^t \right)^2 \bigg); \tag{21}
$$

$$
\left\| \boldsymbol{\Delta}_{\mathcal{S}^c}^{t+1} \right\|_\infty \leq \rho\left\| \boldsymbol{\Delta}_{\mathcal{S}^c}^t \right\|_\infty \left( 1 + 4\eta\left( \sqrt{s}\delta_{\max}\left\| \overline{\boldsymbol{u}}_{\mathcal{S}}^t \odot \overline{\boldsymbol{u}}_{\mathcal{S}}^t - \boldsymbol{w}^\star \right\|_\infty + \sqrt{s}\delta_{\max}\left( \left\| \overline{\boldsymbol{u}}_{\mathcal{S}}^t \right\|_\infty + \left\| \boldsymbol{\Delta}_{\mathcal{S}}^t \right\|_\infty \right)^2 \right.\right.
$$

$$
+ 6d\left( \left\| \overline{\boldsymbol{u}}_{\mathcal{S}^c}^t \right\|_\infty + \left\| \boldsymbol{\Delta}_{\mathcal{S}^c}^t \right\|_\infty \right)^2 + \max_i\left\| \frac{\boldsymbol{X}_i^T\boldsymbol{\xi}_i}{n} \right\|_\infty \bigg)\bigg)
$$

$$
+ 4\rho\eta\left\| \overline{\boldsymbol{u}}_{\mathcal{S}^c}^t \right\|_\infty \cdot \left( \sqrt{s}\left( \delta_{\max} + \delta \right)\left\| \overline{\boldsymbol{u}}_{\mathcal{S}}^t \odot \overline{\boldsymbol{u}}_{\mathcal{S}}^t - \boldsymbol{w}^\star \right\|_\infty + \sqrt{s}\delta_{\max}\left\| \boldsymbol{\Delta}_{\mathcal{S}}^t \right\|_\infty \right.
$$

$$
\cdot \left( \left\| \overline{\boldsymbol{u}}_{\mathcal{S}}^t \right\|_\infty + \left\| \boldsymbol{\Delta}_{\mathcal{S}}^t \right\|_\infty \right) + 4d\left\| \overline{\boldsymbol{u}}_{\mathcal{S}^c}^t \right\|_\infty^2 + \max_i\left\| \frac{\boldsymbol{X}_i^T\boldsymbol{\xi}_i}{n} \right\|_\infty + \left\| \frac{\boldsymbol{X}^T\boldsymbol{\xi}}{N} \right\|_\infty \bigg). \tag{22}
$$

*Proof.* Based on the iteration of DGD, there is

$$\boldsymbol{U}^{t+1}\left(\boldsymbol{I}_m - \frac{1}{m}\mathbf{1}_m\mathbf{1}_m^T\right) = \left(\boldsymbol{U}^t - \eta\nabla\boldsymbol{f}(\boldsymbol{U}^t)\right)\boldsymbol{W}\left(\boldsymbol{I}_m - \frac{1}{m}\mathbf{1}_m\mathbf{1}_m^T\right)$$

$$= \left(\boldsymbol{U}^t - \eta\nabla\boldsymbol{f}(\boldsymbol{U}^t)\right)\left(\boldsymbol{W} - \frac{1}{m}\mathbf{1}_m\mathbf{1}_m^T\right)$$

$$= \left(\boldsymbol{U}^t - \overline{\boldsymbol{U}}^t - \eta\nabla\boldsymbol{f}(\boldsymbol{U}^t) + \eta\nabla\boldsymbol{F}(\overline{\boldsymbol{U}}^t)\right)\left(\boldsymbol{W} - \frac{1}{m}\mathbf{1}_m\mathbf{1}_m^T\right). \qquad (23)$$

Thus, the consensus error on support $\mathcal{S}$ has recursion as follows

$$\left\|\boldsymbol{\Delta}_{\mathcal{S}}^{t+1}\right\|_\infty \overset{(i)}{\le} \rho\left\|\boldsymbol{\Delta}_{\mathcal{S}}^t\right\|_\infty + \rho\eta\max_i\left\|\mathbf{1}_{\mathcal{S}}\odot\left(\nabla f_i(\boldsymbol{u}^{t,i}) - \nabla F(\overline{\boldsymbol{u}}^t)\right)\right\|_\infty$$

$$\le \rho\left\|\boldsymbol{\Delta}_{\mathcal{S}}^t\right\|_\infty + \rho\eta\max_i\left\|\mathbf{1}_{\mathcal{S}}\odot\left(\nabla f_i(\boldsymbol{u}^{t,i}) - \nabla f_i(\overline{\boldsymbol{u}}^t)\right)\right\|_\infty$$

$$+ \rho\eta\max_i\left\|\mathbf{1}_{\mathcal{S}}\odot\left(\nabla f_i(\overline{\boldsymbol{u}}^t) - \nabla F(\overline{\boldsymbol{u}}^t)\right)\right\|_\infty$$

$$\overset{(ii)}{\le} \rho\left\|\boldsymbol{\Delta}_{\mathcal{S}}^t\right\|_\infty + 4\rho\eta\left\|\boldsymbol{\Delta}_{\mathcal{S}}^t\right\|_\infty\cdot\left(\left(\sqrt{s}\delta_{\max}+1\right)\left\|\overline{\boldsymbol{u}}_{\mathcal{S}}^t\odot\overline{\boldsymbol{u}}_{\mathcal{S}}^t - \boldsymbol{w}^\star\right\|_\infty +\right.$$

$$+2d\left\|\overline{\boldsymbol{u}}_{\mathcal{S}^c}^t\right\|_\infty^2 + \max_i\left\|\frac{\boldsymbol{X}_i^T\boldsymbol{\xi}_i}{n}\right\|_\infty + 2\left(\sqrt{s}\delta_{\max}+1\right)\left\|\overline{\boldsymbol{u}}_{\mathcal{S}}^t\right\|_\infty\left\|\boldsymbol{\Delta}_{\mathcal{S}}^t\right\|_\infty$$

$$\left.+4d\left\|\overline{\boldsymbol{u}}_{\mathcal{S}^c}^t\right\|_\infty\left\|\boldsymbol{\Delta}_{\mathcal{S}^c}^t\right\|_\infty + \left(\sqrt{s}\delta_{\max}+1\right)\left\|\boldsymbol{\Delta}_{\mathcal{S}}^t\right\|_\infty^2 + 2d\left\|\boldsymbol{\Delta}_{\mathcal{S}^c}^t\right\|_\infty^2\right)$$

$$+ 4\rho\eta\left\|\overline{\boldsymbol{u}}_{\mathcal{S}}^t\right\|_\infty\cdot\left(2\left(\sqrt{s}\delta_{\max}+1\right)\left\|\overline{\boldsymbol{u}}_{\mathcal{S}}^t\right\|_\infty\left\|\boldsymbol{\Delta}_{\mathcal{S}}^t\right\|_\infty + 4d\left\|\overline{\boldsymbol{u}}_{\mathcal{S}^c}^t\right\|_\infty\left\|\boldsymbol{\Delta}_{\mathcal{S}^c}^t\right\|_\infty\right.$$

$$+ \left(\sqrt{s}\delta_{\max}+1\right)\left\|\boldsymbol{\Delta}_{\mathcal{S}}^t\right\|_\infty^2 + 2d\left\|\boldsymbol{\Delta}_{\mathcal{S}}^t\right\|_\infty^2 + \left(\sqrt{s}\left(\delta_{\max}+\delta\right)+2\right)\left\|\overline{\boldsymbol{u}}_{\mathcal{S}}^t\odot\overline{\boldsymbol{u}}_{\mathcal{S}}^t - \boldsymbol{w}^\star\right\|_\infty$$

$$\left.+ 2d\left\|\overline{\boldsymbol{u}}_{\mathcal{S}^c}^t\right\|_\infty^2 + \max_i\left\|\frac{\boldsymbol{X}_i^T\boldsymbol{\xi}_i}{n}\right\|_\infty + \left\|\frac{\boldsymbol{X}^T\boldsymbol{\xi}}{N}\right\|_\infty\right)$$

$$\overset{(iii)}{\le} \rho\left\|\boldsymbol{\Delta}_{\mathcal{S}}^t\right\|_\infty\left(1 + 4\eta\left(\left(\sqrt{s}\delta_{\max}+1\right)\left\|\overline{\boldsymbol{u}}_{\mathcal{S}}^t\odot\overline{\boldsymbol{u}}_{\mathcal{S}}^t - \boldsymbol{w}^\star\right\|_\infty + 2d\left(\overline{\boldsymbol{u}}_{\mathcal{S}^c}^t + \boldsymbol{\Delta}_{\mathcal{S}^c}^t\right)^2\right.\right.$$

$$\left.\left.+ 2\left(\sqrt{s}\delta_{\max}+1\right)\left(\left\|\overline{\boldsymbol{u}}_{\mathcal{S}}^t\right\|_\infty + \left\|\boldsymbol{\Delta}_{\mathcal{S}}^t\right\|_\infty\right)^2 + \max_i\left\|\frac{\boldsymbol{X}_i^T\boldsymbol{\xi}_i}{n}\right\|_\infty\right)\right)$$

$$+ 4\rho\eta\left\|\overline{\boldsymbol{u}}_{\mathcal{S}}^t\right\|_\infty\cdot\left(\sqrt{s}\left(\delta_{\max}+\delta\right)\left\|\overline{\boldsymbol{u}}_{\mathcal{S}}^t\odot\overline{\boldsymbol{u}}_{\mathcal{S}}^t - \boldsymbol{w}^\star\right\|_\infty + \max_i\left\|\frac{\boldsymbol{X}_i^T\boldsymbol{\xi}_i}{n}\right\|_\infty\right.$$

$$\left.+ \left\|\frac{\boldsymbol{X}^T\boldsymbol{\xi}}{N}\right\|_\infty + 2d\left(\overline{\boldsymbol{u}}_{\mathcal{S}^c}^t + \boldsymbol{\Delta}_{\mathcal{S}^c}^t\right)^2\right), \qquad (24)$$

where $(i)$ is due to the defined spectral gap of network in Assumption 1 and $(ii)$ uses the gradient difference formula in (17) and

$$\nabla f_i(\overline{\boldsymbol{u}}^t) - \nabla F(\overline{\boldsymbol{u}}^t) = \overline{\boldsymbol{u}}^t\odot\left(\frac{4\boldsymbol{X}_i^T\boldsymbol{X}_i}{n}\left(\overline{\boldsymbol{u}}^t\odot\overline{\boldsymbol{u}}^t - \boldsymbol{w}^\star\right) - \frac{4\boldsymbol{X}_i^T\boldsymbol{\xi}_i}{n}\right)$$

$$- \overline{\boldsymbol{u}}^t\odot\left(\frac{4\boldsymbol{X}^T\boldsymbol{X}}{N}\left(\overline{\boldsymbol{u}}^t\odot\overline{\boldsymbol{u}}^t - \boldsymbol{w}^\star\right) - \frac{4\boldsymbol{X}^T\boldsymbol{\xi}}{N}\right) \qquad (25)$$

$$= \overline{\boldsymbol{u}}^t\odot\left(4\left(\left(\frac{\boldsymbol{X}_i^T\boldsymbol{X}_i}{n} - \boldsymbol{I}\right) - \left(\frac{\boldsymbol{X}^T\boldsymbol{X}}{N} - \boldsymbol{I}\right)\right)\left(\overline{\boldsymbol{u}}^t\odot\overline{\boldsymbol{u}}^t - \boldsymbol{w}^\star\right)\right.$$

$$\left.+ \left(\frac{4\boldsymbol{X}^T\boldsymbol{\xi}}{N} - \frac{4\boldsymbol{X}_i^T\boldsymbol{\xi}_i}{n}\right)\right)$$

and local and global RIP conditions. The $(iii)$ is summing up terms involved $\left\|\boldsymbol{\Delta}_{\mathcal{S}}^{t+1}\right\|_\infty$ and $\left\|\overline{\boldsymbol{u}}_{\mathcal{S}}^t\right\|_\infty$ separately.

The recursion of consensus error on non-support $\mathcal{S}^c$ part is as follows

$$
\begin{aligned}
\left\|\boldsymbol{\Delta}_{\mathcal{S}^c}^{t+1}\right\|_\infty &\le \rho\left\|\boldsymbol{\Delta}_{\mathcal{S}^c}^t\right\|_\infty + \rho\eta\max_i\left\|\mathbf{1}_{\mathcal{S}^c}\odot\left(\nabla f_i(\boldsymbol{u}_i^t)-\nabla F(\overline{\boldsymbol{u}}^t)\right)\right\|_\infty \\
&\le \rho\left\|\boldsymbol{\Delta}_{\mathcal{S}^c}^t\right\|_\infty + \rho\eta\max_i\left\|\mathbf{1}_{\mathcal{S}^c}\odot\left(\nabla f_i(\boldsymbol{u}_i^t)-\nabla f_i(\overline{\boldsymbol{u}}^t)\right)\right\|_\infty \\
&\quad + \rho\eta\max_i\left\|\mathbf{1}_{\mathcal{S}^c}\odot\left(\nabla f_i(\overline{\boldsymbol{u}}^t)-\nabla F(\overline{\boldsymbol{u}}^t)\right)\right\|_\infty \\
&\le \rho\left\|\boldsymbol{\Delta}_{\mathcal{S}^c}^t\right\|_\infty + 4\rho\eta\left\|\boldsymbol{\Delta}_{\mathcal{S}^c}^t\right\|_\infty\cdot\left(\sqrt{s}\delta_{\max}\left\|\overline{\boldsymbol{u}}_{\mathcal{S}}^t\odot\overline{\boldsymbol{u}}_{\mathcal{S}}^t-\boldsymbol{w}^\star\right\|_\infty+\right. \\
&\quad +2d\left\|\overline{\boldsymbol{u}}_{\mathcal{S}^c}^t\right\|_\infty^2+\max_i\left\|\frac{\boldsymbol{X}_i^T\boldsymbol{\xi}_i}{n}\right\|_\infty+2\sqrt{s}\delta_{\max}\left\|\overline{\boldsymbol{u}}_{\mathcal{S}}^t\right\|_\infty\left\|\boldsymbol{\Delta}_{\mathcal{S}}^t\right\|_\infty \\
&\quad +4d\left\|\overline{\boldsymbol{u}}_{\mathcal{S}^c}^t\right\|_\infty\left\|\boldsymbol{\Delta}_{\mathcal{S}^c}^t\right\|_\infty+\sqrt{s}\delta_{\max}\left\|\boldsymbol{\Delta}_{\mathcal{S}}^t\right\|_\infty^2+2d\left\|\boldsymbol{\Delta}_{\mathcal{S}^c}^t\right\|_\infty^2\Big) \\
&\quad + 4\rho\eta\left\|\overline{\boldsymbol{u}}_{\mathcal{S}^c}^t\right\|_\infty\cdot\left(2\sqrt{s}\delta_{\max}\left\|\overline{\boldsymbol{u}}_{\mathcal{S}}^t\right\|_\infty\left\|\boldsymbol{\Delta}_{\mathcal{S}}^t\right\|_\infty+4d\left\|\overline{\boldsymbol{u}}_{\mathcal{S}^c}^t\right\|_\infty\left\|\boldsymbol{\Delta}_{\mathcal{S}^c}^t\right\|_\infty\right. \\
&\quad +\sqrt{s}\delta_{\max}\left\|\boldsymbol{\Delta}_{\mathcal{S}}^t\right\|_\infty^2+2d\left\|\boldsymbol{\Delta}_{\mathcal{S}^c}^t\right\|_\infty^2+\sqrt{s}\left(\delta_{\max}+\delta\right)\left\|\overline{\boldsymbol{u}}_{\mathcal{S}}^t\odot\overline{\boldsymbol{u}}_{\mathcal{S}}^t-\boldsymbol{w}^\star\right\|_\infty \\
&\quad +4d\left\|\overline{\boldsymbol{u}}_{\mathcal{S}^c}^t\right\|_\infty^2+\max_i\left\|\frac{\boldsymbol{X}_i^T\boldsymbol{\xi}_i}{n}\right\|_\infty+\left\|\frac{\boldsymbol{X}^T\boldsymbol{\xi}}{N}\right\|_\infty\Big) \\
&\le \rho\left\|\boldsymbol{\Delta}_{\mathcal{S}^c}^t\right\|_\infty\left(1+4\eta\left(\sqrt{s}\delta_{\max}\left\|\overline{\boldsymbol{u}}_{\mathcal{S}}^t\odot\overline{\boldsymbol{u}}_{\mathcal{S}}^t-\boldsymbol{w}^\star\right\|_\infty+\sqrt{s}\delta_{\max}\left(\left\|\overline{\boldsymbol{u}}_{\mathcal{S}}^t\right\|_\infty+\left\|\boldsymbol{\Delta}_{\mathcal{S}}^t\right\|_\infty\right)^2\right.\right. \\
&\quad +6d\left(\left\|\overline{\boldsymbol{u}}_{\mathcal{S}^c}^t\right\|_\infty+\left\|\boldsymbol{\Delta}_{\mathcal{S}^c}^t\right\|_\infty\right)^2+\max_i\left\|\frac{\boldsymbol{X}_i^T\boldsymbol{\xi}_i}{n}\right\|_\infty\bigg)\bigg) \\
&\quad +4\rho\eta\left\|\overline{\boldsymbol{u}}_{\mathcal{S}^c}^t\right\|_\infty\cdot\left(\sqrt{s}\left(\delta_{\max}+\delta\right)\left\|\overline{\boldsymbol{u}}_{\mathcal{S}}^t\odot\overline{\boldsymbol{u}}_{\mathcal{S}}^t-\boldsymbol{w}^\star\right\|_\infty+\sqrt{s}\delta_{\max}\left\|\boldsymbol{\Delta}_{\mathcal{S}}^t\right\|_\infty\right. \\
&\quad \cdot\left(\left\|\overline{\boldsymbol{u}}_{\mathcal{S}}^t\right\|_\infty+\left\|\boldsymbol{\Delta}_{\mathcal{S}}^t\right\|_\infty\right)+4d\left\|\overline{\boldsymbol{u}}_{\mathcal{S}^c}^t\right\|_\infty^2+\max_i\left\|\frac{\boldsymbol{X}_i^T\boldsymbol{\xi}_i}{n}\right\|_\infty+\left\|\frac{\boldsymbol{X}^T\boldsymbol{\xi}}{N}\right\|_\infty\bigg). \quad (27)
\end{aligned}
$$

$\square$

The following proposition shows the dynamics of average variable $\overline{\boldsymbol{u}}^t$ and consensus error $\boldsymbol{\Delta}^{t,i}$ in the form of an inductive hypothesis. Before showing the proposition, we define the following quantities. We define $T:=\frac{1}{\eta w_{\max}^\star}\log\frac{1}{\alpha^4}$ and for any integer $k\ge -1$, $T_k:=2^kT$ and $\overline{T}_k:=\sum_{i=0}^k T_i$ with $\overline{T}_{-1}=0$ where $T_k$ denotes the number of iterations between $(k-1)$-th and $k$-th induction step. Defining $K:=\left\lceil\log_2\frac{w_{\max}^\star}{\zeta}\right\rceil$ as the number of induction steps, $B_k:=\frac{w_{\max}^\star}{40\times 2^k}$ denotes the upper bound of perturbed error in $(k-1)$-th induction step and constant scale parameter $\beta:=\frac{32}{1-\sqrt{\rho}}$.

**Proposition 2.** *With the same setting as Theorem 1, the following claims hold $k=0,1,\cdots,K-1$ steps.*

- *(a) For $\overline{T}_{k-1}\le t<\overline{T}_k$ that $\forall k\in[K]$, there is $\left\|\overline{\boldsymbol{u}}_{\mathcal{S}}^t\odot\overline{\boldsymbol{u}}_{\mathcal{S}}^t-\boldsymbol{w}^\star\right\|_\infty\le\frac{w_{\max}^\star}{2^k}$.*

- *(b) For $\forall k\in[K]$, there is $\left\|\overline{\boldsymbol{u}}_{\mathcal{S}}^{\overline{T}_{k-1}}\odot\overline{\boldsymbol{u}}_{\mathcal{S}}^{\overline{T}_{k-1}}-\boldsymbol{w}^\star\right\|_\infty\le\frac{w_{\max}^\star}{2^k}$.*

- *(c) For $\overline{T}_{k-1}\le t<\overline{T}_k$ that $\forall k\in[K]$, it has $\left\|\boldsymbol{\Delta}_{\mathcal{S}}^t\right\|_\infty\le 4\beta\rho^{\frac{3}{4}}\eta\left\|\overline{\boldsymbol{u}}_{\mathcal{S}}^t\right\|_\infty B_k$. In addition, the refined element-wise bound is $|\boldsymbol{\Delta}_j^t|\le 4\beta\rho^{\frac{3}{4}}\eta|\overline{u}_j^t|B_k,\forall j\in\mathcal{S}$.*

- *(d) For $\overline{T}_{k-1} \leq t < \overline{T}_k$ that $\forall k \in [K]$, it has $\left\|\boldsymbol{\Delta}_{\mathcal{S}^c}^t\right\|_\infty \leq 4\beta\rho^{\frac{3}{4}}\eta\left\|\overline{\boldsymbol{u}}_{\mathcal{S}^c}^t\right\|_\infty B_k$. In addition, the refined element-wise bound is $|\boldsymbol{\Delta}_j^t| \leq 4\beta\rho^{\frac{3}{4}}\eta|\overline{u}_j^t|B_k, \forall j \in \mathcal{S}^c$.*

- *(e) For $\forall k \in [K]$ and $\forall j \in \mathcal{S}$, $\alpha^3 \leq \overline{u}_j^{\overline{T}_{k-1}} \leq w_j^\star + 4B_k$.*

*Proof.* **Proof idea:** Inductions (a), (b), (e) indicate that if the connectivity of the network is sufficiently well ($\rho$ is small enough), the trajectory of the averaged signal $\overline{\boldsymbol{u}}^t$ would mimic that of the centralized case [36]. Different from the centralized setting, these three claims are based on inductions (c) and (d), which guarantee that the consensus error along both support $\mathcal{S}$ and non-support $\mathcal{S}^c$ can be controlled based on the magnitude of the respective signals. We utilize this property to reparameterize consensus error by Hadamard product based on the averaged signal. Thus, the perturbed error terms induced by the decentralized network in the recursion of the averaged signal can be quantitatively through the reparameterized consensus error. Then conditions on network connectivity $\rho$ and step size $\eta$ can guarantee that the averaged signal in decentralized would have properties in inductions (a), (b), (e) based on inductions (c), (d).

*Base case*: As the initialization $\boldsymbol{u}^{0,i} = \alpha\boldsymbol{1}_d, \forall i \in [m]$. Due to the condition on $\alpha$, the base case is true.

*Induction Step*: If the above (a)-(e) induction hypotheses hold all until some $0 \leq k \leq K - 1$, we should prove they still hold at $k+1$-th induction step.

(a) The magnitude of $\boldsymbol{p}^t$ in (12) under this induction step can be bounded based on inductions (c), (d). $\forall \overline{T}_{k-1} \leq t < t+1 < \overline{T}_k$, if $\left\|\overline{\boldsymbol{u}}_{\mathcal{S}^c}^t\right\|_\infty$ keep same order as initialization, then there is

$$\left\|\boldsymbol{p}_{\mathcal{S}}^t\right\|_\infty \leq 8\sqrt{s}\delta_{\max}\beta\rho^{\frac{3}{4}}\eta B_k\left\|\overline{\boldsymbol{u}}_{\mathcal{S}}^t\right\|_\infty^2 + \left(\sqrt{s}\delta_{\max} + 3\right)\left(4\beta\rho^{\frac{3}{4}}\eta B_k\left\|\overline{\boldsymbol{u}}_{\mathcal{S}}^t\right\|_\infty\right)^2$$
$$+ 2d\left(8\beta\rho^{\frac{3}{4}}\eta\left\|\overline{\boldsymbol{u}}_{\mathcal{S}^c}^t\right\|_\infty^2 B_k + \left(4\beta\rho^{\frac{3}{4}}\eta\left\|\overline{\boldsymbol{u}}_{\mathcal{S}^c}^t\right\|_\infty B_k\right)^2\right)$$
$$\leq \frac{3\sqrt{\rho}B_k}{16}, \tag{28}$$

where the first inequality is based on Lemma 2 and Lemma 3, the last inequality is due to step size, value of $\beta$, $\left\|\overline{\boldsymbol{u}}_{\mathcal{S}}^t\right\|_\infty^2 \leq 2w_{\max}^\star$, network connectivity condition and global RIP condition that $\rho^{\frac{1}{4}}\sqrt{s}\delta_{\max} \leq \sqrt{s}\delta \leq 1$.

For the perturbation $\boldsymbol{q}^t$ in (13), which is an error term outside the multiplicative updates in (10), based on induction (a), fine-grained upper in (c), there is $\forall j \in \mathcal{S}$

$$|q_j^t| \leq 4\beta\rho^{\frac{1}{4}}\eta B_k|\overline{u}_j|\left(\sqrt{s}\delta_{\max}\left\|\overline{\boldsymbol{u}}_{\mathcal{S}}^t \odot \overline{\boldsymbol{u}}_{\mathcal{S}}^t - \boldsymbol{w}^\star\right\|_\infty + \frac{3\sqrt{\rho}B_k}{16} + \max_i\left\|\frac{\boldsymbol{X}_i^T\boldsymbol{\xi}_i}{n}\right\|_\infty\right)$$
$$\leq \frac{\sqrt{\rho}B_k|\overline{u}_j|}{32}, \tag{29}$$

where the first inequality is due to (28) by comparing formula of $\boldsymbol{q}^t$ with $\boldsymbol{p}^t$ and last inequality is due to step size condition and such that $\rho^{\frac{1}{4}}\max_i\left\|\frac{\boldsymbol{X}_i^T\boldsymbol{\xi}_i}{n}\right\|_\infty \leq \left\|\frac{\boldsymbol{X}^T\boldsymbol{\xi}}{N}\right\|_\infty < w_{\max}^\star$. Then $\boldsymbol{q}^t$ could be reparameterized as $\boldsymbol{q}^t = \boldsymbol{r}_q^t \odot \overline{\boldsymbol{u}}_{\mathcal{S}}^t$ where $\left\|\boldsymbol{r}_q^t\right\|_\infty \leq \frac{\sqrt{\rho}B_k}{16}$ for $\forall t$ that $\overline{T}_{k-1} \leq t < \overline{T}_k$, the perturbed optimization recursion on support over decentralized network in (10) becomes

$$\left(\overline{\boldsymbol{u}}_{\mathcal{S}}^{t+1}\right)^2 = \left(\overline{\boldsymbol{u}}_{\mathcal{S}}^t\right)^2 \odot \left(\boldsymbol{1}_d - 4\eta\left(\overline{\boldsymbol{u}}_{\mathcal{S}}^t \odot \overline{\boldsymbol{u}}_{\mathcal{S}}^t - \boldsymbol{w}^\star + \boldsymbol{E}_2^t + \boldsymbol{E}_3^t + \boldsymbol{p}^t + \boldsymbol{r}_q^t\right)\right)^2. \tag{30}$$

where the perturbation errors $\boldsymbol{E}_2^t$ and $\boldsymbol{E}_3^t$ are defined as

$$\boldsymbol{E}_2^t = \left(\frac{\boldsymbol{X}^T\boldsymbol{X}}{N} - \boldsymbol{I}\right)\left(\overline{\boldsymbol{u}}_{\mathcal{S}}^t \odot \overline{\boldsymbol{u}}_{\mathcal{S}}^t - \boldsymbol{w}^\star\right)$$

$$\boldsymbol{E}_3^t = \frac{\boldsymbol{X}^T\boldsymbol{X}}{N}\left(\overline{\boldsymbol{u}}_{\mathcal{S}^c}^t \odot \overline{\boldsymbol{u}}_{\mathcal{S}^c}^t\right) - \frac{\boldsymbol{X}^T\boldsymbol{\xi}}{N}. \tag{31}$$

Because there is $\left\|\boldsymbol{E}_2^t\right\|_\infty + \left\|\boldsymbol{E}_3^t\right\|_\infty + \left\|\boldsymbol{p}^t\right\|_\infty + \left\|\boldsymbol{r}_q^t\right\|_\infty \le B_k$, which is based on the upper bound in (70). Then the proof is divided into the following two cases based on the magnitude of the element in $\boldsymbol{w}_{\mathcal{S}+}^\star$.

(1) $\forall j$ that $w_j^\star \ge 20B_k$, based on induction hypothesis (e) that $\left(\overline{u}_j^{\overline{T}_{k-1}}\right)^2 \le w_j^\star + 4B_k$, there is $\left(\overline{u}_j^{\overline{T}_{k-1}}\right)^2 \le \frac{6}{5}w_j^\star$, which illustrates that it satisfies the conditions in Lemma B.10 in [36]. Then because induction hypothesis (a) and (b) are true until $t$-th iteration, then if $B_k < \left\|\left(\overline{u}_j^t\right)^2 - w_j^\star\right\|_\infty \le \frac{w_{\max}^\star}{2^k}$, then $\left\|\left(\overline{u}_j^{t+1}\right)^2 - w_j^\star\right\|_\infty \le \left\|\left(\overline{u}_j^t\right)^2 - w_j^\star\right\|_\infty \le \frac{w_{\max}^\star}{2^k}$, else if $\left\|\left(\overline{u}_j^t\right)^2 - w_j^\star\right\|_\infty \le B_k$, then $\left\|\left(\overline{u}_j^{t+1}\right)^2 - w_j^\star\right\|_\infty \le B_k$. Combined with two cases, we can conclude that (a) also holds for $(t+1)$-th iteration for $j$ that $w_j^\star \ge 20B_k$.

(2) For arbitrary $j$-th elements whose magnitude is not sufficiently larger than the perturbation that $w_j^\star \le 20B_k$, based on the upper bound in induction (e), perturbation bound and monotonic property in Lemma B.6 in [36], we can guarantee that $\left(\overline{u}_j^t\right)^2$ would keep staying in $(0, w_j^\star + 4B_k]$. With condition $w_j^\star \le 20B_k$, we can conclude that $\left\|\left(\overline{u}_j^{t+1}\right)^2 - w_j^\star\right\|_\infty \le \max\{w_j^\star, 4B_k\} \le 20B_k \le \frac{w_{\max}^\star}{2^k}$.

Combined these two cases would finish proof of (a).

(b) To prove this statement, we should guarantee that there are sufficient iterative steps in the $(k-1)$-th induction that can make $\left\|\left(\boldsymbol{u}_{\mathcal{S}}^{\overline{T}_{k-1}}\right)^2 - \boldsymbol{w}^\star\right\|_\infty$ decrease at least by half from the beginning iteration of current induction stage to that of next induction stage. The proof is also divided into two cases.

(1) The one case is that $\forall j$ that it already has $\left|\left(\overline{u}_j^{\overline{T}_{k-1}}\right)^2 - w_j^\star\right| < \frac{w_{\max}^\star}{2^{k+1}}$, then with similar proof in (a), we can guarantee that $\forall t \ge \overline{T}_{k-1}$, $\left|\left(\overline{u}_j^t\right)^2 - w_j^\star\right| < \frac{w_{\max}^\star}{2^{k+1}}$. Thus, for these supports, we prove the $(k+1)$-th induction also holds.

(2) The second case is $\forall j$ that there is $\left|\left(\overline{u}_j^{\overline{T}_{k-1}}\right)^2 - w_j^\star\right| \ge \frac{w_{\max}^\star}{2^{k+1}} = 20B_k$. Based on the upper bound in induction (e) that $\left(\overline{u}_j^{\overline{T}_{k-1}}\right)^2 \le w_j^\star + 4B_k$, then $\overline{u}_j^{\overline{T}_{k-1}}, w_j^\star$ must satisfy $0 \le \left(\overline{u}_j^{\overline{T}_{k-1}}\right)^2 \le w_j^\star - 20B_k, w_j^\star \ge 20B_k$, respectively. This means $\overline{u}_j^{\overline{T}_{k-1}}$ is far away from $w_j^\star$ at least $20B_k$ distance. According to Lemma B.12 in [36], to achieve $\left|\left(\overline{u}_j^{\overline{T}_{k-1}}\right)^2 - w_j^\star\right| \le 20B_k$, the sufficient condition for the number of iterations $t$ in current induction stage is $t \ge \frac{15}{32\eta w_j^\star}\log\frac{(w_j^\star)^2}{19\left(\overline{u}_j^{\overline{T}_{k-1}}\right)^2 B_k}$. Now we verify the setting of $T_k$ as follows

$$T_k = \frac{2^k}{\eta w_{\max}^\star}\log\frac{1}{\alpha^4}$$

$$\ge \frac{1}{40\eta B_k}\log\left(\frac{3\left(w_{\max}^\star\right)^2}{\zeta}\cdot\frac{1}{\alpha^3}\right)$$

$$\ge \frac{1}{2\eta w_j^\star}\log\left(\frac{3\left(w_j^\star\right)^2}{\zeta}\cdot\frac{1}{\left(\overline{u}_j^{\overline{T}_{k-1}}\right)^2}\right)$$

$$\geq \frac{1}{2\eta w_j^\star} \log \frac{(w_j^\star)^2}{16\left(\overline{u}_j^{\overline{T}_{k-1}}\right)^2 B_k}, \tag{32}$$

where the first inequality is due to the definition of $B_k$ and small initialization condition that $\alpha \leq \frac{\zeta}{3(w_{\max}^\star)^2}$ and the second inequality is due to $w_j^\star \geq 20B_k$ and lower bound in induction (e) that $\left(\overline{u}_j^{\overline{T}_{k-1}}\right)^2 \geq \alpha^3$. The last inequality is because $\frac{\zeta}{3} \leq \frac{16}{40}\frac{w_{\max}^\star}{2^{K-1}} \leq \frac{16}{40}\frac{w_{\max}^\star}{2^k} = 16B_k$. Thus, combining the above two cases and similar proof, we can conclude that $\forall t \geq \overline{T}_k$, there is $\left\|(\overline{u}_{\mathcal{S}}^t)^2 - w^\star\right\|_\infty \leq \frac{w_{\max}^\star}{2^{k+1}}$. This completes the proof of induction (b).

(c) To make the consensus error satisfy the above induction, $\forall t$ that $\overline{T}_{k-1} \leq t < t+1 < \overline{T}_k$, based on one step iteration in (24) of Lemma 6, induction (a), (c) and step size condition $\eta \leq \frac{1-\sqrt{\rho}}{64\rho^{\frac{1}{2}}w_{\max}^\star}$, it has

$$\sqrt{\rho}\left(1 + 4\eta\left((\sqrt{s}\delta_{\max}+1)\left\|\overline{u}_{\mathcal{S}}^t \odot \overline{u}_{\mathcal{S}}^t - w^\star\right\|_\infty + 2d\left(\overline{u}_{\mathcal{S}^c}^t + \Delta_{\mathcal{S}^c}^t\right)^2 + 2(\sqrt{s}\delta_{\max}+1)\right.\right.$$
$$\left.\left.\cdot \left(\left\|\overline{u}_{\mathcal{S}}^t\right\|_\infty + \left\|\Delta_{\mathcal{S}}^t\right\|_\infty\right)^2 + \max_i\left\|\frac{X_i^T \xi_i}{n}\right\|_\infty\right)\right) \leq 1 + \frac{1-\sqrt{\rho}}{2}. \tag{33}$$

Then the recursion (24) in Lemma 6 becomes

$$\left\|\Delta_{\mathcal{S}}^{t+1}\right\|_\infty \leq 4\left(\frac{1+\sqrt{\rho}}{2}\right)\beta\rho^{\frac{3}{4}}\eta\left\|\overline{u}_{\mathcal{S}}^t\right\|_\infty B_k + 4\rho^{\frac{3}{4}}\eta\left\|\overline{u}_{\mathcal{S}}^t\right\|_\infty B_k$$
$$\leq \frac{4}{\left(1-c_{10}\left(1-\sqrt{\rho}\right)\right)^2}\left(\left(\frac{1+\sqrt{\rho}}{2}\right)\beta+1\right)\rho^{\frac{3}{4}}\eta\left\|\overline{u}_{\mathcal{S}}^{t+1}\right\|_\infty B_k, \tag{34}$$

where the last inequality is based on induction such as if $\left\|\Delta_{\mathcal{S}}^t\right\|_\infty \leq 4\beta\rho^{\frac{3}{4}}\eta\left\|\overline{u}_{\mathcal{S}}^t\right\|_\infty B_k$, then based on (30) and step size condition $\eta \leq \frac{c_{10}\left(1-\sqrt{\rho}\right)}{w_{\max}^\star}$, there is

$$\left\|\overline{u}_{\mathcal{S}}^{t+1}\right\|_\infty \geq \left\|\overline{u}_{\mathcal{S}}^t\right\|_\infty \left(1-c_{10}\left(1-\sqrt{\rho}\right)\right)^2. \tag{35}$$

To guarantee that (34) holds induction (c), the sufficient condition is

$$\frac{\left(\frac{1+\sqrt{\rho}}{2}\right)\beta+1}{\left(1-c_{10}\left(1-\sqrt{\rho}\right)\right)^2} \leq \beta. \tag{36}$$

The $c_{10}$ should be chosen that $\frac{1+\sqrt{\rho}}{2\left(1-c_{10}\left(1-\sqrt{\rho}\right)\right)^2} < 1$, which means that $c_{10}$ should satisfy $c_{10} \leq \frac{1-\sqrt{\frac{1+\sqrt{\rho}}{2}}}{1-\sqrt{\rho}}$. Then we can set $c_{10} = \frac{1-\left(\frac{1+\sqrt{\rho}}{2}\right)^{\frac{1}{4}}}{1-\sqrt{\rho}}$, which results in $\frac{1}{\left(1-c_{10}\left(1-\sqrt{\rho}\right)\right)^2} = \sqrt{\frac{2}{1+\sqrt{\rho}}}$. Based on (36), the lower bound for $\beta$ is $\beta \geq \frac{2\left(\sqrt{2(1+\sqrt{\rho})}+1+\sqrt{\rho}\right)}{1-\rho}$.

Unfolding the recursion in (24) from the beginning of $(k-1)$-th induction to the beginning of $k$-th induction based on (34) and combining induction (a) would have

$$\left\|\Delta_{\mathcal{S}}^{\overline{T}_k}\right\|_\infty \leq \left(\frac{1+\sqrt{\rho}}{2}\right)^{2^kT}\left\|\Delta_{\mathcal{S}}^{\overline{T}_{k-1}}\right\|_\infty + 4\rho^{\frac{3}{4}}\eta B_k \sum_{i=0}^{2^kT-1}\left(\frac{1+\sqrt{\rho}}{2}\right)^{2^kT-1-i}\left\|\overline{u}_{\mathcal{S}}^{\overline{T}_{k-1}+i}\right\|_\infty$$
$$\leq \frac{4\sqrt{2}\rho^{\frac{3}{4}}\eta B_k}{\sqrt{1+\sqrt{\rho}}}\sum_{i=0}^{2^kT-1}\left(\frac{1+\sqrt{\rho}}{2}\cdot\sqrt{\frac{2}{1+\sqrt{\rho}}}\right)^{2^kT-1-i}\left\|\overline{u}_{\mathcal{S}}^{\overline{T}_k}\right\|_\infty$$
$$+ 4\beta\rho^{\frac{3}{4}}\eta\left(\frac{1+\sqrt{\rho}}{2}\right)^{2^kT}\left\|\overline{u}_{\mathcal{S}}^{\overline{T}_{k-1}}\right\|_\infty B_k$$

$$\leq \frac{4\sqrt{2}\rho^{\frac{3}{4}}\eta B_k}{\sqrt{1+\sqrt{\rho}}} \sum_{i=0}^{2^kT-1} \left(\sqrt{\frac{1+\sqrt{\rho}}{2}}\right)^{2^kT-1-i} \left\|\overline{\boldsymbol{u}}_{\mathcal{S}}^{\overline{T}_k}\right\|_\infty$$

$$+ 4\beta\rho^{\frac{3}{4}}\eta \left(\frac{1+\sqrt{\rho}}{2}\cdot\sqrt{\frac{2}{1+\sqrt{\rho}}}\right)^{2^kT} \left\|\overline{\boldsymbol{u}}_{\mathcal{S}}^{\overline{T}_k}\right\|_\infty B_k$$

$$\leq 4\beta\rho^{\frac{3}{4}}\eta \left(\sqrt{\frac{1+\sqrt{\rho}}{2}}\right)^{2^kT} \left\|\overline{\boldsymbol{u}}_{\mathcal{S}}^{\overline{T}_k}\right\|_\infty B_k + \frac{2\sqrt{2}\left(1+\sqrt{\frac{1+\sqrt{\rho}}{2}}\right)}{\left(1-\sqrt{\rho}\right)\sqrt{1+\sqrt{\rho}}} \cdot 4\rho^{\frac{3}{4}}\eta \left\|\overline{\boldsymbol{u}}_{\mathcal{S}}^{\overline{T}_k}\right\|_\infty B_k$$

$$\leq 4\rho^{\frac{3}{4}}\eta \left\|\overline{\boldsymbol{u}}_{\mathcal{S}}^{\overline{T}_k}\right\|_\infty \left(\beta\left(\sqrt{\frac{1+\sqrt{\rho}}{2}}\right)^{2^kT} + \frac{8}{1-\sqrt{\rho}}\right) B_k, \tag{37}$$

where both the second and third inequalities use lower bound in (35). To guarantee that the last inequality satisfies the induction (c), the $\beta$ and $\rho$ should satisfy the following condition

$$B_k\left(\beta\left(\sqrt{\frac{1+\sqrt{\rho}}{2}}\right)^{2^kT} + \frac{8}{1-\sqrt{\rho}}\right) \leq B_{k+1}\beta. \tag{38}$$

One sufficient condition for achieving above inequality is $\frac{8}{1-\sqrt{\rho}} \leq \frac{\beta}{4}$. Combine above lower bound of $\beta$, we can verify that $\beta = \frac{32}{1-\sqrt{\rho}}$ satisfies this condition and following inequality is attained

$$\left(\sqrt{\frac{1+\sqrt{\rho}}{2}}\right)^{\frac{1}{\eta w_{\max}^\star}\log\frac{1}{\alpha^4}} \leq \frac{1}{4}, \tag{39}$$

which implies that the step size $\eta$ should satisfy

$$\eta \leq \frac{\log\frac{1}{\alpha^4}\ln\left(1+\sqrt{\frac{1+\sqrt{\rho}}{2}}-1\right)}{-2\ln 2w_{\max}^\star}. \tag{40}$$

One sufficient condition for achieving inequality is $\eta \leq \frac{\log\frac{1}{\alpha^4}\left(1-\sqrt{\frac{1+\sqrt{\rho}}{2}}\right)}{4w_{\max}^\star}$ based on inequality that $\ln\left(1+\sqrt{\frac{1+\sqrt{\rho}}{2}}-1\right) \leq \sqrt{\frac{1+\sqrt{\rho}}{2}}-1$. Combining all the above conditions on $\eta$, we obtain the upper bound as shown in condition (5).

(d) $\forall t$ that $\overline{T}_{k-1} \leq t < t+1 < \overline{T}_k$, based on one step iteration in (27) of Lemma 6 and inductions (a), (c), there is

$$\left\|\boldsymbol{\Delta}_{\mathcal{S}^c}^{t+1}\right\|_\infty \leq 4\left(\frac{1+\sqrt{\rho}}{2}\right)\gamma\rho^{\frac{3}{4}}\eta\left\|\overline{\boldsymbol{u}}_{\mathcal{S}^c}^t\right\|_\infty B_k + 4\rho^{\frac{3}{4}}\eta\left\|\overline{\boldsymbol{u}}_{\mathcal{S}^c}^t\right\|_\infty$$

$$\cdot\left(B_k + 4\sqrt{s}\delta\beta\rho^{\frac{3}{4}}\eta\left\|\overline{\boldsymbol{u}}_{\mathcal{S}}^t\right\|_\infty B_k\left(4\beta\rho^{\frac{3}{4}}\eta\left\|\overline{\boldsymbol{u}}_{\mathcal{S}}^t\right\|_\infty B_k + \left\|\overline{\boldsymbol{u}}_{\mathcal{S}}^t\right\|_\infty\right)\right)$$

$$\leq 4\left(\frac{1+\sqrt{\rho}}{2}\right)\gamma\rho^{\frac{3}{4}}\eta\left\|\overline{\boldsymbol{u}}_{\mathcal{S}^c}^t\right\|_\infty B_k + 4\rho^{\frac{3}{4}}\eta\left\|\overline{\boldsymbol{u}}_{\mathcal{S}^c}^t\right\|_\infty\left(B_k + 4\sqrt{s}\delta\beta\rho^{\frac{3}{4}}\eta B_k\left\|\overline{\boldsymbol{u}}_{\mathcal{S}}^t\right\|_\infty^2\right)$$

$$\leq 4\left(\frac{1+\sqrt{\rho}}{2}\right)\gamma\rho^{\frac{3}{4}}\eta\left\|\overline{\boldsymbol{u}}_{\mathcal{S}^c}^t\right\|_\infty B_j + 4\rho^{\frac{3}{4}}\eta\left\|\overline{\boldsymbol{u}}_{\mathcal{S}^c}^t\right\|_\infty B_k$$

$$\leq \frac{4}{\left(1-c_{10}\left(1-\sqrt{\rho}\right)\right)^2}\left(\left(\frac{1+\sqrt{\rho}}{2}\right)\gamma + 1\right)\rho^{\frac{3}{4}}\eta\left\|\overline{\boldsymbol{u}}_{\mathcal{S}^c}^{t+1}\right\|_\infty B_k, \tag{41}$$

where the second inequality is due to $4\beta\rho^{\frac{3}{4}}\eta B_j \leq 4\times\frac{32}{1-\sqrt{\rho}}\times\frac{1-\sqrt{\rho}}{256w_{\max}^\star}B_j \leq 1$ and third inequality is due to global RIP condition on $\delta$ that is order of $\frac{1}{\sqrt{s}}$ and $\beta\eta\left\|\overline{\boldsymbol{u}}_{\mathcal{S}}^t\right\|_\infty^2 \leq 1$, the last inequality is

based on induction such that if $\left\|\boldsymbol{\Delta}_{\mathcal{S}^c}^t\right\|_\infty \leq 4\gamma\rho^{\frac{3}{4}}\eta \left\|\overline{\boldsymbol{u}}_{\mathcal{S}^c}^t\right\|_\infty B_k$, then based (45), if $\eta \leq \frac{c_{10}(1-\sqrt{\rho})}{w_{\max}^\star}$, we have

$$\left\|\overline{\boldsymbol{u}}_{\mathcal{S}^c}^{t+1}\right\|_\infty \geq \left\|\overline{\boldsymbol{u}}_{\mathcal{S}^c}^t\right\|_\infty \left(1 - c_{10}\left(1 - \sqrt{\rho}\right)\right)^2. \tag{42}$$

With the same derivation in proving induction (c), the lower bound for $\gamma$ is the same as that of $\beta$.

Then also unrolling the recursion in (27) from beginning of $(k-1)$-th induction to beginning of $k$-th induction and combining induction (a) would have

$$\left\|\boldsymbol{\Delta}_{\mathcal{S}^c}^{\overline{T}_k}\right\|_\infty \leq \left(\frac{1+\sqrt{\rho}}{2}\right)^{2^k T} \left\|\boldsymbol{\Delta}_{\mathcal{S}^c}^{\overline{T}_{k-1}}\right\|_\infty + 4\rho^{\frac{3}{4}}\eta B_k \sum_{i=0}^{2^k T-1} \left(\frac{1+\sqrt{\rho}}{2}\right)^{2^k T-1-i} \left\|\overline{\boldsymbol{u}}_{\mathcal{S}^c}^{\overline{T}_{k-1}+i}\right\|_\infty$$

$$\leq 4\rho^{\frac{3}{4}}\eta \left\|\overline{\boldsymbol{u}}_{\mathcal{S}^c}^{\overline{T}_k}\right\|_\infty B_k \left(\gamma \left(\sqrt{\frac{1+\sqrt{\rho}}{2}}\right)^{2^k T} + \frac{8}{1-\sqrt{\rho}}\right). \tag{43}$$

The derivation of above inequality is similar with (37). To guarantee the above two inequality satisfy induction (d), the $\gamma$ and $\rho$ can have the same value as in (38) that $\gamma = \beta = \frac{32}{1-\sqrt{\rho}}$.

(e) For the upper bound, the proof is divided into three cases.

(1) One of case is $\left(\overline{u}_j^{\overline{T}_{k-1}}\right)^2 \leq w_j^\star + B_k$, based on the Lemma B.6 in [36], we can conclude that $\forall t \geq \overline{T}_{k-1}, \left(\overline{u}_j^t\right)^2$ would keep below $w_j^\star + B_k$.

(2) Another case is $w_j^\star + B_k \leq \left(\overline{u}_j^{\overline{T}_{k-1}}\right)^2 \leq w_j^\star + 2B_k$, in this case $\forall t \geq \overline{T}_{k-1}$, Either $0 \leq \left(\overline{u}_j^t\right)^2 \leq w_j^\star + B_k$, which can use result in case (1) that $\forall t' \geq t, 0 \leq \left(\overline{u}_j^{t'}\right)^2 \leq w_j^\star + B_k$ or that $w_j^\star + B_k \leq \left(\overline{u}_j^t\right)^2 \leq w_j^\star + 2B_k$, which would guarantee that $\left(\overline{u}_j^{t+1}\right)^2 \leq \left(\overline{u}_j^t\right)^2$.

(3) The last case is $w_j^\star + 2B_k < \left(\overline{u}_j^t\right)^2 \leq w_j^\star + 4B_k$, then based on Lemma B.14 [36], after the sufficient number of iterations that $\forall t \geq \frac{1}{10\eta B_k} = \frac{4 \times 2^k}{\eta w_{\max}^\star}$ and we can check that $T_k$ is large enough that satisfies this condition, it can keep that $\left(\overline{u}_j^{\overline{T}_{k-1}+t}\right)^2 \leq w_j^\star + 2B_k$.

Following the above three cases, we can guarantee that there exists $\left(\overline{u}_j^{\overline{T}_k}\right)^2 \leq w_j^\star + \frac{1}{2} \times 4B_k = w_j^\star + 4B_{k+1}$.

For the lower bound, we can also have bound $\left\|\boldsymbol{g}^t\right\|_\infty \leq \frac{3\sqrt{\rho}B_k}{8}$ by comparing formulas between $\boldsymbol{p}^t$ and $\boldsymbol{g}^t$, $\forall t$ that $\overline{T}_{k-1} \leq t < \overline{T}_k$. For perturbation $\boldsymbol{f}^t$, based on induction (a), fine-grained bound in (d), $\forall j \in \mathcal{S}^c$ it has

$$|f_j^t| \leq 4\beta\rho^{\frac{3}{4}}\eta|\overline{u}_j^t|B_k \left(\sqrt{s}\delta_{\max}\left\|\overline{\boldsymbol{u}}_{\mathcal{S}}^t \odot \overline{\boldsymbol{u}}_{\mathcal{S}}^t - \boldsymbol{w}^\star\right\|_\infty + \frac{3\sqrt{\rho}B_k}{16} + \max_i \left\|\frac{\boldsymbol{X}_i^T \boldsymbol{\xi}_i}{n}\right\|_\infty\right)$$

$$\leq \frac{\sqrt{\rho}B_k|\overline{u}_j^t|}{16}, \tag{44}$$

where the first inequality is due to bound for $\left\|\boldsymbol{g}^t\right\|_\infty$ and the last inequality is the same reason as (29). $\boldsymbol{f}^t$ can also be reparameterized as $\boldsymbol{f}^t = \boldsymbol{r}_f^t \odot \overline{\boldsymbol{u}}_{\mathcal{S}^c}^t$ where $\left\|\boldsymbol{r}_f^t\right\|_\infty \leq \frac{\sqrt{\rho}B_k}{16}$ for $\forall t$ that $\overline{T}_{k-1} \leq t < \overline{T}_k$, the perturbed optimization recursion on non-support $\mathcal{S}^c$ over decentralized network in (11) becomes

$$\left(\overline{\boldsymbol{u}}_{\mathcal{S}^c}^{t+1}\right)^2 = \left(\overline{\boldsymbol{u}}_{\mathcal{S}^c}^t\right)^2 \odot \left(\boldsymbol{1}_d - 4\eta \left(\boldsymbol{E}_2^t + \boldsymbol{E}_3^t + \boldsymbol{g}^t + \boldsymbol{r}_f^t\right)\right)^2. \tag{45}$$

Based on the similar lower bound (79) and upper bound in (80), we can conclude that $\forall t \leq \mathcal{O}\left(\frac{1}{\eta\zeta}\log\frac{1}{\alpha^4}\right)$, there is

$$\prod_{i=0}^{t}\left(1+8\eta\left\|\boldsymbol{E}_2^t\right\|_\infty+\left\|\boldsymbol{E}_3^t\right\|_\infty+\left\|\boldsymbol{g}^t\right\|_\infty+\left\|\boldsymbol{r}_f^t\right\|_\infty\right)^2\leq\frac{1}{\alpha}, \tag{46}$$

where this inequality guarantees that the averaged signal on non-support remains $\left\|\overline{\boldsymbol{u}}_{\mathcal{S}^c}^t\right\|_\infty\leq\sqrt{\alpha}$ until early stopping. $\forall j\in\mathcal{S}$, we use $j_k$ to denote the largest index that $w_j^\star\leq B_{j_k}+\alpha^3$. As $B_0=w_{\max}^\star$, the existence of $j_k$ is guaranteed. Then for $t=0,\cdots,\overline{T}_{j_k}-1$ and based on (30), the $\left(\overline{u}_j^t\right)^2$ would shrinkage from initialization $\alpha^2$, to obtain the lower bound, we should consider the maximum shrinkage as follows

$$\left(\overline{u}_j^{\overline{T}_{j_k}}\right)^2\geq\alpha^2\prod_{i=0}^{\overline{T}_{j_k}-1}\left(1-4\eta\left(\left(\overline{u}_j^i\right)^2+\left\|\boldsymbol{E}_2^i\right\|_\infty+\left\|\boldsymbol{E}_3^i\right\|_\infty+\left\|\boldsymbol{g}^i\right\|_\infty+\left\|\boldsymbol{r}_f^i\right\|_\infty\right)\right)^2$$

$$\geq\alpha^3, \tag{47}$$

where the last inequality is due to (46), $\left(\overline{u}_j^i\right)^2\leq\alpha^2,\forall i=1,\cdots,\overline{T}_{j_k}-1$ and step size condition that $\eta\left(\left(\overline{u}_j^i\right)^2+\left\|\boldsymbol{E}_2^i\right\|_\infty+\left\|\boldsymbol{E}_3^i\right\|_\infty+\left\|\boldsymbol{g}^i\right\|_\infty+\left\|\boldsymbol{r}_f^i\right\|_\infty\right)\leq\frac{3}{40}$ and $(1-5x)(1+8x)\geq1,\forall x\in[0,\frac{3}{40}]$. Thus, we have $\left(\overline{u}_j^t\right)^2\geq\alpha^3$ for $\forall t=0,\cdots,\overline{T}_{j_k}$.

For $t=\overline{T}_{j_k}+1,\cdots,\overline{T}_{j_k+1}-1$ and $\forall j\in\mathcal{S}$, let consider the auxiliary iterations that $\left(\hat{u}_j^t\right)^2=\left(\hat{u}_j^{t-1}\right)^2\odot\left(1-4\eta\left(\left(\hat{u}_j^{t-1}\right)^2-\left(w_j^\star-B_{j_k+1}\right)\right)\right)^2$ where $\hat{u}_j^{\overline{T}_{j_k}}=\overline{u}_j^{\overline{T}_{j_k}}\geq\alpha^3$ as above proved. Based on definition of $j_k$, there is $w_j^\star-B_{j_k+1}\geq\alpha^3$. According-ing to monotonic property in Lemma B.6 in [36], we can guarantee that $\left(\hat{u}_j^{\overline{T}_{j_k+1}}\right)^2\in\left[\min\left\{\hat{u}_j^{\overline{T}_{j_k}},w_j^\star-B_{j_k+1}\right\},\max\left\{\hat{u}_j^{\overline{T}_{j_k}},w_j^\star-B_{j_k+1}\right\}\right]$ that ensures that $\left(\overline{u}_j^{\overline{T}_{j_k+1}}\right)^2\geq\left(\hat{u}_j^{\overline{T}_{j_k+1}}\right)^2\geq\alpha^3$ where the first inequality is due to the squeezing property in Lemma B.9 in [36].

Then we can follow the same analysis to prove that left $i$-th induction step that $i=j_k+2,\cdots,m-1$ based on the monotonic property of $B_k$.

$\square$

## A.4 General Case

This section considers the general setting where the ground truth $\boldsymbol{w}^\star$ includes both positive and negative elements in its support. The analysis here is more complex than in the previous section due to the presence of $\boldsymbol{v}^{t,i}$ and its consensus error terms. The following formulates the recursion of perturbed average variable $\overline{\boldsymbol{u}}^t$ and similar derivation could be applied to $\overline{\boldsymbol{v}}^t$.

**Lemma 7.** *Consider the sequence $\{\boldsymbol{u}_i^t,\boldsymbol{v}_i^t\}, i\in[m]$ generated according to (3) and (4) by DGD for solving problem (2), the average signal $\{\overline{\boldsymbol{u}}^t\}$ on positive support $\mathcal{S}^+$, negative support $\mathcal{S}^-$ and non-support $\mathcal{S}^c$ are updated according to the following formulas*

$$\overline{\boldsymbol{u}}_{\mathcal{S}^+}^{t+1}=\overline{\boldsymbol{u}}_{\mathcal{S}^+}^t\odot\left(\mathbf{1}_d-4\eta\left(\left(\overline{\boldsymbol{u}}_{\mathcal{S}^+}^t\right)^2-\boldsymbol{w}_{\mathcal{S}^+}^\star\right)-4\eta\frac{\boldsymbol{X}^T\boldsymbol{X}}{N}\left(\left(\overline{\boldsymbol{u}}_{\mathcal{S}^c}^t\right)^2-\left(\overline{\boldsymbol{v}}_{\mathcal{S}^c}^t\right)^2+\left(\overline{\boldsymbol{u}}_{\mathcal{S}^-}^t\right)^2-\left(\overline{\boldsymbol{v}}_{\mathcal{S}^+}^t\right)^2\right)\right.$$

$$\left.+4\eta\frac{\boldsymbol{X}^T\boldsymbol{\xi}}{N}-4\eta\left(\frac{\boldsymbol{X}^T\boldsymbol{X}}{N}-\boldsymbol{I}\right)\left(\left(\overline{\boldsymbol{u}}_{\mathcal{S}^+}^t\right)^2-\boldsymbol{w}_{\mathcal{S}^+}^\star-\left(\overline{\boldsymbol{v}}_{\mathcal{S}^-}^t\right)^2-\boldsymbol{w}_{\mathcal{S}^-}^\star\right)-4\eta\boldsymbol{p}_u^t\right)-4\eta\boldsymbol{q}_u^t; \tag{48}$$

$$\overline{\boldsymbol{u}}_{\mathcal{S}^-}^{t+1}=\overline{\boldsymbol{u}}_{\mathcal{S}^-}^t\odot\left(\mathbf{1}_d-4\eta\left(-\left(\overline{\boldsymbol{v}}_{\mathcal{S}^-}^t\right)^2-\boldsymbol{w}_{\mathcal{S}^-}^\star+\frac{\boldsymbol{X}^T\boldsymbol{X}}{N}\left(\left(\overline{\boldsymbol{u}}_{\mathcal{S}^c}^t\right)^2-\left(\overline{\boldsymbol{v}}_{\mathcal{S}^c}^t\right)^2+\left(\overline{\boldsymbol{u}}_{\mathcal{S}^-}^t\right)^2-\left(\overline{\boldsymbol{v}}_{\mathcal{S}^+}^t\right)^2\right)\right.$$

$$-\frac{\boldsymbol{X}^T\boldsymbol{\xi}}{N} + \left(\frac{\boldsymbol{X}^T\boldsymbol{X}}{N} - \boldsymbol{I}\right)\left((\overline{\boldsymbol{u}}_{\mathcal{S}+}^t)^2 - \boldsymbol{w}_{\mathcal{S}+}^\star - (\overline{\boldsymbol{v}}_{\mathcal{S}-}^t)^2 - \boldsymbol{w}_{\mathcal{S}-}^\star\right)\bigg) - 4\eta\boldsymbol{y}_u^t\bigg) - 4\eta\boldsymbol{z}_u^t;$$

$$(49)$$

$$\overline{\boldsymbol{u}}_{\mathcal{S}^c}^{t+1} = \overline{\boldsymbol{u}}_{\mathcal{S}^c}^t \odot \left(\mathbf{1}_d - 4\eta\left(\frac{\boldsymbol{X}^T\boldsymbol{X}}{N}\left((\overline{\boldsymbol{u}}_{\mathcal{S}^c}^t)^2 - (\overline{\boldsymbol{v}}_{\mathcal{S}^c}^t)^2 + (\overline{\boldsymbol{u}}_{\mathcal{S}-}^t)^2 - (\overline{\boldsymbol{v}}_{\mathcal{S}+}^t)^2\right) - \frac{\boldsymbol{X}^T\boldsymbol{\xi}}{N}\right.$$

$$\left. + \left(\frac{\boldsymbol{X}^T\boldsymbol{X}}{N} - \boldsymbol{I}\right)\left((\overline{\boldsymbol{u}}_{\mathcal{S}+}^t)^2 - \boldsymbol{w}_{\mathcal{S}+}^\star - (\overline{\boldsymbol{v}}_{\mathcal{S}-}^t)^2 - \boldsymbol{w}_{\mathcal{S}-}^\star\right)\right) - 4\eta\boldsymbol{g}_u^t\right) - 4\eta\boldsymbol{f}_u^t, \qquad (50)$$

where the perturbed error terms $\boldsymbol{p}_u^t, \boldsymbol{q}_u^t, \boldsymbol{y}_u^t, \boldsymbol{z}_u^t, \boldsymbol{g}_u^t, \boldsymbol{f}_u^t$ induced from decentralized network are defined as

$$\boldsymbol{p}_u^t := \frac{1}{m}\sum_{i=1}^m \left(\frac{\boldsymbol{X}_i^T\boldsymbol{X}_i}{n} - \boldsymbol{I}\right)\left(2\overline{\boldsymbol{u}}_{\mathcal{S}+}^t \odot \boldsymbol{\Delta}_{\boldsymbol{u},\mathcal{S}+}^{t,i} + \left(\boldsymbol{\Delta}_{\boldsymbol{u},\mathcal{S}+}^{t,i}\right)^2 - 2\overline{\boldsymbol{v}}_{\mathcal{S}-}^t \odot \boldsymbol{\Delta}_{\boldsymbol{v},\mathcal{S}-}^{t,i} - \left(\boldsymbol{\Delta}_{\boldsymbol{v},\mathcal{S}-}^{t,i}\right)^2\right)$$

$$+ 3\left(\boldsymbol{\Delta}_{\boldsymbol{u},\mathcal{S}+}^{t,i}\right)^2 + \frac{\boldsymbol{X}_i^T\boldsymbol{X}_i}{n}\left(2\overline{\boldsymbol{u}}_{\mathcal{S}^c}^t \odot \boldsymbol{\Delta}_{\boldsymbol{u},\mathcal{S}^c}^{t,i} + \left(\boldsymbol{\Delta}_{\mathcal{S}^c}^{t,i}\right)^2 + 2\overline{\boldsymbol{u}}_{\mathcal{S}-}^t \odot \boldsymbol{\Delta}_{\boldsymbol{u},\mathcal{S}-}^{t,i} + \left(\boldsymbol{\Delta}_{\boldsymbol{u},\mathcal{S}-}^{t,i}\right)^2\right.$$

$$\left. - 2\overline{\boldsymbol{v}}_{\mathcal{S}^c}^t \odot \boldsymbol{\Delta}_{\boldsymbol{v},\mathcal{S}^c}^{t,i} - \left(\boldsymbol{\Delta}_{\boldsymbol{v},\mathcal{S}^c}^{t,i}\right)^2 - 2\overline{\boldsymbol{v}}_{\mathcal{S}+}^t \odot \boldsymbol{\Delta}_{\boldsymbol{v},\mathcal{S}+}^{t,i} - \left(\boldsymbol{\Delta}_{\boldsymbol{v},\mathcal{S}+}^{t,i}\right)^2\right); \qquad (51)$$

$$\boldsymbol{q}_u^t := \frac{1}{m}\sum_{i=1}^m \boldsymbol{\Delta}_{\boldsymbol{u},\mathcal{S}+}^{t,i} \odot \left(\left(\frac{\boldsymbol{X}_i^T\boldsymbol{X}_i}{n} - \boldsymbol{I}\right)\left((\overline{\boldsymbol{u}}_{\mathcal{S}+}^t)^2 - \boldsymbol{w}_{\mathcal{S}+}^\star - (\overline{\boldsymbol{v}}_{\mathcal{S}-}^t)^2 - \boldsymbol{w}_{\mathcal{S}-}^\star\right.\right.$$

$$\left. + 2\overline{\boldsymbol{u}}_{\mathcal{S}+}^t \odot \boldsymbol{\Delta}_{\boldsymbol{u},\mathcal{S}+}^{t,i} + \left(\boldsymbol{\Delta}_{\boldsymbol{u},\mathcal{S}+}^{t,i}\right)^2 - 2\overline{\boldsymbol{v}}_{\mathcal{S}-}^t \odot \boldsymbol{\Delta}_{\boldsymbol{v},\mathcal{S}-}^{t,i} - \left(\boldsymbol{\Delta}_{\boldsymbol{v},\mathcal{S}-}^{t,i}\right)^2\right)$$

$$+ \frac{\boldsymbol{X}_i^T\boldsymbol{X}_i}{n}\left(\left(\overline{\boldsymbol{u}}_{\mathcal{S}^c}^t + \boldsymbol{\Delta}_{\boldsymbol{u},\mathcal{S}^c}^{t,i}\right)^2 - \left(\overline{\boldsymbol{v}}_{\mathcal{S}^c}^t + \boldsymbol{\Delta}_{\boldsymbol{v},\mathcal{S}^c}^{t,i}\right)^2 + \left(\overline{\boldsymbol{u}}_{\mathcal{S}-}^t + \boldsymbol{\Delta}_{\boldsymbol{u},\mathcal{S}-}^{t,i}\right)^2 - \left(\overline{\boldsymbol{v}}_{\mathcal{S}+}^t + \boldsymbol{\Delta}_{\boldsymbol{v},\mathcal{S}+}^{t,i}\right)^2\right)$$

$$+ \left(\boldsymbol{\Delta}_{\boldsymbol{u},\mathcal{S}+}^{t,i}\right)^2 - \frac{\boldsymbol{X}_i^T\boldsymbol{\xi}_i}{n}\right); \qquad (52)$$

$$\boldsymbol{g}_u^t := \boldsymbol{p}_u^t - \frac{3}{m}\sum_{i=1}^m\left(\frac{\boldsymbol{X}_i^T\boldsymbol{X}_i}{n} - \boldsymbol{I}\right)\left(\boldsymbol{\Delta}_{\boldsymbol{u},\mathcal{S}+}^{t,i}\right)^2; \qquad (53)$$

$$\boldsymbol{f}_u^t := \frac{1}{m}\sum_{i=1}^m \boldsymbol{\Delta}_{\boldsymbol{u},\mathcal{S}^c}^{t,i} \odot \left(\left(\frac{\boldsymbol{X}_i^T\boldsymbol{X}_i}{n} - \boldsymbol{I}\right)\left((\overline{\boldsymbol{u}}_{\mathcal{S}+}^t)^2 - \boldsymbol{w}_{\mathcal{S}+}^\star - (\overline{\boldsymbol{v}}_{\mathcal{S}-}^t)^2 - \boldsymbol{w}_{\mathcal{S}-}^\star\right.\right.$$

$$\left. + 2\overline{\boldsymbol{u}}_{\mathcal{S}+}^t \odot \boldsymbol{\Delta}_{\boldsymbol{u},\mathcal{S}+}^{t,i} + \left(\boldsymbol{\Delta}_{\boldsymbol{u},\mathcal{S}+}^{t,i}\right)^2 - 2\overline{\boldsymbol{v}}_{\mathcal{S}-}^t \odot \boldsymbol{\Delta}_{\boldsymbol{v},\mathcal{S}-}^{t,i} - \left(\boldsymbol{\Delta}_{\boldsymbol{v},\mathcal{S}-}^{t,i}\right)^2\right) - \frac{\boldsymbol{X}_i^T\boldsymbol{\xi}_i}{n}$$

$$+ \frac{\boldsymbol{X}_i^T\boldsymbol{X}_i}{n}\left(\left(\overline{\boldsymbol{u}}_{\mathcal{S}^c}^t + \boldsymbol{\Delta}_{\boldsymbol{u},\mathcal{S}^c}^{t,i}\right)^2 - \left(\overline{\boldsymbol{v}}_{\mathcal{S}^c}^t + \boldsymbol{\Delta}_{\boldsymbol{v},\mathcal{S}^c}^{t,i}\right)^2 + \left(\overline{\boldsymbol{u}}_{\mathcal{S}-}^t + \boldsymbol{\Delta}_{\boldsymbol{u},\mathcal{S}-}^{t,i}\right)^2 - \left(\overline{\boldsymbol{v}}_{\mathcal{S}+}^t + \boldsymbol{\Delta}_{\boldsymbol{v},\mathcal{S}+}^{t,i}\right)^2\right)\right);$$

$$(54)$$

$$\boldsymbol{y}_u^t := \boldsymbol{g}_u^t - \frac{1}{m}\sum_{i=1}^m\left(\boldsymbol{\Delta}_{\boldsymbol{v},\mathcal{S}-}^{t,i}\right)^2; \qquad (55)$$

$$\boldsymbol{z}_u^t := \frac{1}{m}\sum_{i=1}^m \boldsymbol{\Delta}_{\boldsymbol{u},\mathcal{S}-}^{t,i} \odot \left(-2\overline{\boldsymbol{v}}_{\mathcal{S}-}^t \odot \boldsymbol{\Delta}_{\boldsymbol{v},\mathcal{S}-}^{t,i} - \left(\boldsymbol{\Delta}_{\boldsymbol{v},\mathcal{S}-}^{t,i}\right)^2 + \left(\frac{\boldsymbol{X}_i^T\boldsymbol{X}_i}{n} - \boldsymbol{I}\right)\left((\overline{\boldsymbol{u}}_{\mathcal{S}+}^t)^2 - \boldsymbol{w}_{\mathcal{S}+}^\star - (\overline{\boldsymbol{v}}_{\mathcal{S}-}^t)^2\right.\right.$$

$$\left. - \boldsymbol{w}_{\mathcal{S}-}^\star + 2\overline{\boldsymbol{u}}_{\mathcal{S}+}^t \odot \boldsymbol{\Delta}_{\boldsymbol{u},\mathcal{S}+}^{t,i} + \left(\boldsymbol{\Delta}_{\boldsymbol{u},\mathcal{S}+}^{t,i}\right)^2 - 2\overline{\boldsymbol{v}}_{\mathcal{S}-}^t \odot \boldsymbol{\Delta}_{\boldsymbol{v},\mathcal{S}-}^{t,i} - \left(\boldsymbol{\Delta}_{\boldsymbol{v},\mathcal{S}-}^{t,i}\right)^2\right)$$

$$+ \frac{\boldsymbol{X}_i^T\boldsymbol{X}_i}{n}\left(\left(\overline{\boldsymbol{u}}_{\mathcal{S}^c}^t + \boldsymbol{\Delta}_{\boldsymbol{u},\mathcal{S}^c}^{t,i}\right)^2 - \left(\overline{\boldsymbol{v}}_{\mathcal{S}^c}^t + \boldsymbol{\Delta}_{\boldsymbol{v},\mathcal{S}^c}^{t,i}\right)^2 + \left(\overline{\boldsymbol{u}}_{\mathcal{S}-}^t + \boldsymbol{\Delta}_{\boldsymbol{u},\mathcal{S}-}^{t,i}\right)^2 - \left(\overline{\boldsymbol{v}}_{\mathcal{S}+}^t + \boldsymbol{\Delta}_{\boldsymbol{v},\mathcal{S}+}^{t,i}\right)^2\right)$$

$$+ \left(\boldsymbol{\Delta}_{\boldsymbol{u},\mathcal{S}+}^{t,i}\right)^2 - \frac{\boldsymbol{X}_i^T\boldsymbol{\xi}_i}{n}\right); \qquad (56)$$

*Proof.* The average of optimization for $\boldsymbol{u}^{t,i}, i \in [m]$ under DGD are as follows

$$\overline{\boldsymbol{u}}^{t+1} = \overline{\boldsymbol{u}}^t - \frac{\eta}{m} \sum_{i=1}^{m} \nabla_{\boldsymbol{u}} f_i \left( \boldsymbol{u}^{t,i}, \boldsymbol{v}^{t,i} \right)$$

$$= \overline{\boldsymbol{u}}^t - \frac{\eta}{m} \sum_{i=1}^{m} \nabla_{\boldsymbol{u}} f_i \left( \overline{\boldsymbol{u}}^t, \overline{\boldsymbol{v}}^t \right) + \frac{\eta}{m} \sum_{i=1}^{m} \left( \underbrace{\nabla_{\boldsymbol{u}} f_i \left( \overline{\boldsymbol{u}}^t, \overline{\boldsymbol{v}}^t \right) - \nabla_{\boldsymbol{u}} f_i \left( \overline{\boldsymbol{u}}^t, \boldsymbol{v}^{t,i} \right)}_{\Pi_{1,i}} \right)$$

$$+ \frac{\eta}{m} \sum_{i=1}^{m} \left( \underbrace{\nabla_{\boldsymbol{u}} f_i \left( \overline{\boldsymbol{u}}^t, \boldsymbol{v}^{t,i} \right) - \nabla_{\boldsymbol{u}} f_i \left( \boldsymbol{u}^{t,i}, \boldsymbol{v}^{t,i} \right)}_{\Pi_{2,i}} \right). \tag{57}$$

Based on error decomposition on positive support $\mathcal{S}^+$, negative support $\mathcal{S}^-$ and non-support $\mathcal{S}^c$, there has

$$\nabla_{\boldsymbol{u}} F \left( \overline{\boldsymbol{u}}^t, \overline{\boldsymbol{v}}^t \right) = 4\overline{\boldsymbol{u}}^t \odot \left( \left( \overline{\boldsymbol{u}}_{\mathcal{S}+}^t \right)^2 - \left( \overline{\boldsymbol{v}}_{\mathcal{S}-}^t \right)^2 - \boldsymbol{w}^\star + \left( \frac{\boldsymbol{X}^T \boldsymbol{X}}{N} - \boldsymbol{I} \right) \left( \left( \overline{\boldsymbol{u}}_{\mathcal{S}+}^t \right)^2 - \left( \overline{\boldsymbol{v}}_{\mathcal{S}-}^t \right)^2 - \boldsymbol{w}^\star \right) \right.$$

$$\left. + \frac{\boldsymbol{X}^T \boldsymbol{X}}{N} \left( \left( \overline{\boldsymbol{u}}_{\mathcal{S}^c}^t \right)^2 - \left( \overline{\boldsymbol{v}}_{\mathcal{S}^c}^t \right)^2 + \left( \overline{\boldsymbol{u}}_{\mathcal{S}-}^t \right)^2 - \left( \overline{\boldsymbol{v}}_{\mathcal{S}+}^t \right)^2 \right) - \frac{\boldsymbol{X}^T \boldsymbol{\xi}}{N} \right). \tag{58}$$

With same decomposition on $\mathcal{S}^+, \mathcal{S}^-, \mathcal{S}^c$ for $\Pi_{1,i}$ and $\Pi_{2,i}$, there are

$$\Pi_{1,i} = 4\overline{\boldsymbol{u}}^t \odot \left( 2\overline{\boldsymbol{v}}_{\mathcal{S}-}^t \odot \boldsymbol{\Delta}_{\boldsymbol{v},\mathcal{S}-}^{t,i} + \left( \boldsymbol{\Delta}_{\boldsymbol{v},\mathcal{S}-}^{t,i} \right)^2 + \left( \frac{\boldsymbol{X}_i^T \boldsymbol{X}_i}{n} - \boldsymbol{I} \right) \left( 2\overline{\boldsymbol{v}}_{\mathcal{S}-}^t \odot \boldsymbol{\Delta}_{\boldsymbol{v},\mathcal{S}-}^{t,i} + \left( \boldsymbol{\Delta}_{\boldsymbol{v},\mathcal{S}-}^{t,i} \right)^2 \right) \right.$$

$$\left. \frac{\boldsymbol{X}_i^T \boldsymbol{X}_i}{n} \left( 2\overline{\boldsymbol{v}}_{\mathcal{S}^c}^t \odot \boldsymbol{\Delta}_{\boldsymbol{v},\mathcal{S}^c}^{t,i} + \left( \boldsymbol{\Delta}_{\boldsymbol{v},\mathcal{S}^c}^{t,i} \right)^2 + 2\overline{\boldsymbol{v}}_{\mathcal{S}+}^t \odot \boldsymbol{\Delta}_{\boldsymbol{v},\mathcal{S}+}^{t,i} + \left( \boldsymbol{\Delta}_{\boldsymbol{v},\mathcal{S}+}^{t,i} \right)^2 \right) \right);$$

$$\Pi_{2,i} = 4\overline{\boldsymbol{u}}^t \odot \left( -2\overline{\boldsymbol{u}}_{\mathcal{S}+}^t \odot \boldsymbol{\Delta}_{\boldsymbol{u},\mathcal{S}+}^{t,i} - \left( \boldsymbol{\Delta}_{\boldsymbol{u},\mathcal{S}+}^{t,i} \right)^2 + \left( \frac{\boldsymbol{X}_i^T \boldsymbol{X}_i}{n} - \boldsymbol{I} \right) \left( -2\overline{\boldsymbol{u}}_{\mathcal{S}+}^t \odot \boldsymbol{\Delta}_{\boldsymbol{u},\mathcal{S}+}^{t,i} - \left( \boldsymbol{\Delta}_{\boldsymbol{u},\mathcal{S}+}^{t,i} \right)^2 \right) \right.$$

$$+ \frac{\boldsymbol{X}_i^T \boldsymbol{X}_i}{n} \left( -2\overline{\boldsymbol{u}}_{\mathcal{S}^c}^t \odot \boldsymbol{\Delta}_{\boldsymbol{u},\mathcal{S}^c}^{t,i} - \left( \boldsymbol{\Delta}_{\boldsymbol{u},\mathcal{S}^c}^{t,i} \right)^2 - 2\overline{\boldsymbol{u}}_{\mathcal{S}-}^t \odot \boldsymbol{\Delta}_{\boldsymbol{u},\mathcal{S}-}^{t,i} - \left( \boldsymbol{\Delta}_{\boldsymbol{u},\mathcal{S}-}^{t,i} \right)^2 \right) \right)$$

$$- 4\boldsymbol{\Delta}_{\boldsymbol{u}}^{t,i} \odot \left( \left( \overline{\boldsymbol{u}}_{\mathcal{S}+}^t \right)^2 - \left( \overline{\boldsymbol{v}}_{\mathcal{S}-}^t \right)^2 - \boldsymbol{w}^\star + 2\overline{\boldsymbol{u}}_{\mathcal{S}+}^t \odot \boldsymbol{\Delta}_{\boldsymbol{u},\mathcal{S}+}^{t,i} + \left( \boldsymbol{\Delta}_{\boldsymbol{u},\mathcal{S}+}^{t,i} \right)^2 - \boldsymbol{\Delta}_{\boldsymbol{v},\mathcal{S}-}^{t,i} \left( 2\overline{\boldsymbol{v}}_{\mathcal{S}-}^t + \boldsymbol{\Delta}_{\boldsymbol{v},\mathcal{S}-}^{t,i} \right) \right.$$

$$+ \left( \frac{\boldsymbol{X}_i^T \boldsymbol{X}_i}{n} - \boldsymbol{I} \right) \left( \left( \overline{\boldsymbol{u}}_{\mathcal{S}+}^t \right)^2 - \left( \overline{\boldsymbol{v}}_{\mathcal{S}-}^t \right)^2 - \boldsymbol{w}^\star + 2\overline{\boldsymbol{u}}_{\mathcal{S}+}^t \odot \boldsymbol{\Delta}_{\boldsymbol{u},\mathcal{S}+}^{t,i} + \left( \boldsymbol{\Delta}_{\boldsymbol{u},\mathcal{S}+}^{t,i} \right)^2 - 2\overline{\boldsymbol{v}}_{\mathcal{S}-}^t \odot \boldsymbol{\Delta}_{\boldsymbol{v},\mathcal{S}-}^{t,i} \right.$$

$$\left. - \left( \boldsymbol{\Delta}_{\boldsymbol{v},\mathcal{S}-}^{t,i} \right)^2 \right) + \frac{\boldsymbol{X}_i^T \boldsymbol{X}_i}{n} \left( \left( \overline{\boldsymbol{u}}_{\mathcal{S}^c}^t \right)^2 - \left( \overline{\boldsymbol{v}}_{\mathcal{S}^c}^t \right)^2 + 2\overline{\boldsymbol{u}}_{\mathcal{S}^c}^t \odot \boldsymbol{\Delta}_{\boldsymbol{u},\mathcal{S}^c}^{t,i} + \left( \boldsymbol{\Delta}_{\boldsymbol{u},\mathcal{S}^c}^{t,i} \right)^2 - 2\overline{\boldsymbol{v}}_{\mathcal{S}^c}^t \odot \boldsymbol{\Delta}_{\boldsymbol{v},\mathcal{S}^c}^{t,i} \right.$$

$$\left. - \left( \boldsymbol{\Delta}_{\boldsymbol{v},\mathcal{S}^c}^{t,i} \right)^2 + \left( \overline{\boldsymbol{u}}_{\mathcal{S}-}^t \right)^2 - \left( \overline{\boldsymbol{v}}_{\mathcal{S}+}^t \right)^2 + 2\overline{\boldsymbol{u}}_{\mathcal{S}-}^t \odot \boldsymbol{\Delta}_{\boldsymbol{u},\mathcal{S}-}^{t,i} + \left( \boldsymbol{\Delta}_{\boldsymbol{u},\mathcal{S}-}^{t,i} \right)^2 - 2\overline{\boldsymbol{v}}_{\mathcal{S}+}^t \odot \boldsymbol{\Delta}_{\boldsymbol{v},\mathcal{S}+}^{t,i} \right.$$

$$\left. - \left( \boldsymbol{\Delta}_{\boldsymbol{v},\mathcal{S}+}^{t,i} \right)^2 - \frac{\boldsymbol{X}_i^T \boldsymbol{\xi}_i}{n} \right) \right). \tag{59}$$

Substituting (59) into (57) would obtain the perturbed recursion of averaged $\overline{\boldsymbol{u}}_{\mathcal{S}+}^t, \overline{\boldsymbol{u}}_{\mathcal{S}-}^t, \overline{\boldsymbol{u}}_{\mathcal{S}^c}^t$ in Lemma 7. $\qquad \square$

The following lemma separately bounds the consensus errors on $\mathcal{S}^+$, $\mathcal{S}^-$, and $\mathcal{S}^c$ by the corresponding magnitudes of parameters, which is different from the current analysis in the decentralized optimization literature.

**Lemma 8.** *Consider the sequence $\{\boldsymbol{u}^{t,i}, \boldsymbol{v}^{t,i}\}, i \in [m]$ generated according to (3) and (4) by DGD for solving (2), the consensus error of $\boldsymbol{\Delta}_u^t$ on positive support $\mathcal{S}^+$, negative support $\mathcal{S}^-$ and non-support $\mathcal{S}^c$ have the following recursions*

$$
\begin{aligned}
\left\|\boldsymbol{\Delta}_{\boldsymbol{u},\mathcal{S}^+}^{t+1}\right\|_\infty \leq{}& \rho \left\|\boldsymbol{\Delta}_{\boldsymbol{u},\mathcal{S}^+}^t\right\|_\infty \cdot \Bigg(1 + 4\eta\Bigg( (\sqrt{s}\delta_{\max}+1)\Big(\left\|(\overline{\boldsymbol{u}}_{\mathcal{S}^+}^t)^2 - \boldsymbol{w}_{\mathcal{S}^+}^\star\right\|_\infty + \left\|(\overline{\boldsymbol{v}}_{\mathcal{S}^-}^t)^2 - \boldsymbol{w}_{\mathcal{S}^-}^\star\right\|_\infty\Big) \\
&+ 2d\Big( \big(\|\overline{\boldsymbol{u}}_{\mathcal{S}^c}^t\|_\infty + \|\boldsymbol{\Delta}_{\boldsymbol{u},\mathcal{S}^c}^t\|_\infty\big)^2 + \big(\|\overline{\boldsymbol{v}}_{\mathcal{S}^c}^t\|_\infty + \|\boldsymbol{\Delta}_{\boldsymbol{v},\mathcal{S}^c}^t\|_\infty\big)^2 \\
&+ \big(\|\overline{\boldsymbol{u}}_{\mathcal{S}^-}^t\|_\infty + \|\boldsymbol{\Delta}_{\boldsymbol{u},\mathcal{S}^-}^t\|_\infty\big)^2 + \big(\|\overline{\boldsymbol{v}}_{\mathcal{S}^+}^t\|_\infty + \|\boldsymbol{\Delta}_{\boldsymbol{v},\mathcal{S}^+}^t\|_\infty\big)^2\Big) \\
&+ 2\left(\sqrt{s}\delta_{\max}+1\right)\big(\|\overline{\boldsymbol{u}}_{\mathcal{S}^+}^t\|_\infty + \|\boldsymbol{\Delta}_{\boldsymbol{u},\mathcal{S}^+}^t\|\big)^2 + \max_i\left\|\frac{\boldsymbol{X}_i^T\boldsymbol{\xi}_i}{n}\right\|_\infty\Bigg)\Bigg) \\
&+ 4\rho\eta\|\overline{\boldsymbol{u}}_{\mathcal{S}^+}^t\|_\infty \cdot \Big(\sqrt{s}\delta_{\max}\big(2\|\overline{\boldsymbol{v}}_{\mathcal{S}^-}^t\|_\infty \|\boldsymbol{\Delta}_{\boldsymbol{v},\mathcal{S}^-}^t\|_\infty + \|\boldsymbol{\Delta}_{\boldsymbol{v},\mathcal{S}^-}^t\|_\infty^2\big) + C_{err}^t\Big);
\end{aligned}
\tag{60}
$$

$$
\begin{aligned}
\left\|\boldsymbol{\Delta}_{\boldsymbol{u},\mathcal{S}^-}^{t+1}\right\|_\infty \leq{}& \rho \left\|\boldsymbol{\Delta}_{\boldsymbol{u},\mathcal{S}^-}^t\right\|_\infty \cdot \Bigg(1 + 4\rho\eta\Bigg((\sqrt{s}\delta_{\max}+1)\Big(\left\|(\overline{\boldsymbol{u}}_{\mathcal{S}^+}^t)^2 - \boldsymbol{w}_{\mathcal{S}^+}^\star\right\|_\infty + \left\|(\overline{\boldsymbol{v}}_{\mathcal{S}^-}^t)^2 - \boldsymbol{w}_{\mathcal{S}^-}^\star\right\|_\infty \\
&+ 2\|\overline{\boldsymbol{u}}_{\mathcal{S}^+}^t\|_\infty \|\boldsymbol{\Delta}_{\boldsymbol{u},\mathcal{S}^+}^t\|_\infty + \|\boldsymbol{\Delta}_{\boldsymbol{u},\mathcal{S}^+}^t\|_\infty^2 + 2\|\overline{\boldsymbol{v}}_{\mathcal{S}^-}^t\|_\infty \|\boldsymbol{\Delta}_{\boldsymbol{v},\mathcal{S}^-}^t\|_\infty + \|\boldsymbol{\Delta}_{\boldsymbol{v},\mathcal{S}^-}^t\|_\infty^2\Big) \\
&+ 4d\Big( \big(\|\overline{\boldsymbol{u}}_{\mathcal{S}^c}^t\|_\infty + \|\boldsymbol{\Delta}_{\boldsymbol{u},\mathcal{S}^c}^t\|_\infty\big)^2 + \big(\|\overline{\boldsymbol{v}}_{\mathcal{S}^c}^t\|_\infty + \|\boldsymbol{\Delta}_{\boldsymbol{v},\mathcal{S}^c}^t\|_\infty\big)^2 \\
&+ \big(\|\overline{\boldsymbol{u}}_{\mathcal{S}^-}^t\|_\infty + \|\boldsymbol{\Delta}_{\boldsymbol{u},\mathcal{S}^-}^t\|_\infty\big)^2 + \big(\|\overline{\boldsymbol{v}}_{\mathcal{S}^+}^t\|_\infty + \|\boldsymbol{\Delta}_{\boldsymbol{v},\mathcal{S}^+}^t\|_\infty\big)^2\Big) + \max_i\left\|\frac{\boldsymbol{X}_i^T\boldsymbol{\xi}_i}{n}\right\|_\infty\Bigg)\Bigg) \\
&+ 4\rho\eta\|\overline{\boldsymbol{u}}_{\mathcal{S}^-}^t\|_\infty \cdot \Big((\sqrt{s}\delta_{\max}+1)\cdot\big(2\|\overline{\boldsymbol{u}}_{\mathcal{S}^+}^t\|_\infty \|\boldsymbol{\Delta}_{\boldsymbol{u},\mathcal{S}^+}^t\|_\infty + \|\boldsymbol{\Delta}_{\boldsymbol{u},\mathcal{S}^+}^t\|_\infty^2 \\
&+ 2\|\overline{\boldsymbol{v}}_{\mathcal{S}^-}^t\|_\infty \|\boldsymbol{\Delta}_{\boldsymbol{v},\mathcal{S}^-}^t\|_\infty + \|\boldsymbol{\Delta}_{\boldsymbol{v},\mathcal{S}^-}^t\|_\infty^2\big) + C_{err}\Big);
\end{aligned}
\tag{61}
$$

$$
\begin{aligned}
\left\|\boldsymbol{\Delta}_{\boldsymbol{u},\mathcal{S}^c}^{t+1}\right\|_\infty \leq{}& \rho \left\|\boldsymbol{\Delta}_{\boldsymbol{u},\mathcal{S}^c}^t\right\|_\infty \cdot \Bigg(1 + 4\eta\Bigg(\sqrt{s}\delta_{\max}\Big(\left\|(\overline{\boldsymbol{u}}_{\mathcal{S}^+}^t)^2 - \boldsymbol{w}_{\mathcal{S}^+}^\star\right\|_\infty + \left\|(\overline{\boldsymbol{v}}_{\mathcal{S}^-}^t)^2 - \boldsymbol{w}_{\mathcal{S}^-}^\star\right\|_\infty \\
&+ 2\|\overline{\boldsymbol{u}}_{\mathcal{S}^+}^t\|_\infty \|\boldsymbol{\Delta}_{\boldsymbol{u},\mathcal{S}^+}^t\|_\infty + \|\boldsymbol{\Delta}_{\boldsymbol{u},\mathcal{S}^+}^t\|_\infty^2 + 2\|\overline{\boldsymbol{v}}_{\mathcal{S}^-}^t\|_\infty \|\boldsymbol{\Delta}_{\boldsymbol{v},\mathcal{S}^-}^t\|_\infty + \|\boldsymbol{\Delta}_{\boldsymbol{v},\mathcal{S}^-}^t\|_\infty^2\Big) \\
&+ 4d\Big( \big(\|\overline{\boldsymbol{u}}_{\mathcal{S}^c}^t\|_\infty + \|\boldsymbol{\Delta}_{\boldsymbol{u},\mathcal{S}^c}^t\|_\infty\big)^2 + \big(\|\overline{\boldsymbol{v}}_{\mathcal{S}^c}^t\|_\infty + \|\boldsymbol{\Delta}_{\boldsymbol{v},\mathcal{S}^c}^t\|_\infty\big)^2 \\
&+ \big(\|\overline{\boldsymbol{u}}_{\mathcal{S}^-}^t\|_\infty + \|\boldsymbol{\Delta}_{\boldsymbol{u},\mathcal{S}^-}^t\|_\infty\big)^2 + \big(\|\overline{\boldsymbol{v}}_{\mathcal{S}^+}^t\|_\infty + \|\boldsymbol{\Delta}_{\boldsymbol{v},\mathcal{S}^+}^t\|_\infty\big)^2\Big) + \max_i\left\|\frac{\boldsymbol{X}_i^T\boldsymbol{\xi}_i}{n}\right\|_\infty\Bigg)\Bigg) \\
&+ 4\rho\eta\|\overline{\boldsymbol{u}}_{\mathcal{S}^c}^t\|_\infty \cdot \Big(\sqrt{s}\delta_{\max}\cdot\big(2\|\overline{\boldsymbol{u}}_{\mathcal{S}^+}^t\|_\infty \|\boldsymbol{\Delta}_{\boldsymbol{u},\mathcal{S}^+}^t\|_\infty + \|\boldsymbol{\Delta}_{\boldsymbol{u},\mathcal{S}^+}^t\|_\infty^2 \\
&+ 2\|\overline{\boldsymbol{v}}_{\mathcal{S}^-}^t\|_\infty \|\boldsymbol{\Delta}_{\boldsymbol{v},\mathcal{S}^-}^t\|_\infty + \|\boldsymbol{\Delta}_{\boldsymbol{v},\mathcal{S}^-}^t\|_\infty^2\big) + C_{err}^t\Big),
\end{aligned}
\tag{62}
$$

*where the $C_{err}^t$ is defined as*

$$
\begin{aligned}
C_{err}^t :={}& \sqrt{s}(\delta_{\max}+\delta)\cdot\Big(\left\|(\overline{\boldsymbol{u}}_{\mathcal{S}^+}^t)^2 - \boldsymbol{w}_{\mathcal{S}^+}^\star\right\|_\infty + \left\|(\overline{\boldsymbol{v}}_{\mathcal{S}^-}^t)^2 - \boldsymbol{w}_{\mathcal{S}^-}^\star\right\|_\infty\Big) \\
&+ 2d\Big( \big(\|\overline{\boldsymbol{u}}_{\mathcal{S}^c}^t\|_\infty + \|\boldsymbol{\Delta}_{\boldsymbol{u},\mathcal{S}^c}^t\|_\infty\big)^2 + \big(\|\overline{\boldsymbol{v}}_{\mathcal{S}^c}^t\|_\infty + \|\boldsymbol{\Delta}_{\boldsymbol{v},\mathcal{S}^c}^t\|_\infty\big)^2 \\
&+ \big(\|\overline{\boldsymbol{u}}_{\mathcal{S}^-}^t\|_\infty + \|\boldsymbol{\Delta}_{\boldsymbol{u},\mathcal{S}^-}^t\|_\infty\big)^2 + \big(\|\overline{\boldsymbol{v}}_{\mathcal{S}^+}^t\|_\infty + \|\boldsymbol{\Delta}_{\boldsymbol{v},\mathcal{S}^+}^t\|_\infty\big)^2\Big) \\
&+ \max_i\left\|\frac{\boldsymbol{X}_i^T\boldsymbol{\xi}_i}{n}\right\|_\infty + \left\|\frac{\boldsymbol{X}^T\boldsymbol{\xi}}{N}\right\|_\infty.
\end{aligned}
\tag{63}
$$

*Proof.* Based on the updating of DGD, there is

$$\boldsymbol{U}^{t+1}\left(\boldsymbol{I}_m - \frac{1}{m}\boldsymbol{1}_m\boldsymbol{1}_m^T\right) = \left(\boldsymbol{U}^t - \eta\nabla_{\boldsymbol{u}}\boldsymbol{f}(\boldsymbol{U}^t,\boldsymbol{V}^t)\right)\boldsymbol{W}\left(\boldsymbol{I}_m - \frac{1}{m}\boldsymbol{1}_m\boldsymbol{1}_m^T\right)$$

$$= \left(\boldsymbol{U}^t - \nabla_{\boldsymbol{u}}\boldsymbol{f}\left(\boldsymbol{U}^t,\boldsymbol{V}^t\right) + \nabla_{\boldsymbol{u}}\boldsymbol{F}\left(\overline{\boldsymbol{U}}^t,\overline{\boldsymbol{V}}^t\right)\right)\left(\boldsymbol{W} - \frac{1}{m}\boldsymbol{1}_m\boldsymbol{1}_m^T\right)$$

$$= \left(\boldsymbol{U}^t - \nabla_{\boldsymbol{u}}\boldsymbol{f}\left(\boldsymbol{U}^t,\boldsymbol{V}^t\right) + \nabla_{\boldsymbol{u}}\boldsymbol{f}\left(\overline{\boldsymbol{U}}^t,\boldsymbol{V}^t\right)\right.$$

$$+ \nabla_{\boldsymbol{u}}\boldsymbol{f}\left(\overline{\boldsymbol{U}}^t,\overline{\boldsymbol{V}}^t\right) - \nabla_{\boldsymbol{u}}\boldsymbol{f}\left(\overline{\boldsymbol{U}}^t,\boldsymbol{V}^t\right)$$

$$\left.+ \nabla_{\boldsymbol{u}}\boldsymbol{F}\left(\overline{\boldsymbol{U}}^t,\overline{\boldsymbol{V}}^t\right) - \nabla_{\boldsymbol{u}}\boldsymbol{f}\left(\overline{\boldsymbol{U}}^t,\overline{\boldsymbol{V}}^t\right)\right)\left(\boldsymbol{W} - \frac{1}{m}\boldsymbol{1}_m\boldsymbol{1}_m^T\right). \quad (64)$$

Thus, there is

$$\left\|\boldsymbol{\Delta}_{\boldsymbol{u},\mathcal{S}^+}^{t+1}\right\|_\infty \le \rho\left\|\boldsymbol{\Delta}_{\boldsymbol{u},\mathcal{S}^+}^t\right\|_\infty + \rho\eta\max_i\left\|\boldsymbol{1}_{\mathcal{S}^+}\odot\left(\nabla_{\boldsymbol{u}}f_i\left(\boldsymbol{u}_i^t,\boldsymbol{v}_i^t\right) - \nabla_{\boldsymbol{u}}F\left(\overline{\boldsymbol{u}}^t,\overline{\boldsymbol{v}}^t\right)\right)\right\|_\infty$$

$$\le \rho\left\|\boldsymbol{\Delta}_{\boldsymbol{u},\mathcal{S}^+}^t\right\|_\infty + \rho\eta\max_i\left\|\boldsymbol{1}_{\mathcal{S}^+}\odot\Pi_{1,i}\right\|_\infty + \rho\eta\max_i\left\|\boldsymbol{1}_{\mathcal{S}^+}\odot\Pi_{2,i}\right\|_\infty$$

$$+ \rho\eta\max_i\left\|\boldsymbol{1}_{\mathcal{S}^+}\odot\left(\nabla_{\boldsymbol{u}}f_i\left(\overline{\boldsymbol{u}}^t,\overline{\boldsymbol{v}}^t\right) - \nabla_{\boldsymbol{u}}F\left(\overline{\boldsymbol{u}}^t,\overline{\boldsymbol{v}}^t\right)\right)\right\|_\infty$$

$$\le \rho\left\|\boldsymbol{\Delta}_{\boldsymbol{u},\mathcal{S}^+}^t\right\|_\infty + 4\rho\eta\left\|\overline{\boldsymbol{u}}_{\mathcal{S}^+}^t\right\|_\infty\left(\sqrt{s}\delta_{\max}\left(2\left\|\overline{\boldsymbol{v}}_{\mathcal{S}^-}^t\right\|_\infty\left\|\boldsymbol{\Delta}_{\boldsymbol{v},\mathcal{S}^-}^t\right\|_\infty + \left\|\boldsymbol{\Delta}_{\boldsymbol{v},\mathcal{S}^-}^t\right\|_\infty^2\right)\right.$$

$$+ 2d\left(2\left\|\overline{\boldsymbol{v}}_{\mathcal{S}^c}^t\right\|_\infty\left\|\boldsymbol{\Delta}_{\boldsymbol{v},\mathcal{S}^c}^t\right\|_\infty + \left\|\boldsymbol{\Delta}_{\boldsymbol{v},\mathcal{S}^c}^t\right\|_\infty^2 + 2\left\|\overline{\boldsymbol{v}}_{\mathcal{S}^+}^t\right\|_\infty\left\|\boldsymbol{\Delta}_{\boldsymbol{v},\mathcal{S}^+}^t\right\|_\infty + \left\|\boldsymbol{\Delta}_{\boldsymbol{v},\mathcal{S}^+}^t\right\|_\infty^2\right)$$

$$+ \left\|\boldsymbol{\Delta}_{\boldsymbol{u},\mathcal{S}^+}^t\right\|_\infty\left(\left\|\boldsymbol{\Delta}_{\boldsymbol{u},\mathcal{S}^+}^t\right\|_\infty + 2\left\|\overline{\boldsymbol{u}}_{\mathcal{S}^+}^t\right\|_\infty\right) + \sqrt{s}\delta_{\max}\left\|\boldsymbol{\Delta}_{\boldsymbol{u},\mathcal{S}^+}^t\right\|_\infty\left(2\left\|\overline{\boldsymbol{u}}_{\mathcal{S}^+}^t\right\|_\infty + \left\|\boldsymbol{\Delta}_{\boldsymbol{u},\mathcal{S}^+}^t\right\|_\infty\right)$$

$$+ 2d\left(2\left\|\overline{\boldsymbol{u}}_{\mathcal{S}^c}^t\right\|_\infty\left\|\boldsymbol{\Delta}_{\boldsymbol{u},\mathcal{S}^c}^t\right\|_\infty + \left\|\boldsymbol{\Delta}_{\boldsymbol{u},\mathcal{S}^c}^t\right\|_\infty^2 + 2\left\|\overline{\boldsymbol{u}}_{\mathcal{S}^-}^t\right\|_\infty\left\|\boldsymbol{\Delta}_{\boldsymbol{u},\mathcal{S}^-}^t\right\|_\infty + \left\|\boldsymbol{\Delta}_{\boldsymbol{u},\mathcal{S}^-}^t\right\|_\infty^2\right)\right)$$

$$+ 4\rho\eta\left\|\boldsymbol{\Delta}_{\boldsymbol{u},\mathcal{S}^+}^t\right\|_\infty\cdot\left(\left\|\left(\overline{\boldsymbol{u}}_{\mathcal{S}^+}^t\right)^2 - \boldsymbol{w}_{\mathcal{S}^+}^\star\right\|_\infty + 2\left\|\overline{\boldsymbol{u}}_{\mathcal{S}^+}^t\right\|_\infty\left\|\boldsymbol{\Delta}_{\boldsymbol{u},\mathcal{S}^+}^t\right\|_\infty + \left\|\boldsymbol{\Delta}_{\boldsymbol{u},\mathcal{S}^+}^t\right\|_\infty^2\right.$$

$$+ \sqrt{s}\delta_{\max}\left(\left\|\left(\overline{\boldsymbol{u}}_{\mathcal{S}^+}^t\right)^2 - \boldsymbol{w}_{\mathcal{S}^+}^\star\right\|_\infty + \left\|\left(\overline{\boldsymbol{v}}_{\mathcal{S}^-}^t\right)^2 - \boldsymbol{w}_{\mathcal{S}^-}^\star\right\|_\infty + 2\left\|\overline{\boldsymbol{u}}_{\mathcal{S}^+}^t\right\|_\infty\left\|\boldsymbol{\Delta}_{\boldsymbol{u},\mathcal{S}^+}^t\right\|_\infty\right.$$

$$\left.+ \left\|\boldsymbol{\Delta}_{\boldsymbol{u},\mathcal{S}^+}^t\right\|_\infty^2\right) + 2d\left(\left(\left\|\overline{\boldsymbol{u}}_{\mathcal{S}^c}^t\right\|_\infty + \left\|\boldsymbol{\Delta}_{\boldsymbol{u},\mathcal{S}^c}^t\right\|_\infty\right)^2 + \left(\left\|\overline{\boldsymbol{v}}_{\mathcal{S}^c}^t\right\|_\infty + \left\|\boldsymbol{\Delta}_{\boldsymbol{v},\mathcal{S}^c}^t\right\|_\infty\right)^2\right.$$

$$\left.+ \left(\left\|\overline{\boldsymbol{u}}_{\mathcal{S}^-}^t\right\|_\infty + \left\|\boldsymbol{\Delta}_{\boldsymbol{u},\mathcal{S}^-}^t\right\|_\infty\right)^2 + \left(\left\|\overline{\boldsymbol{v}}_{\mathcal{S}^+}^t\right\|_\infty + \left\|\boldsymbol{\Delta}_{\boldsymbol{v},\mathcal{S}^+}^t\right\|_\infty\right)^2\right) + \max_i\left\|\frac{\boldsymbol{X}_i^T\boldsymbol{\xi}_i}{n}\right\|_\infty\right)$$

$$+ 4\rho\eta\left\|\overline{\boldsymbol{u}}_{\mathcal{S}^+}^t\right\|_\infty\cdot\left(\sqrt{s}\delta_{\max}\left(\left\|\left(\overline{\boldsymbol{u}}_{\mathcal{S}^+}^t\right)^2 - \boldsymbol{w}_{\mathcal{S}^+}^\star\right\|_\infty + \left\|\left(\overline{\boldsymbol{v}}_{\mathcal{S}^-}^t\right)^2 - \boldsymbol{w}_{\mathcal{S}^-}^\star\right\|_\infty\right)\right.$$

$$\left.+ \max_i\left\|\frac{\boldsymbol{X}_i^T\boldsymbol{\xi}_i}{n}\right\|_\infty + \left\|\frac{\boldsymbol{X}^T\boldsymbol{\xi}}{N}\right\|_\infty\right), \quad (65)$$

where the second inequality is due to (64) and the last inequality is substituting into the formula $\Pi_{1,i}, \Pi_{2,i}$ in (59) and gradient difference at averaged pair. Merging terms involved $\left\|\overline{\boldsymbol{u}}_{\mathcal{S}^+}^t\right\|_\infty$ and $\left\|\boldsymbol{\Delta}_{\boldsymbol{u},\mathcal{S}^+}^t\right\|_\infty$ separately would obtain the result in (60). Performing analogous proof would yield results on negative support $\mathcal{S}^-$ in (49) and non-support $\mathcal{S}^c$ in (50). □

The following proposition describes the dynamics of $\boldsymbol{u}^{t,i}, \boldsymbol{v}^{t,i}$ by conducting inductive hypothesis for both $\boldsymbol{u}^{t,i}$ and $\boldsymbol{v}^{t,i}$, which is different with centralized setting [36]. Because the complicated consensus errors prevent transferring the proof of non-negative case to general case trivially.

**Proposition 3.** *This proposition inherits Proposition 2 for the general case with considering dynamic of $\boldsymbol{v}_i^t$ for learning negative part signal $\boldsymbol{w}_{\mathcal{S}^-}^\star$. Recall the definitions of $T, T_k, \overline{T}_k, B_k, K, \beta$ in Proposition 2, the following statements hold in each induction step.*

- *(a)* $\forall t$ *that* $\overline{T}_{k-1} \le t < \overline{T}_k$ *with* $\forall k \in [K]$, *there is* $\max\left\{\left\|\left(\overline{\boldsymbol{u}}_{\mathcal{S}^+}^t\right)^2 - \boldsymbol{w}_{\mathcal{S}^+}^\star\right\|_\infty, \left\|\left(\overline{\boldsymbol{v}}_{\mathcal{S}^-}^t\right)^2 - \boldsymbol{w}_{\mathcal{S}^-}^\star\right\|_\infty\right\} \le \frac{w_{\max}^\star}{2^k}$.

- *(b) For $\forall k \in [K]$, there is* $\max\left\{\left\|\left(\overline{\boldsymbol{u}}_{\mathcal{S}^+}^{\overline{T}_{k-1}}\right)^2 - \boldsymbol{w}_{\mathcal{S}^+}^\star\right\|_\infty, \left\|\left(\overline{\boldsymbol{v}}_{\mathcal{S}^-}^{\overline{T}_{k-1}}\right)^2 - \boldsymbol{w}_{\mathcal{S}^-}^\star\right\|_\infty\right\} \le \frac{w_{\max}^\star}{2^k}.$

- *(c) $\forall t$ that $\overline{T}_{k-1} \le t < \overline{T}_k$ with $\forall k \in [K]$, there are* $\max_i\left\|\boldsymbol{\Delta}_{\boldsymbol{u},\mathcal{S}^+}^{t,i}\right\|_\infty \le 4\beta\rho^{\frac{3}{4}}\eta\left\|\overline{\boldsymbol{u}}_{\mathcal{S}^+}^t\right\|_\infty B_k$
  *and* $\max_i\left\|\boldsymbol{\Delta}_{\boldsymbol{v},\mathcal{S}^-}^{t,i}\right\|_\infty \le 4\beta\rho^{\frac{3}{4}}\eta\left\|\overline{\boldsymbol{v}}_{\mathcal{S}^-}^t\right\|_\infty B_k.$

- *(d) For any $\overline{T}_{k-1} \le t < \overline{T}_k$ that $\forall k \in [K]$, there are* $\max_i\left\|\boldsymbol{\Delta}_{\boldsymbol{u},\mathcal{S}^c}^{t,i}\right\|_\infty \le 4\beta\rho^{\frac{3}{4}}\eta\left\|\overline{\boldsymbol{u}}_{\mathcal{S}^c}^t\right\|_\infty B_k$
  *and* $\max_i\left\|\boldsymbol{\Delta}_{\boldsymbol{v},\mathcal{S}^c}^{t,i}\right\|_\infty \le 4\beta\rho^{\frac{3}{4}}\eta\left\|\overline{\boldsymbol{v}}_{\mathcal{S}^c}^t\right\|_\infty B_k.$

- *(e) For any $\overline{T}_{k-1} \le t < \overline{T}_k$ that $\forall k \in [K]$, there are* $\max_i\left\|\boldsymbol{\Delta}_{\boldsymbol{u},\mathcal{S}^-}^{t,i}\right\|_\infty \le 4\beta\rho^{\frac{3}{4}}\eta\left\|\overline{\boldsymbol{u}}_{\mathcal{S}^-}^t\right\|_\infty B_k$
  *and* $\max_i\left\|\boldsymbol{\Delta}_{\boldsymbol{v},\mathcal{S}^+}^{t,i}\right\|_\infty \le 4\beta\rho^{\frac{3}{4}}\eta\left\|\overline{\boldsymbol{v}}_{\mathcal{S}^+}^t\right\|_\infty B_k.$

- *(f) $\forall t$ that $\overline{T}_{k-1} \le t < \overline{T}_k$ with $k \in [K]$, there are refined element-wise bounds for consensus errors as* $|\boldsymbol{\Delta}_{\boldsymbol{u},j}^t| \le 4\beta\rho^{\frac{3}{4}}\eta|\overline{u}_j|B_k$ *and* $|\boldsymbol{\Delta}_{\boldsymbol{v},j}^t| \le 4\beta\rho^{\frac{3}{4}}\eta|\overline{v}_j|B_k, \forall j \in \mathcal{S}.$

- *(g) For $\forall j \in \mathcal{S}^+$ and $k \in [K]$, there is* $\alpha^3 \le \left(\overline{\boldsymbol{u}}_j^{\overline{T}_{k-1}}\right)^2 \le w_j^\star + 4B_k$. *For $\forall j \in \mathcal{S}^-$ and $k \in [K]$,*
  *there is* $\alpha^3 \le \left(\overline{\boldsymbol{v}}_j^{\overline{T}_{k-1}}\right)^2 \le w_j^\star + 4B_k.$

- *(h) $\forall t \le \mathcal{O}\left(\frac{1}{\eta\zeta}\log\frac{1}{\alpha}\right)$,* $\max\left\{\left\|\overline{\boldsymbol{u}}_{\mathcal{S}^c}^t\right\|_\infty, \left\|\overline{\boldsymbol{v}}_{\mathcal{S}^c}^t\right\|_\infty\right\} \le \sqrt{\alpha}.$

- *(i) $\forall t \le \mathcal{O}\left(\frac{1}{\eta\zeta}\log\frac{1}{\alpha}\right)$ and $\forall j \in \mathcal{S}$, there is* $\overline{u}_j^t\overline{v}_j^t \le \alpha^{\frac{3}{2}}.$

- *(j) $\forall t \le \mathcal{O}\left(\frac{1}{\eta\zeta}\log\frac{1}{\alpha}\right)$, it has* $\max\left\{\left\|\overline{\boldsymbol{u}}_{\mathcal{S}^-}^t\right\|_\infty, \left\|\overline{\boldsymbol{v}}_{\mathcal{S}^+}^t\right\|_\infty\right\} \le \sqrt{\alpha}.$

*Proof.* **Proof idea:** The key difference in the proof between centralized and decentralized settings lies in the fact that we cannot directly transfer the proof from the simpler non-negative $\boldsymbol{w}^\star$ to the general $\boldsymbol{w}^\star$. This is because the error terms induced in (48), (49),(50) by the decentralized network in Lemma 7 prevent obtaining the results in inductions (i), (j) without induction steps as Lemma B.16 in [36], which allows to apply proof of non-negative case to the general case directly in the centralized case. Therefore, we have to conduct the comprehensive induction process for the general $\boldsymbol{w}^\star$, which can ensure that the magnitudes of $\overline{\boldsymbol{u}}^t$ on the negative support $\mathcal{S}^-$ and $\overline{\boldsymbol{v}}^t$ on the positive support $\mathcal{S}^+$ remain small up to the early stopping iteration steps. The inductions (a)-(h) show that the averaged signal $\overline{\boldsymbol{u}}^t$ on the positive support $\mathcal{S}^+$ and the averaged signal $\overline{\boldsymbol{v}}^t$ on the negative support $\mathcal{S}^-$ exhibit dynamics similar to those in non-negative case, as outlined in Proposition 2.

*Base case:* As the initialization $\boldsymbol{u}^{0,i} = \boldsymbol{v}^{0,i} = \alpha\boldsymbol{1}_d, \forall i \in [m]$. Due to the condition on $\alpha$, the base case is true.

*Induction Step*: Under the assumption that all above induction hypotheses hold until some $0 \le k \le K - 1$, we should prove they still should hold at $k + 1$-th induction.

(a) We should prove that $\forall t \in \{\overline{T}_{k-1}, \cdots, \overline{T}_k - 1\}$, the condition $\left\|\left(\overline{\boldsymbol{u}}_{\mathcal{S}^+}^t\right)^2 - \boldsymbol{w}_{\mathcal{S}^+}^\star\right\|_\infty \le \frac{w_{\max}^\star}{2^k}$ and $\left\|\left(\overline{\boldsymbol{v}}_{\mathcal{S}^-}^t\right)^2 - \boldsymbol{w}_{\mathcal{S}^-}^\star\right\|_\infty \le \frac{w_{\max}^\star}{2^k}$ still holds. If the condition is true for $t$-th iteration, then based on claims (c),(d), (e), (h) and (j) the $\|\boldsymbol{p}_u^t\|_\infty$ in (51) under current induction step could be bounded as

$$\|\boldsymbol{p}_u^t\|_\infty \le 8\sqrt{s}\delta_{\max}\beta\rho^{\frac{3}{4}}B_k\eta\left(\left\|\overline{\boldsymbol{u}}_{\mathcal{S}^+}^t\right\|_\infty^2 + \left\|\overline{\boldsymbol{v}}_{\mathcal{S}^-}^t\right\|_\infty^2\right) + \sqrt{s}\delta_{\max}\left(4\beta\rho^{\frac{3}{4}}B_k\eta\left\|\overline{\boldsymbol{v}}_{\mathcal{S}^-}^t\right\|_\infty\right)^2$$
$$+ \left(\sqrt{s}\delta_{\max} + 3\right)\left(4\beta\rho^{\frac{3}{4}}\eta\left\|\overline{\boldsymbol{u}}_{\mathcal{S}^+}^t\right\|_\infty B_k\right)^2 + 2d\left(8\beta\rho^{\frac{3}{4}}\eta B_k + \left(4\beta\rho^{\frac{3}{4}}\eta B_k\right)^2\right)$$
$$\cdot\left(\left\|\overline{\boldsymbol{u}}_{\mathcal{S}^c}^t\right\|_\infty^2 + \left\|\overline{\boldsymbol{v}}_{\mathcal{S}^c}^t\right\|_\infty^2 + \left\|\overline{\boldsymbol{u}}_{\mathcal{S}^-}^t\right\|_\infty^2 + \left\|\overline{\boldsymbol{v}}_{\mathcal{S}^+}^t\right\|_\infty^2\right)$$
$$\le 32\sqrt{s}\delta_{\max}\beta\rho^{\frac{3}{4}}B_k\eta w_{\max}^\star + 32\sqrt{s}\delta_{\max}\beta^2\rho^{\frac{3}{2}}B_k^2\eta^2 w_{\max}^\star + 32\left(\sqrt{s}\delta_{\max} + 3\right)\beta^2\rho^{\frac{3}{2}}\eta^2 B_k^2 w_{\max}^\star$$

$$+ 2d \left( 8\beta\rho^{\frac{3}{4}}\eta B_k + 16\beta^2\rho^{\frac{3}{2}}\eta^2 B_k^2 \right) \left( \left\| \overline{\boldsymbol{u}}_{\mathcal{S}^c}^t \right\|_\infty^2 + \left\| \overline{\boldsymbol{v}}_{\mathcal{S}^c}^t \right\|_\infty^2 + \left\| \overline{\boldsymbol{u}}_{\mathcal{S}-}^t \right\|_\infty^2 + \left\| \overline{\boldsymbol{v}}_{\mathcal{S}+}^t \right\|_\infty^2 \right)$$

$$\leq \frac{3\left(\sqrt{s}\delta_{\max} + 3\right)\rho^{\frac{3}{4}} B_k}{16} + \frac{d\alpha\rho^{\frac{3}{4}}}{32}, \tag{66}$$

where the first inequality is due to definition of local RIP condition and Lemma 2 and Lemma 3, second inequality is due to hypothesis (a) holds at $t$-th iteration that $\left\| \overline{\boldsymbol{u}}_{\mathcal{S}+}^t \right\|_\infty^2 \leq 2w_{\max}^\star, \left\| \overline{\boldsymbol{v}}_{\mathcal{S}-}^t \right\|_\infty^2 \leq 2w_{\max}^\star$, the last inequality is due to definitions of $\beta, B_k$, step size condition and hypothesis that $\left\| \overline{\boldsymbol{u}}_{\mathcal{S}^c}^t \right\|_\infty, \left\| \overline{\boldsymbol{v}}_{\mathcal{S}^c}^t \right\|_\infty, \left\| \overline{\boldsymbol{u}}_{\mathcal{S}-}^t \right\|_\infty, \left\| \overline{\boldsymbol{v}}_{\mathcal{S}+}^t \right\|_\infty$ would keep below $\sqrt{\alpha}$ up to early stopping under hypothesis (h) and (j).

The element-wise bound for $\boldsymbol{q}_u^t$ in (52) under current induction step can be bounded as follows that $\forall j \in \mathcal{S}^+$, there is

$$|q_{u,j}^t| \leq 4\beta\rho^{\frac{3}{4}}\eta|\overline{u}_j^t|B_k \left( \sqrt{s}\delta_{\max} \left( \left\| \left(\overline{\boldsymbol{u}}_{\mathcal{S}+}^t\right)^2 - \boldsymbol{w}_{\mathcal{S}+}^\star \right\|_\infty + \left\| \left(\overline{\boldsymbol{v}}_{\mathcal{S}-}^t\right)^2 - \boldsymbol{w}_{\mathcal{S}-}^\star \right\|_\infty \right) \right.$$

$$\left. + \frac{3\left(\sqrt{s}\delta_{\max} + 3\right)\rho^{\frac{3}{4}} B_k}{16} + 8d\left( \left\| \overline{\boldsymbol{u}}_{\mathcal{S}^c}^t \right\|_\infty^2 + \left\| \overline{\boldsymbol{v}}_{\mathcal{S}^c}^t \right\|_\infty^2 + \left\| \overline{\boldsymbol{u}}_{\mathcal{S}-}^t \right\|_\infty^2 + \left\| \overline{\boldsymbol{v}}_{\mathcal{S}+}^t \right\|_\infty^2 \right) + \max_i \left\| \frac{\boldsymbol{X}_i^T\boldsymbol{\xi}_i}{n} \right\|_\infty \right)$$

$$\leq 4\beta\rho^{\frac{1}{2}}\eta|\overline{u}_j^t|B_k \left( \rho^{\frac{1}{4}}\sqrt{s}\delta_{\max} \left( \left\| \left(\overline{\boldsymbol{u}}_{\mathcal{S}+}^t\right)^2 - \boldsymbol{w}_{\mathcal{S}+}^\star \right\|_\infty + \left\| \left(\overline{\boldsymbol{v}}_{\mathcal{S}-}^t\right)^2 - \boldsymbol{w}_{\mathcal{S}-}^\star \right\|_\infty \right) \right.$$

$$\left. + \frac{3\left(\sqrt{s}\delta_{\max} + 3\right)\rho B_k}{16} + 32d\rho^{\frac{1}{4}}\alpha + \rho^{\frac{1}{4}}\max_i \left\| \frac{\boldsymbol{X}_i^T\boldsymbol{\xi}_i}{n} \right\|_\infty \right)$$

$$\leq \frac{\rho^{\frac{1}{2}} B_k|\overline{u}_j^t|}{32}, \tag{67}$$

where the first inequality is based on refined bound in hypothesis (c), result in (66) and inequality $\|\boldsymbol{x} + \boldsymbol{y}\|_\infty^2 \leq 2\|\boldsymbol{x}\|_\infty^2 + 2\|\boldsymbol{y}\|_\infty^2$ and step size condition that $4\beta\rho^{\frac{3}{4}}\eta B_k < 1$. The second inequality uses hypothesis (h) and (j). The last inequality is because of hypothesis (a), the step size condition, network connectivity condition, small initialization $\alpha$ and definition of $B_k$.

Then $\boldsymbol{q}_u^t$ could be reparameterized as $\boldsymbol{q}_u^t = \boldsymbol{r}_{q_u}^t \odot \overline{\boldsymbol{u}}_{\mathcal{S}+}^t$ where $\left\| \boldsymbol{r}_{q_u}^t \right\|_\infty \leq \frac{\sqrt{\rho}B_k}{32}, \forall t, \overline{T}_{k-1} \leq t < \overline{T}_k$. Then the perturbed recursion for $\overline{\boldsymbol{u}}^t$ on positive support $\mathcal{S}^+$ over decentralized network in (48) becomes

$$\left(\overline{\boldsymbol{u}}_{\mathcal{S}+}^{t+1}\right)^2 = \left(\overline{\boldsymbol{u}}_{\mathcal{S}+}^t\right)^2 \odot \left( \mathbf{1}_d - 4\eta \left( \left(\overline{\boldsymbol{u}}_{\mathcal{S}+}^t\right)^2 - \boldsymbol{w}_{\mathcal{S}+}^\star + \boldsymbol{E}_{2s}^t + \boldsymbol{E}_{3s}^t + \boldsymbol{p}_u^t + \boldsymbol{r}_{q_u}^t \right) \right)^2, \tag{68}$$

where the new perturbed error terms $\boldsymbol{E}_{2s}^t, \boldsymbol{E}_{2s}^t$ are defined as

$$\boldsymbol{E}_{2s}^t := \left( \frac{\boldsymbol{X}^T\boldsymbol{X}}{N} - \boldsymbol{I} \right) \left( \left(\overline{\boldsymbol{u}}_{\mathcal{S}+}^t\right)^2 - \boldsymbol{w}_{\mathcal{S}+}^\star - \left(\overline{\boldsymbol{v}}_{\mathcal{S}-}^t\right)^2 - \boldsymbol{w}_{\mathcal{S}-}^\star \right)$$

$$\boldsymbol{E}_{3s}^t := \frac{\boldsymbol{X}^T\boldsymbol{X}}{N} \left( \left(\overline{\boldsymbol{u}}_{\mathcal{S}^c}^t\right)^2 - \left(\overline{\boldsymbol{v}}_{\mathcal{S}+}\right)^2 + \left(\overline{\boldsymbol{u}}_{\mathcal{S}-}^t\right)^2 - \left(\overline{\boldsymbol{v}}_{\mathcal{S}^c}^t\right)^2 \right) - \frac{\boldsymbol{X}^T\boldsymbol{\xi}}{N}. \tag{69}$$

Then the total error terms in (68) can be bounded as follows

$$\left\| \boldsymbol{E}_{2s}^t \right\|_\infty + \left\| \boldsymbol{E}_{3s}^t \right\|_\infty + \left\| \boldsymbol{p}_u^t \right\|_\infty + \left\| \boldsymbol{r}_{q_u}^t \right\|_\infty \overset{(i)}{\leq} 2\sqrt{s}\delta\frac{w_{\max}^\star}{2^k} + 8d\alpha + \left\| \frac{\boldsymbol{X}^T\boldsymbol{\xi}}{N} \right\|_\infty$$

$$+ \frac{3\left(\sqrt{s}\delta_{\max} + 3\right)\rho^{\frac{3}{4}} B_k}{16} + \frac{d\alpha\rho^{\frac{3}{4}}}{32} + \frac{\rho^{\frac{1}{2}} B_k}{32}$$

$$\overset{(ii)}{\leq} C_\gamma\frac{w_{\max}^\star}{2^k} + C_b \cdot \frac{2}{C_b}\max\left\{ \left\| \frac{\boldsymbol{X}^T\boldsymbol{\xi}}{N} \right\|_\infty, \epsilon \right\} + \frac{B_k}{2}$$

$$\overset{(iii)}{\leq} C_\gamma\frac{w_{\max}^\star}{2^k} + C_b\zeta + \frac{B_k}{2}$$

$$\overset{(iv)}{\leq} \left(C_\gamma + 2C_b\right)\frac{w_{\max}^\star}{2^k} + \frac{B_k}{2}$$

$$\overset{(v)}{\leq} \frac{1}{2} \cdot \frac{1}{40} \frac{w^\star_{\max}}{2^k} + \frac{B_k}{2}$$

$$\overset{(vi)}{\leq} B_k, \tag{70}$$

where $(i)$ is because of Lemma 2 and Lemma 3 under the global RIP condition and induction hypothesis (a) and substituting the results in (66) and (67). $(ii)$ is due to condition on global RIP parameter that $\delta \leq \frac{C_\gamma}{2\sqrt{s}\left(\log\lceil \frac{w^\star_{\max}}{\zeta}\rceil + 1\right)}$, condition $\alpha \leq \frac{\zeta}{(3d+1)^2}$ and network connectivity condition $\rho \leq \frac{1}{36\left(\sqrt{s}\delta_{\max}+3\right)^{\frac{4}{3}}}$. $(iii)$ is due to definition of $\zeta$ and $(iv)$ is based on definition of $K$ that $\zeta \leq \frac{w^\star_{\max}}{2^{K-1}} \leq \frac{w^\star_{\max}}{2^{k-1}}$. $(v)$ is because definition of $B_k$ and $C_\gamma, C_b$ are determined later that are small enough that $C_\gamma + 2C_b \leq \frac{1}{80}$.

Based on iteration (68) and perturbation bound in (70), we would prove the $\left\|\left(\overline{u}^{t+1}_{\mathcal{S}^+}\right)^2 - w^\star_{\mathcal{S}^+}\right\|_\infty \leq \frac{w^\star_{\max}}{2^k}$ should be also true based on the proof of hypothesis (a) in Proposition 2 by replacing $\mathcal{S}$ with $\mathcal{S}^+$. We can obtain similar results for $\overline{v}^t_{\mathcal{S}^-}$. Combined two cases and induction on $t$, we have conclusion that $\forall t \geq \overline{T}_{k-1}$, there is

$$\max\left\{\left\|\left(\overline{u}^t_{\mathcal{S}^+}\right)^2 - w^\star_{\mathcal{S}^+}\right\|_\infty, \left\|\left(\overline{v}^t_{\mathcal{S}^-}\right)^2 - w^\star_{\mathcal{S}^-}\right\|_\infty\right\} \leq \frac{w^\star_{\max}}{2^k}. \tag{71}$$

Thus, we finish the proof of induction (a).

(b) Comparing the (68), (70) with (30), the proof of this hypothesis can follow the proof of hypothesis (b) in Proposition 2.

(c) $\forall t$ that $\overline{T}_{k-1} \leq t < t+1 < \overline{T}_k$, based on one step iteration in (60) and induction (a), there is

$$\left\|\Delta^{t+1}_{u,\mathcal{S}^+}\right\|_\infty \overset{(i)}{\leq} \rho \left\|\Delta^t_{u,\mathcal{S}^+}\right\|_\infty \cdot \left(1 + 4\eta\left(2\left(\sqrt{s}\delta_{\max}+1\right)w^\star_{\max} + 8d\left(\left\|\overline{u}^t_{\mathcal{S}^c}\right\|^2_\infty + \left\|\overline{v}^t_{\mathcal{S}^c}\right\|^2_\infty\right.\right.\right.$$
$$\left.\left.+ \left\|\overline{u}^t_{\mathcal{S}^-}\right\|^2_\infty + \left\|\overline{v}^t_{\mathcal{S}^+}\right\|^2_\infty\right) + 8\left(\sqrt{s}\delta_{\max}+1\right)\left\|\overline{u}^t_{\mathcal{S}^+}\right\|^2_\infty + \max_i\left\|\frac{X^T_i\xi_i}{n}\right\|_\infty\right)\right)$$
$$+ 4\rho\eta\left\|\overline{u}^t_{\mathcal{S}^+}\right\|_\infty \cdot \left(4\sqrt{s}\delta_{\max}\beta\rho^{\frac{3}{4}}\eta\left\|\overline{v}^t_{\mathcal{S}^-}\right\|_\infty B_k \cdot \left(2\left\|\overline{v}^t_{\mathcal{S}^-}\right\|_\infty + 4\beta\rho^{\frac{3}{4}}\eta\left\|\overline{v}^t_{\mathcal{S}^-}\right\|_\infty B_k\right)\right.$$
$$+ 4\sqrt{s}\delta_{\max}\frac{w^\star_{\max}}{2^k} + 8d\left(\left\|\overline{u}^t_{\mathcal{S}^c}\right\|^2_\infty + \left\|\overline{v}^t_{\mathcal{S}^c}\right\|^2_\infty + \left\|\overline{u}^t_{\mathcal{S}^-}\right\|^2_\infty + \left\|\overline{v}^t_{\mathcal{S}^+}\right\|^2_\infty\right)$$
$$\left.+ \max_i\left\|\frac{X^T_i\xi_i}{n}\right\|_\infty + \left\|\frac{X^T\xi}{N}\right\|_\infty\right)$$

$$\overset{(ii)}{\leq} \rho^{\frac{3}{4}}\left\|\Delta^t_{u,\mathcal{S}^+}\right\|_\infty \cdot \left(1 + 4\eta\left(18\rho^{\frac{1}{4}}w^\star_{\max}\left(\sqrt{s}\delta_{\max}+1\right) + 32\rho^{\frac{1}{4}}d\alpha\right.\right.$$
$$\left.\left.+ \rho^{\frac{1}{4}}\max_i\left\|\frac{X^T_i\xi_i}{n}\right\|_\infty\right)\right) + 4\rho^{\frac{3}{4}}\eta\left\|\overline{u}^t_{\mathcal{S}^+}\right\|_\infty \cdot \left(16\sqrt{s}\delta_{\max}\beta\rho\eta\left\|\overline{v}^t_{\mathcal{S}^-}\right\|^2_\infty B_k\right.$$
$$\left.+ 4\rho^{\frac{1}{4}}\sqrt{s}\delta_{\max}\frac{w^\star_{\max}}{2^k} + 32\rho^{\frac{1}{4}}d\alpha + 2\rho^{\frac{1}{4}}\max_i\left\|\frac{X^T_i\xi_i}{n}\right\|_\infty\right)$$

$$\overset{(iii)}{\leq} \rho^{\frac{3}{4}}\left\|\Delta^t_{u,\mathcal{S}^+}\right\|_\infty \cdot \left(1 + 80\eta w^\star_{\max}\right) + 4\rho^{\frac{3}{4}}\eta\left\|\overline{u}^t_{\mathcal{S}^+}\right\|_\infty \cdot \left(16\sqrt{s}\delta_{\max}\beta\rho\eta\left\|\overline{v}^t_{\mathcal{S}^-}\right\|^2_\infty B_k + \frac{3}{4}B_k\right)$$

$$\overset{(iv)}{\leq} 4\left(\frac{1+\sqrt{\rho}}{2}\right)\beta\rho^{\frac{3}{4}}\eta\left\|\overline{u}^t_{\mathcal{S}^+}\right\|_\infty B_k + 4\rho^{\frac{3}{4}}\eta\left\|\overline{u}^t_{\mathcal{S}^+}\right\|_\infty B_k$$

$$\overset{(v)}{\leq} \frac{4}{\left(1 - c_{10}\left(1-\sqrt{\rho}\right)\right)^2}\left(\left(\frac{1+\sqrt{\rho}}{2}\right)\beta + 1\right)\rho^{\frac{3}{4}}\eta\left\|\overline{u}^{t+1}_{\mathcal{S}^+}\right\|_\infty B_k, \tag{72}$$

where $(i)$ is due to the claim (a) and step size condition that $4\beta\rho^{\frac{3}{4}}\eta B_k \leq 1$ and $\delta \leq \delta_{\max}$. $(ii)$ is due to $4\beta\rho^{\frac{3}{4}}\eta B_k \leq 1$, condition that all $\left\|\overline{\boldsymbol{u}}_{\mathcal{S}^c}^t\right\|_\infty, \left\|\overline{\boldsymbol{v}}_{\mathcal{S}^c}^t\right\|_\infty, \left\|\overline{\boldsymbol{u}}_{\mathcal{S}-}^t\right\|_\infty, \left\|\overline{\boldsymbol{v}}_{\mathcal{S}+}^t\right\|_\infty$ are not larger than $\sqrt{\alpha}$ before early stopping in hypothesis (h), (j) and $\left\|\frac{\boldsymbol{X}^T\boldsymbol{\xi}}{N}\right\|_\infty \leq \max_i \left\|\frac{\boldsymbol{X}_i^T\boldsymbol{\xi}_i}{n}\right\|_\infty$.

The $(iii)$ is because $\rho^{\frac{1}{4}} \leq \min\left\{\frac{1}{\sqrt{s}\delta_{\max}+1}, \frac{\delta}{8\delta_{\max}}, \frac{\left\|\frac{\boldsymbol{X}^T\boldsymbol{\xi}}{N}\right\|_\infty}{8\max_i\left\|\frac{\boldsymbol{X}_i^T\boldsymbol{\xi}_i}{n}\right\|_\infty}\right\}$, $\alpha \leq \frac{\epsilon}{(12d+1)^2}$ and $\max\left\{\left\|\frac{\boldsymbol{X}^T\boldsymbol{\xi}}{N}\right\|_\infty, \epsilon\right\} \leq w^\star_{\max}$. The $(iv)$ is because in the current induction step hypothesis (c) holds in $t$-th iteration and step size condition that satisfies $\eta \leq \frac{1-\sqrt{\rho}}{160\rho^{\frac{1}{4}}w^\star_{\max}}$ such that $\rho^{\frac{1}{2}}\left(\rho^{\frac{1}{4}} + 80\rho^{\frac{1}{4}}\eta w^\star_{\max}\right) \leq (1-(1-\rho^{\frac{1}{2}}))(1+\frac{1-\sqrt{\rho}}{2}) \leq \frac{1+\sqrt{\rho}}{2}$ and $16\sqrt{s}\delta_{\max}\beta\rho\eta\left\|\overline{\boldsymbol{v}}_{\mathcal{S}-}^t\right\|_\infty^2 \leq \frac{B_k}{4}$ where $\left\|\overline{\boldsymbol{v}}_{\mathcal{S}-}^t\right\|_\infty^2 \leq 2w^\star_{\max}$ is according to claim (a). The last inequality $(v)$ is based on the (68) and step size condition $\eta \leq \frac{c_{10}(1-\sqrt{\rho})}{w^\star_{\max}}$ such that $4\eta\left(\left\|\left(\overline{\boldsymbol{u}}_{\mathcal{S}+}^t\right)^2 - \boldsymbol{w}^\star_{\mathcal{S}+}\right\|_\infty + B_k\right) \leq 1$ based on hypothesis (a). Comparing the formula of (72) with (34), we can follow the poof of hypothesis (c) in Proposition 2 to finish the proof.

(d) The proof is similar to that of hypothesis (c). $\forall t$ that $\overline{T}_{k-1} \leq t < t+1 \leq \overline{T}_k$, according to one step iteration in (49) and hypothesis (a), we have

$$\left\|\boldsymbol{\Delta}_{\boldsymbol{u},\mathcal{S}^c}^{t+1}\right\|_\infty \leq \rho\left\|\boldsymbol{\Delta}_{\boldsymbol{u},\mathcal{S}^c}^t\right\|_\infty \cdot \left(1 + 4\eta\left(\sqrt{s}\delta_{\max}\left(2w^\star_{\max} + 4\beta\rho^{\frac{3}{4}}\eta\left\|\overline{\boldsymbol{u}}_{\mathcal{S}+}^t\right\|_\infty B_k\left(2\left\|\overline{\boldsymbol{u}}_{\mathcal{S}+}^t\right\|_\infty\right.\right.\right.\right.$$
$$\left.+4\beta\rho^{\frac{3}{4}}\eta\left\|\overline{\boldsymbol{u}}_{\mathcal{S}+}^t\right\|_\infty B_k\right) + 4\beta\rho^{\frac{3}{4}}\eta\left\|\overline{\boldsymbol{v}}_{\mathcal{S}-}^t\right\|_\infty B_k\left(2\left\|\overline{\boldsymbol{v}}_{\mathcal{S}-}^t\right\|_\infty + 4\beta\rho^{\frac{3}{4}}\eta\left\|\overline{\boldsymbol{v}}_{\mathcal{S}-}^t\right\|_\infty B_k\right)\right)$$
$$\left.+64d\alpha + \max_i\left\|\frac{\boldsymbol{X}_i^T\boldsymbol{\xi}_i}{n}\right\|_\infty\right)\right) + 4\rho\eta\left\|\overline{\boldsymbol{u}}_{\mathcal{S}^c}^t\right\|_\infty \cdot \left(\sqrt{s}\delta_{\max}\left(4\beta\rho^{\frac{3}{4}}\eta\left\|\overline{\boldsymbol{u}}_{\mathcal{S}+}^t\right\|_\infty B_k\right.\right.$$
$$\cdot\left(2\left\|\overline{\boldsymbol{u}}_{\mathcal{S}+}^t\right\|_\infty + 4\beta\rho^{\frac{3}{4}}\eta\left\|\overline{\boldsymbol{u}}_{\mathcal{S}+}^t\right\|_\infty B_k\right) + 4\beta\rho^{\frac{3}{4}}\eta\left\|\overline{\boldsymbol{v}}_{\mathcal{S}-}^t\right\|_\infty B_k \cdot \left(2\left\|\overline{\boldsymbol{v}}_{\mathcal{S}-}^t\right\|_\infty\right.\right.$$
$$\left.\left.\cdot +4\beta\rho^{\frac{3}{4}}\eta\left\|\overline{\boldsymbol{v}}_{\mathcal{S}-}^t\right\|_\infty B_k\right)\right) + 4\sqrt{s}\delta_{\max}\frac{w^\star_{\max}}{2^k} + 32d\alpha + 2\max_i\left\|\frac{\boldsymbol{X}_i^T\boldsymbol{\xi}_i}{n}\right\|_\infty\right)$$
$$\leq \rho^{\frac{3}{4}}\left\|\boldsymbol{\Delta}_{\boldsymbol{u},\mathcal{S}^c}^t\right\|_\infty \cdot \left(1 + 4\eta\left(\rho^{\frac{1}{4}}\sqrt{s}\delta_{\max}\left(2w^\star_{\max} + 16\beta\rho^{\frac{3}{4}}\eta B_k\left(\left\|\overline{\boldsymbol{u}}_{\mathcal{S}+}^t\right\|_\infty^2 + \left\|\overline{\boldsymbol{v}}_{\mathcal{S}-}^t\right\|_\infty^2\right)\right)\right.\right.$$
$$\left.\left.+\frac{4}{9}\zeta + \rho^{\frac{1}{4}}\left\|\frac{\boldsymbol{X}_i^T\boldsymbol{\xi}_i}{n}\right\|_\infty\right)\right) + 4\rho^{\frac{3}{4}}\eta\left\|\overline{\boldsymbol{u}}_{\mathcal{S}^c}^t\right\|_\infty \cdot \left(16\sqrt{s}\delta_{\max}\beta\rho\eta B_k\left(\left\|\overline{\boldsymbol{u}}_{\mathcal{S}+}^t\right\|_\infty^2 + \left\|\overline{\boldsymbol{v}}_{\mathcal{S}-}^t\right\|_\infty^2\right)\right.$$
$$\left.+ 4\rho^{\frac{1}{4}}\sqrt{s}\delta_{\max}\frac{w^\star_{\max}}{2^k} + \frac{\zeta}{4} + 2\rho^{\frac{1}{4}}\left\|\frac{\boldsymbol{X}_i^T\boldsymbol{\xi}_i}{N}\right\|_\infty\right)$$
$$\leq 4\left(\frac{1+\sqrt{\rho}}{2}\right)\beta\rho^{\frac{3}{4}}\eta\left\|\overline{\boldsymbol{u}}_{\mathcal{S}^c}^t\right\|_\infty B_k + 4\rho^{\frac{3}{4}}\eta\left\|\overline{\boldsymbol{u}}_{\mathcal{S}^c}^t\right\|_\infty B_k$$
$$\leq \frac{4}{\left(1 - c_{10}\left(1-\sqrt{\rho}\right)\right)^2}\left(\left(\frac{1+\sqrt{\rho}}{2}\right)\beta + 1\right)\rho^{\frac{3}{4}}\eta\left\|\overline{\boldsymbol{u}}_{\mathcal{S}^c}^{t+1}\right\|_\infty B_k, \tag{73}$$

where the first two inequalities have the same reasons in (72). The third inequality is due to $16\beta\rho^{\frac{3}{4}}B_k\left(\left\|\overline{\boldsymbol{u}}_{\mathcal{S}+}^t\right\|_\infty^2 + \left\|\overline{\boldsymbol{v}}_{\mathcal{S}-}^t\right\|_\infty^2\right) \leq \frac{8}{5}w^\star_{\max}$ and $\rho^{\frac{1}{4}}\sqrt{s}\delta_{\max} \leq \frac{\sqrt{s}\delta}{8} \leq \frac{1}{640}$ due to upper bound of global RIP condition in Proposition 1 with $C_\gamma \leq \frac{1}{80}$. The last inequality is due to (79). Then we can use analogized proof for hypothesis (c) to finish the proof.

(e) The proof is similar to hypothesis (d). $\forall t$ that $\overline{T}_{k-1} \le t < t+1 \le \overline{T}_k$, based on one step iteration in (50) and hypothesis (a), there is

$$
\left\| \boldsymbol{\Delta}_{\boldsymbol{u},\mathcal{S}^-}^{t+1} \right\|_\infty \le \rho \left\| \boldsymbol{\Delta}_{\boldsymbol{u},\mathcal{S}^-}^{t} \right\|_\infty \cdot \left( 1 + 4\rho\eta \left( (\sqrt{s}\delta_{\max} + 1) \left( 2w_{\max}^\star + 4\beta\rho^{\frac{3}{4}}\eta \left\| \overline{\boldsymbol{u}}_{\mathcal{S}^+}^t \right\|_\infty B_k \left( 2\left\| \overline{\boldsymbol{u}}_{\mathcal{S}^+}^t \right\|_\infty \right.\right.\right.\right.
$$
$$
\left. +4\beta\rho^{\frac{3}{4}}\eta \left\| \overline{\boldsymbol{u}}_{\mathcal{S}^+}^t \right\|_\infty B_k \right) + 4\beta\rho^{\frac{3}{4}}\eta \left\| \overline{\boldsymbol{v}}_{\mathcal{S}^-}^t \right\|_\infty B_k \left( 2\left\| \overline{\boldsymbol{v}}_{\mathcal{S}^-}^t \right\|_\infty + 4\beta\rho^{\frac{3}{4}}\eta \left\| \overline{\boldsymbol{v}}_{\mathcal{S}^-}^t \right\|_\infty B_k \right) \right)
$$
$$
\left. + 64d\alpha + \max_i \left\| \frac{\boldsymbol{X}_i^T \boldsymbol{\xi}_i}{n} \right\|_\infty \right) \right) + 4\rho\eta \left\| \overline{\boldsymbol{u}}_{\mathcal{S}^-}^t \right\|_\infty \cdot \left( (\sqrt{s}\delta_{\max} + 1) \left( 4\beta\rho^{\frac{3}{4}}\eta \left\| \overline{\boldsymbol{u}}_{\mathcal{S}^+}^t \right\|_\infty B_k \right.\right.
$$
$$
\cdot \left( 2\left\| \overline{\boldsymbol{u}}_{\mathcal{S}^+}^t \right\|_\infty + 4\beta\rho^{\frac{3}{4}}\eta \left\| \overline{\boldsymbol{u}}_{\mathcal{S}^+}^t \right\|_\infty B_k \right) + 4\beta\rho^{\frac{3}{4}}\eta \left\| \overline{\boldsymbol{v}}_{\mathcal{S}^-}^t \right\|_\infty B_k \left( 2\left\| \overline{\boldsymbol{v}}_{\mathcal{S}^-}^t \right\|_\infty \right.\right.
$$
$$
\left.\left.\left. +4\beta\rho^{\frac{3}{4}}\eta \left\| \overline{\boldsymbol{v}}_{\mathcal{S}^-}^t \right\|_\infty B_k \right) \right) + 2\sqrt{s} \left( \delta + \delta_{\max} \right) \frac{w_{\max}^\star}{2^k} + 32d\alpha + 2\max_i \left\| \frac{\boldsymbol{X}_i^T \boldsymbol{\xi}_i}{n} \right\|_\infty \right)
$$
$$
\le \frac{4}{\left( 1 - c_{10} \left( 1 - \sqrt{\rho} \right) \right)^2} \left( \left( \frac{1 + \sqrt{\rho}}{2} \right) \beta + 1 \right) \rho^{\frac{3}{4}}\eta \left\| \overline{\boldsymbol{u}}_{\mathcal{S}^-}^{t+1} \right\|_\infty B_k, \tag{74}
$$

where the last inequality and the left proof have analogized derivations in proof of hypothesis (d).

(f) To absorb the perturbed error $\boldsymbol{q}_u^t$ in (48), we need to prove the fine-grained bound for consensus error in hypotheses (c), (d), (d). The idea is that we focus on the $\forall j \in \mathcal{S}^+$, then the upper bound on the consensus error on $j$-th entry is

$$
|\boldsymbol{\Delta}_{\boldsymbol{u},j}^{t+1}| \le \rho|\boldsymbol{\Delta}_{\boldsymbol{u},j}^{t}| \cdot \left( 1 + 4\eta \left( \sqrt{s}\delta_{\max} \left( \left\| \left(\overline{\boldsymbol{u}}_{\mathcal{S}^+}^t\right)^2 - \boldsymbol{w}_{\mathcal{S}^+}^\star \right\|_\infty + \left\| -\left(\overline{\boldsymbol{v}}_{\mathcal{S}^-}^t\right)^2 - \boldsymbol{w}_{\mathcal{S}^-}^\star \right\|_\infty \right.\right.\right.
$$
$$
\left. +2\left\| \overline{\boldsymbol{u}}_{\mathcal{S}^+}^t \right\|_\infty \left\| \boldsymbol{\Delta}_{\boldsymbol{u},\mathcal{S}^+}^t \right\|_\infty + \left\| \boldsymbol{\Delta}_{\boldsymbol{u},\mathcal{S}^+}^t \right\|_\infty^2 + 2\left\| \overline{\boldsymbol{v}}_{\mathcal{S}^-}^t \right\|_\infty \left\| \boldsymbol{\Delta}_{\boldsymbol{v},\mathcal{S}^-}^t \right\|_\infty + \left\| \boldsymbol{\Delta}_{\boldsymbol{v},\mathcal{S}^-}^t \right\|_\infty^2 \right)
$$
$$
\left.\left. +|(\overline{u}_j^t)^2 - w_j^\star| + 2(\overline{u}_j^t)^2 + 3|\overline{u}_j^t| \left\| \boldsymbol{\Delta}_{\boldsymbol{u},j}^t \right\|_\infty + \left\| \boldsymbol{\Delta}_{\boldsymbol{u},j}^t \right\|_\infty^2 + 24d\alpha + \max_i \left\| \frac{\boldsymbol{X}_i^T \boldsymbol{\xi}_i}{n} \right\|_\infty \right) \right)
$$
$$
+ 4\rho\eta|\overline{u}_j^t| \cdot \left( \sqrt{s}\delta_{\max} \left( 2\left\| \overline{\boldsymbol{u}}_{\mathcal{S}^+}^t \right\|_\infty \left\| \boldsymbol{\Delta}_{\boldsymbol{u},\mathcal{S}^+}^t \right\|_\infty + \left\| \boldsymbol{\Delta}_{\boldsymbol{u},\mathcal{S}^+}^t \right\|_\infty^2 + 2\left\| \overline{\boldsymbol{v}}_{\mathcal{S}^-}^t \right\|_\infty \left\| \boldsymbol{\Delta}_{\boldsymbol{v},\mathcal{S}^-}^t \right\|_\infty \right.\right.
$$
$$
\left. + \left\| \boldsymbol{\Delta}_{\boldsymbol{v},\mathcal{S}^-}^t \right\|_\infty^2 \right) + \sqrt{s} \left( \delta_{\max} + \delta \right) \left( \left\| \left(\overline{\boldsymbol{u}}_{\mathcal{S}^+}^t\right)^2 - \boldsymbol{w}_{\mathcal{S}^+}^\star \right\|_\infty + \left\| -\left(\overline{\boldsymbol{v}}_{\mathcal{S}^-}^t\right)^2 - \boldsymbol{w}_{\mathcal{S}^-}^\star \right\|_\infty \right)
$$
$$
\left. +40d\alpha + \max_i \left\| \frac{\boldsymbol{X}_i^T \boldsymbol{\xi}_i}{n} \right\|_\infty + \left\| \frac{\boldsymbol{X}^T \boldsymbol{\xi}}{N} \right\|_\infty \right)
$$
$$
\le \frac{4}{\left( 1 - c_{10} \left( 1 - \sqrt{\rho} \right) \right)^2} \left( \left( \frac{1 + \sqrt{\rho}}{2} \right) \beta + 1 \right) \rho^{\frac{3}{4}}\eta B_k|\overline{u}_j^{t+1}|, \tag{75}
$$

where the last inequality is due to substituting the crude bound and analogous derivation in (72). Following the proof of crude bound would finish the proof. The proof for $\forall j \in \mathcal{S}^-$ and $\forall j \in \mathcal{S}^c$ can combine this proof with proofs of hypotheses (e) and (d), respectively. The fine-grained bound for $v^t$ can be proved by using analogous proofs for $\boldsymbol{u}^t$.

(g) The proof can follow the proof of hypothesis (e) in Proposition 2 by replacing the $\mathcal{S}$ with $\mathcal{S}^+$. The proof of $\overline{\boldsymbol{v}}_{\mathcal{S}^-}^t$ is analogous to $\overline{\boldsymbol{u}}_{\mathcal{S}^+}^t$.

(h) $\forall t$ that $\overline{T}_{k-1} \le t < t+1 < \overline{T}_k$. First, we bound the perturbation error $\left\| \boldsymbol{g}_u^t \right\|_\infty, \left\| \boldsymbol{f}_u^t \right\|_\infty$ in (50) induced from decentralized network

$$
\left\| \boldsymbol{g}_u^t \right\|_\infty \le \left\| \boldsymbol{p}_u^t \right\|_\infty + 3\sqrt{s}\delta_{\max} \left( 4\beta\rho^{\frac{3}{4}}\eta \left\| \overline{\boldsymbol{u}}_{\mathcal{S}^+}^t \right\|_\infty B_k \right)^2
$$
$$
\le \frac{3 \left( \sqrt{s}\delta_{\max} + 3 \right) \rho^{\frac{3}{4}} B_k}{16} + \frac{d\alpha\rho^{\frac{3}{4}}}{32} + 96\sqrt{s}\delta_{\max}\rho^{\frac{3}{2}}\beta^2\eta^2 w_{\max}^\star B_k^2
$$

$$\leq \frac{\rho^{\frac{3}{4}} B_k \left(150 \left(\sqrt{s}\delta_{\max} + 3\right) + 3\sqrt{\rho}\right)}{800} + \frac{d\alpha\rho^{\frac{3}{4}}}{32}, \tag{76}$$

where the second inequality is substituting the upper bound in (66) and $\left\|\overline{\boldsymbol{u}}_{\mathcal{S}+}^t\right\|_\infty \leq 2w_{\max}^\star$ due to hypothesis (a). $\forall j \in \mathcal{S}^c$, there is

$$
\begin{aligned}
|f_j^t| &\leq 4\beta\rho^{\frac{3}{4}}\eta|\overline{u}_j^t|B_k \cdot \left(\sqrt{s}\delta_{\max} \cdot \left(80B_k + 16\beta\rho^{\frac{3}{4}}\eta B_k \left(\left\|\overline{\boldsymbol{u}}_{\mathcal{S}+}^t\right\|_\infty^2 + \left\|\overline{\boldsymbol{v}}_{\mathcal{S}-}^t\right\|_\infty^2\right)\right)\right. \\
&\quad \left. + 32d\alpha + \max_i \left\|\frac{\boldsymbol{X}_i^T\boldsymbol{\xi}_i}{n}\right\|_\infty\right) \\
&\leq \frac{\sqrt{\rho}B_k|\overline{u}_j^t|}{32}, \tag{77}
\end{aligned}
$$

where the first inequality is due to hypotheses (a), (c)-(f), (h), (g) and step size condition. The second inequality is due to the definition of $B_k$, the condition on $\rho$, and small initialization for $\alpha$.

Then $\boldsymbol{f}_u^t$ could be reparameterized as $\boldsymbol{f}_u^t = \boldsymbol{r}_{f_u}^t \odot \overline{\boldsymbol{u}}_{\mathcal{S}^c}^t$, where $\left\|\boldsymbol{r}_{f_u}^t\right\|_\infty \leq \frac{\sqrt{\rho}B_k}{32}, \forall t$ that $\overline{T}_{k-1} \leq t < \overline{T}_k$. Then the perturbed recursion for $\overline{\boldsymbol{u}}^t$ on non-support $\mathcal{S}^c$ over decentralized network in (50) becomes

$$\left(\overline{\boldsymbol{u}}_{\mathcal{S}^c}^{t+1}\right)^2 = \left(\overline{\boldsymbol{u}}_{\mathcal{S}^c}^t\right)^2 \odot \left(\boldsymbol{1}_d - 4\eta\left(\boldsymbol{E}_{2s}^t + \boldsymbol{E}_{3s}^t + \boldsymbol{g}_u^t + \boldsymbol{r}_{f_u}^t\right)\right)^2. \tag{78}$$

Thus, we can conduct similar calculation in (70) and obtain the upper bound $\left\|\boldsymbol{E}_{2s}^t\right\|_\infty + \left\|\boldsymbol{E}_{3s}^t\right\|_\infty + \left\|\boldsymbol{g}_u^t\right\|_\infty + \left\|\boldsymbol{r}_{f_u}^t\right\|_\infty \leq B_k$. Based on the step size condition that $\eta \leq \frac{c_{10}(1-\sqrt{\rho})}{w_{\max}^\star}$, therefore there is a similar result in (35) as

$$\left\|\overline{\boldsymbol{u}}_{\mathcal{S}^c}^{t+1}\right\|_\infty \geq \left\|\overline{\boldsymbol{u}}_{\mathcal{S}^c}^t\right\|_\infty \left(1 - c_{10}\left(1 - \sqrt{\rho}\right)\right)^2. \tag{79}$$

Based on the recursion in (78), $\forall k$-th induction stage, we have the following upper bound

$$
\prod_{t=0}^{\overline{T}_k - 1} \left(1 + 8\eta\left(\left\|\boldsymbol{E}_{2s}^t\right\|_\infty + \left\|\boldsymbol{E}_{3s}^t\right\|_\infty + \left\|\boldsymbol{g}_u^t\right\|_\infty + \left\|\boldsymbol{r}_{f_u}^t\right\|_\infty\right)\right)^2
$$

$$
\overset{(i)}{\leq} \prod_{t=0}^{\overline{T}_k - 1} \left(1 + 8\eta \cdot 4C_b\zeta\right)^2 \left(1 + 8\eta \cdot \frac{C_\gamma}{K}\left(\left\|\left(\overline{\boldsymbol{u}}_{\mathcal{S}+}^t\right)^2 - \boldsymbol{w}_{\mathcal{S}+}^\star\right\|_\infty + \left\|\left(\overline{\boldsymbol{v}}_{\mathcal{S}-}^t\right)^2 - \boldsymbol{w}_{\mathcal{S}-}^\star\right\|_\infty\right)\right)^2
$$

$$
\cdot \left(1 + 8\eta\left(\frac{\rho^{\frac{3}{4}} B_k \left(150\left(\sqrt{s}\delta_{\max} + 3\right) + 3\sqrt{\rho}\right)}{800} + \frac{d\alpha\rho^{\frac{3}{4}}}{32}\right)\right)^2 + \left(1 + 8\eta\left\|\boldsymbol{r}_{f_u}^t\right\|_\infty\right)^2
$$

$$
\overset{(ii)}{\leq} \left(1 + 32\eta C_b\zeta\right)^{4T_k} \left(1 + 4\eta \cdot \frac{2C_\gamma w_{\max}^\star}{K \times 2^k}\right)^{4(k+1)T_k} \prod_{t=0}^{\overline{T}_k - 1} \left(1 + 8\eta \cdot \frac{\rho^{\frac{3}{4}}\left(\sqrt{s}\delta_{\max} + 4\right)}{200} \cdot \frac{w_{\max}^\star}{2^k}\right)^2
$$

$$
\cdot \left(1 + 8\eta\rho^{\frac{3}{4}}\sqrt{s}\delta_{\max}B_k\right)^2 \left(1 + 8\eta \cdot \rho^{\frac{3}{4}}\left(32d\alpha + \max_i \left\|\frac{\boldsymbol{X}_i^T\boldsymbol{\xi}_i}{n}\right\|_\infty\right)\right)^2
$$

$$
\overset{(iii)}{\leq} \left(1 + 32\eta C_b\zeta\right)^{8T_k} \left(1 + 4\eta \cdot \frac{2C_\gamma w_{\max}^\star}{K \times 2^k}\right)^{12(k+1)T_k}
$$

$$
\overset{(iv)}{\leq} \left(1 + 32\eta C_b\zeta\right)^{8T_{K-1}} \left(1 + 4\eta \cdot \frac{2C_\gamma w_{\max}^\star}{K \times 2^{K-1}}\right)^{12KT_{K-1}}
$$

$$
\overset{(v)}{\leq} \left(1 + 4\eta \cdot \frac{8C_b\zeta}{K}\right)^{8KT_{K-1}} \left(1 + 4\eta \cdot \frac{2C_\gamma w_{\max}^\star}{K \times 2^{K-1}}\right)^{12KT_{K-1}}
$$

$$
\overset{(vi)}{\leq} \left(1 + 4\eta \cdot \frac{2C_\gamma w_{\max}^\star}{K \times 2^{K-1}}\right)^{20KT_{K-1}}
$$

$$
\leq \frac{1}{\alpha} \tag{80}
$$

where $(i)$ is due to $\forall x_1, x_2, \cdots, x_n \geq 0$, there is $1 + \sum_{i=1}^{n} x_i \leq \prod_{i=1}^{n} (1 + x_i)$, substituting the upper bound in (76) and definition of global RIP parameter. For $(ii)$, we use $\overline{T}_k \leq 2T_k$ based on definition of $\overline{T}_k, T_k$, induction (a) with Lemma B.13 in [36], step size condition, small initialization for $\alpha$, definition of $B_k$ and refined upper bound for $\left\| r_{f_u}^t \right\|_\infty$ as $\left\| f_u^t \right\|_\infty \leq \rho^{\frac{3}{4}} \left( \sqrt{s} \delta_{\max} B_k + 32 d\alpha + \max_i \left\| \frac{X_i^T \xi_i}{n} \right\|_\infty \right) \left\| \overline{u}_{\mathcal{S}^c}^t \right\|_\infty$. $(iii)$ is because network connectivity condition that $\rho \leq \min \left\{ \left( \frac{\sqrt{s}\delta}{\sqrt{s}\delta_{\max}+4} \right)^{\frac{4}{3}}, \frac{\| X^T \xi \|_\infty}{8m \max_i \| X_i^T \xi_i \|_\infty} \right\}$. $(iv), (v)$ are due to condition $i \leq K - 1$ and inequality $1 + \sum_{i=1}^{K} x_i \leq \prod_{i=1}^{K} (1 + x_i)$ that $x_1 = , \cdots, x_K = \frac{32\eta C_b \zeta}{K}$. $(vi)$ is based on the definition of $K$ that $\zeta \leq \frac{w_{\max}^\star}{2^{K-1}}$ and condition that $4C_b \leq C_\gamma$. The last inequality results from the Lemma A.2 in [36] with upper bound is set as $B = \frac{2C_\gamma w_{\max}^\star}{K \times 2^{K-1}}$ and initialization $x_0 = \alpha^2$, then for $t \leq \frac{K \times 2^{K-1}}{64\eta C_\gamma w_{\max}^\star} \log \frac{1}{\alpha^4}$, there is

$$\left( 1 + 4\eta \cdot \frac{2C_\gamma w_{\max}^\star}{K \times 2^{K-1}} \right)^t \leq \frac{1}{\alpha} \tag{81}$$

Thus, we select universal constant $C_\gamma \leq \frac{1}{1280}$ such that $\frac{K \times 2^{K-1}}{64\eta C_\gamma w_{\max}^\star} \log \frac{1}{\alpha^4} \geq 20K \times T_{K-1}$, which would lead to the last inequality.

(i) Concentrate on the iteration by DGD algorithm on support $\mathcal{S}$, based on (57), we have the recursive formula as

$$\overline{u}_{\mathcal{S}}^{t+1} = \overline{u}_{\mathcal{S}}^t \odot \left( \mathbf{1}_d - 4\eta \left( \left( \left( \overline{u}_{\mathcal{S}}^t \right)^2 - \left( \overline{v}_{\mathcal{S}}^t \right)^2 - w^\star \right) + E_{d1}^t + E_{d2}^t + E_{d3}^t \right) \right)$$
$$- \frac{4\eta}{m} \sum_{i=1}^{m} \Delta_{u,\mathcal{S}}^{t,i} \odot E_{d4}^{t,i}, \tag{82}$$

where the $E_{d1}^t, E_{d2}^t, E_{d3}^t$ and $E_{d4}^{t,i}$ are defined as

$$E_{d1}^t := \left( \frac{X^T X}{N} - I \right) \left( \left( \overline{u}_{\mathcal{S}}^t \right)^2 - \left( \overline{v}_{\mathcal{S}}^t \right)^2 - w^\star \right);$$

$$E_{d2}^t := \frac{X^T X}{N} \left( \left( \overline{u}_{\mathcal{S}^c}^t \right)^2 - \left( \overline{v}_{\mathcal{S}^c}^t \right)^2 \right) - \frac{X^T \xi}{N};$$

$$E_{d3}^t := - \sum_{i=1}^{m} \frac{X_i^T X_i}{n} \left( \left( \left( \overline{u}_{\mathcal{S}}^t \right)^2 - \left( u_{\mathcal{S}}^{t,i} \right)^2 + \left( v_{\mathcal{S}}^{t,i} \right)^2 - \left( \overline{v}_{\mathcal{S}}^t \right)^2 \right) \right.$$
$$\left. + \left( \left( \overline{u}_{\mathcal{S}^c}^t \right)^2 - \left( u_{\mathcal{S}^c}^{t,i} \right)^2 + \left( v_{\mathcal{S}^c}^{t,i} \right)^2 - \left( \overline{v}_{\mathcal{S}^c}^t \right)^2 \right) \right);$$

$$E_{d4}^{t,i} := \left( u_{\mathcal{S}}^{t,i} \right)^2 - \left( v_{\mathcal{S}}^{t,i} \right)^2 + \left( \frac{X_i^T X_i}{n} - I \right) \left( \left( u_{\mathcal{S}}^{t,i} \right)^2 - \left( v_{\mathcal{S}}^{t,i} \right)^2 - w^\star \right)$$
$$+ \frac{X_i^T X_i}{n} \left( \left( u_{\mathcal{S}^c}^{t,i} \right)^2 - \left( v_{\mathcal{S}^c}^{t,i} \right)^2 \right) - \frac{X_i^T \xi_i}{n}. \tag{83}$$

For the negative part $\overline{v}^t$, with similar decomposition in (57), there is

$$\overline{v}^{t+1} = \overline{v}^t - \frac{\eta}{m} \sum_{i=1}^{m} \nabla_v f_i \left( \overline{u}^t, \overline{v}^t \right) + \frac{\eta}{m} \sum_{i=1}^{m} \left( \nabla_v f_i \left( \overline{u}^t, \overline{v}^t \right) - \nabla_v f_i \left( u^{t,i}, \overline{v}^t \right) \right)$$
$$+ \frac{\eta}{m} \sum_{i=1}^{m} \left( \nabla_v f_i \left( u^{t,i}, \overline{v}^t \right) - \nabla_v f_i \left( u^{t,i}, v^{t,i} \right) \right). \tag{84}$$

After tedious calculation, its formula is as follows

$$\overline{v}_{\mathcal{S}}^{t+1} = \overline{v}_{\mathcal{S}}^t \odot \left( \mathbf{1}_d + 4\eta \left( \left( \left( \overline{u}_{\mathcal{S}}^t \right)^2 - \left( \overline{v}_{\mathcal{S}}^t \right)^2 - w^\star \right) + E_{d1}^t + E_{d2}^t + E_{d3}^t \right) \right)$$

$$+ \frac{4\eta}{m} \sum_{i=1}^{m} \boldsymbol{\Delta}_{v,\mathcal{S}}^{t,i} \odot \boldsymbol{E}_{d4}^{t,i}. \tag{85}$$

Based on induction (c) and (d), we can reparameterize $\boldsymbol{\Delta}_{u,\mathcal{S}}^{t,i}$ and $\boldsymbol{\Delta}_{v,\mathcal{S}}^{t,i}$ as $\boldsymbol{\Delta}_{u,\mathcal{S}}^{t,i} = \boldsymbol{r}_{u,\mathcal{S}}^{t,i} \odot \overline{\boldsymbol{u}}_{\mathcal{S}}^t$ and $\boldsymbol{\Delta}_{v,\mathcal{S}}^{t,i} = \boldsymbol{r}_{v,\mathcal{S}}^{t,i} \odot \overline{\boldsymbol{v}}_{\mathcal{S}}^t$, where $\max\left\{ \left\|\boldsymbol{r}_{u,\mathcal{S}}^{t,i}\right\|_\infty, \left\|\boldsymbol{r}_{v,\mathcal{S}}^{t,i}\right\|_\infty \right\} \le 4\beta\rho^{\frac{3}{4}}\eta B_k$ for $\overline{T}_{k-1} \le t < \overline{T}_k$, Thus, the (82) and (85) can be reformulated as

$$\left(\overline{\boldsymbol{u}}_{\mathcal{S}}^{t+1}\right)^2 = \left(\overline{\boldsymbol{u}}_{\mathcal{S}}^t\right)^2 \odot \left( \mathbf{1}_d - 4\eta\left(\left(\left(\overline{\boldsymbol{u}}_{\mathcal{S}}^t\right)^2 - \left(\overline{\boldsymbol{v}}_{\mathcal{S}}^t\right)^2 - \boldsymbol{w}^\star\right) + \boldsymbol{E}_{d1}^t + \boldsymbol{E}_{d2}^t + \boldsymbol{E}_{d3}^t\right) - 4\eta\frac{\sum_{i=1}^m \boldsymbol{r}_{u,\mathcal{S}}^{t,i} \odot \boldsymbol{E}_{d4}^{t,i}}{m} \right)^2 \tag{86}$$

$$\left(\overline{\boldsymbol{v}}_{\mathcal{S}}^{t+1}\right)^2 = \left(\overline{\boldsymbol{v}}_{\mathcal{S}}^t\right)^2 \odot \left( \mathbf{1}_d + 4\eta\left(\left(\left(\overline{\boldsymbol{u}}_{\mathcal{S}}^t\right)^2 - \left(\overline{\boldsymbol{v}}_{\mathcal{S}}^t\right)^2 - \boldsymbol{w}^\star\right) + \boldsymbol{E}_{d1}^t + \boldsymbol{E}_{d2}^t + \boldsymbol{E}_{d3}^t\right) + 4\eta\frac{\sum_{i=1}^m \boldsymbol{r}_{v,\mathcal{S}}^{t,i} \odot \boldsymbol{E}_{d4}^{t,i}}{m} \right)^2. \tag{87}$$

Based on hypotheses (a), (c)-(f), (j) and step size condition, we have the following bound for additional perturbed error terms

$$\max\left\{ \left\|\boldsymbol{r}_{u,\mathcal{S}}^{t,i} \odot \boldsymbol{E}_{d4}^{t,i}\right\|_\infty, \left\|\boldsymbol{r}_{v,\mathcal{S}}^{t,i} \odot \boldsymbol{E}_{d4}^{t,i}\right\|_\infty \right\} \le 4\beta\rho^{\frac{3}{4}}\eta B_k \left( 4\left(\left\|\overline{\boldsymbol{u}}_{\mathcal{S}}^t\right\|_\infty^2 + \left\|\overline{\boldsymbol{v}}_{\mathcal{S}}^t\right\|_\infty^2\right) \right.$$
$$\left. +2\sqrt{s}\delta_{\max}\left(w_{\max}^\star + \left\|\overline{\boldsymbol{u}}_{\mathcal{S}}^t\right\|_\infty^2 + \left\|\overline{\boldsymbol{v}}_{\mathcal{S}}^t\right\|_\infty^2\right) + 16d\alpha + \max_i \left\|\frac{\boldsymbol{X}_i^T\boldsymbol{\xi}_i}{n}\right\|_\infty \right)$$
$$\le 4\beta\sqrt{\rho}\eta B_k \left( \rho^{\frac{1}{4}}\left(16 + 10\sqrt{s}\delta_{\max}\right)w_{\max}^\star + 16d\alpha + \rho^{\frac{1}{4}}\max_i \left\|\frac{\boldsymbol{X}_i^T\boldsymbol{\xi}_i}{n}\right\|_\infty \right)$$
$$\le 40\beta\sqrt{\rho}\eta \cdot \frac{w_{\max}^\star}{40 \times 2^k} \cdot \rho^{\frac{1}{4}}(2 + \sqrt{s}\delta_{\max})w_{\max}^\star$$
$$\le \frac{C_\gamma w_{\max}^\star}{K \times 2^k}, \tag{88}$$

where the last inequality is due to network connectivity condition that $\rho \le \min\left\{ \left(\frac{\sqrt{s}\delta}{\sqrt{s}\delta_{\max}+4}\right)^{\frac{4}{3}}, \frac{\|\boldsymbol{X}^T\boldsymbol{\xi}\|_\infty}{8m\max_i\|\boldsymbol{X}_i^T\boldsymbol{\xi}_i\|_\infty} \right\}$.

Then multiplying (86) and (87), we can obtain that

$$\left(\overline{\boldsymbol{u}}_{\mathcal{S}}^{t+1} \odot \overline{\boldsymbol{v}}_{\mathcal{S}}^{t+1}\right)^2 \le \left(\overline{\boldsymbol{u}}_{\mathcal{S}}^t \odot \overline{\boldsymbol{v}}_{\mathcal{S}}^t\right)^2 \odot \left( 1 - \left(4\eta\left(\left(\overline{\boldsymbol{u}}_{\mathcal{S}}^t\right)^2 - \left(\overline{\boldsymbol{v}}_{\mathcal{S}}^t\right)^2 - \boldsymbol{w}^\star + \boldsymbol{E}_{d1}^t + \boldsymbol{E}_{d2}^t + \boldsymbol{E}_{d3}^t\right)\right)^2 \right.$$
$$+8\eta\max\left\{ \left\|\boldsymbol{r}_{u,\mathcal{S}}^{t,i} \odot \boldsymbol{E}_{d4}^{t,i}\right\|_\infty, \left\|\boldsymbol{r}_{v,\mathcal{S}}^{t,i} \odot \boldsymbol{E}_{d4}^{t,i}\right\|_\infty \right\}$$
$$\left. + \left(4\eta\max\left\{ \left\|\boldsymbol{r}_{u,\mathcal{S}}^{t,i} \odot \boldsymbol{E}_{d4}^{t,i}\right\|_\infty, \left\|\boldsymbol{r}_{v,\mathcal{S}}^{t,i} \odot \boldsymbol{E}_{d4}^{t,i}\right\|_\infty \right\}\right)^2 \right)^2$$
$$\overset{(i)}{\le} \left(\overline{\boldsymbol{u}}_{\mathcal{S}}^t \odot \overline{\boldsymbol{v}}_{\mathcal{S}}^t\right)^2 \odot \left( 1 + 4\eta \cdot \frac{C_\gamma w_{\max}^\star}{K \times 2^k} \right)^4$$
$$\overset{(ii)}{\le} \alpha^4 \odot \left( 1 + 4\eta \cdot \frac{C_\gamma}{K} 2^{-K+1} w_{\max}^\star \right)^{8KT_{K-1}}$$
$$\overset{(iii)}{\le} \alpha^3, \tag{89}$$

where the $(i)$ is due to (88) and $4\eta\left\|\left(\overline{\boldsymbol{u}}_{\mathcal{S}}^t\right)^2 - \left(\overline{\boldsymbol{v}}_{\mathcal{S}}^t\right)^2 - \boldsymbol{w}^\star + \boldsymbol{E}_{d1}^t + \boldsymbol{E}_{d2}^t + \boldsymbol{E}_{d3}^t\right\|_\infty \le 1$. The $(ii)$ is due to induction (a) and Lemma B.13 in [36]. The reason for $(iii)$ is the same as the that of last inequality in (80) because of the setting of step size.

(j) Recall the recursive formula of $\overline{\boldsymbol{u}}_{\mathcal{S}^-}^t$ in (49) and compare the definition of the perturbation $\boldsymbol{y}_u^t$ with perturbations $\boldsymbol{p}_u^t, \boldsymbol{g}_u^t$, we can obtain $\|\boldsymbol{y}_u^t\|_\infty \leq \frac{150\rho^{\frac{3}{4}}B_k\left(\sqrt{s}\delta_{\max}+4\right)}{600} + \frac{\rho^{\frac{3}{4}}d\alpha}{32}$. Compare the outer perturbation $\boldsymbol{z}_u^t$ with outer perturbations $\boldsymbol{f}_u^t, \boldsymbol{q}_u^t$, we can obtain $\|\boldsymbol{z}_u^t\|_\infty \leq \frac{\sqrt{\rho}B_k\|\overline{\boldsymbol{u}}_{\mathcal{S}^-}^t\|_\infty}{16}$. With similar reparameterization as $\boldsymbol{z}_u^t = \boldsymbol{r}_{z_u}^t \odot \overline{\boldsymbol{u}}_{\mathcal{S}^-}^t$, the (49) would become

$$\left(\overline{\boldsymbol{u}}_{\mathcal{S}^-}^{t+1}\right)^2 = \left(\overline{\boldsymbol{u}}_{\mathcal{S}^-}^t\right)^2 \odot \left(\boldsymbol{1}_d - 4\eta\left(-\boldsymbol{w}_{\mathcal{S}^-}^\star - \left(\overline{\boldsymbol{v}}_{\mathcal{S}^-}^t\right)^2 + \boldsymbol{E}_{2s}^t + \boldsymbol{E}_{3s}^t + \boldsymbol{y}_u^t + \boldsymbol{r}_{z_u}^t\right)\right)^2 \qquad (90)$$

$\forall j \in \mathcal{S}^-$ and $\forall t \leq \mathcal{O}\left(\frac{1}{\eta\zeta}\log\frac{1}{\alpha}\right)$. Let $0 \leq \tau \leq t$ be the largest $\tau$ such that $\left(\overline{v}_j^\tau\right)^2 > -w_j^\star$. If there is no such $\tau$ exists or $\tau = t$, then

$$\left(\overline{u}_j^t\right)^2 \leq \left(\overline{u}_j^{t-1}\right)^2 \left(1 + 4\eta\left(\left\|\boldsymbol{E}_{2s}^{t-1}\right\|_\infty + \left\|\boldsymbol{E}_{3s}^{t-1}\right\| + \left\|\boldsymbol{y}_s^{t-1}\right\|_\infty + \left\|\boldsymbol{r}_{z_s}^{t-1}\right\|_\infty\right)\right)^2$$
$$\leq \alpha, \qquad (91)$$

where the last inequality is due to the similar bound in (80).

If $\tau < t$, then unrolling (90) to $\tau$-th iteration would have

$$\left(\overline{u}_j^t\right)^2 = \left(\overline{u}_j^\tau\right)^2 \prod_{i=\tau}^{t-1}\left(1 - 4\eta\left(-w_j^\star - \left(\overline{v}_j^i\right)^2 + E_{2s,j}^i + E_{3s,j}^i + y_{s,j}^t + r_{z_s,j}^i\right)\right)^2$$

$$\overset{(i)}{\leq} \frac{\alpha^2}{4}\left(1 - 4\eta\left(-w_j^\star - \left(\overline{v}_j^\tau\right)^2 + E_{2s,j}^\tau + E_{3s,j}^\tau + y_{s,j}^\tau + r_{z_s,j}^\tau\right)\right)^2$$

$$\cdot \prod_{i=\tau+1}^{t-1}\left(1 - 4\eta\left(-w_j^\star - \left(\overline{v}_j^i\right)^2 + E_{2s,j}^i + E_{3s,j}^i + y_{s,j}^t + r_{z_s,j}^i\right)\right)^2$$

$$\overset{(ii)}{\leq} \alpha^2 \prod_{i=\tau+1}^{t-1}\left(1 + 4\eta\left(\left\|\boldsymbol{E}_{2s}^i\right\|_\infty + \left\|\boldsymbol{E}_{3s}^i\right\| + \left\|\boldsymbol{y}_s^i\right\|_\infty + \left\|\boldsymbol{r}_{z_s}^i\right\|_\infty\right)\right)^2$$

$$\overset{(iii)}{\leq} \alpha, \qquad (92)$$

where $(i)$ is based on condition for initialization that $\alpha \leq \frac{w_{\min}^\star}{4}$ and induction (h) that $\overline{u}_j^\tau \leq \frac{\alpha^{\frac{3}{2}}}{\overline{v}_j^\tau} \leq \frac{\alpha^{\frac{3}{2}}}{\sqrt{-w_j^\star}} \leq \frac{\alpha}{2}$. $(ii)$ is based on the induction (a), step size condition that $\left(1 - 4\eta\left(-w_j^\star - \left(\overline{v}_j^\tau\right)^2 + E_{2s,j}^\tau + E_{3s,j}^\tau + y_{s,j}^\tau + r_{z_s,j}^\tau\right)\right)^2 \leq 4$ and $\forall i > \tau$ that $\left(\overline{v}_j^i\right)^2 < -w_j^\star$ which is based on definition of $\tau$. The last inequality has the same reason as (91).

## A.5 Proof of Main Results

### A.5.1 Proof of Theorem 1

Based on the proof of hypothesis (a) in Proposition 3, we can conclude that $\forall t \geq \overline{T}_K$, there would be $\max\left\{\left\|\left(\overline{\boldsymbol{u}}_{\mathcal{S}^+}^{\overline{T}_{k-1}}\right)^2 - \boldsymbol{w}_{\mathcal{S}^+}^\star\right\|_\infty, \left\|\left(\overline{\boldsymbol{v}}_{\mathcal{S}^-}^{\overline{T}_{k-1}}\right)^2 - \boldsymbol{w}_{\mathcal{S}^-}^\star\right\|_\infty\right\} \leq \frac{w_{\max}^\star}{2^K} \leq \zeta$, where the last inequality is due to the definition of $K$. The total computational complexity is the value of $\overline{T}_{K-1} = \frac{2^K-1}{\eta w_{\max}^\star}\log\frac{1}{\alpha^4} = \mathcal{O}\left(\frac{1}{\eta\zeta}\log\frac{1}{\alpha}\right)$.

Due to the definition of $\zeta$ in the Theorem 1, if $\zeta \geq w_{\max}^\star$, then our result holds at $t = 0$ based on the small initialization condition. When $\zeta \leq w_{\max}^\star$, we consider the two cases where the first case is the magnitudes of parameters is strong enough that $\zeta = \frac{1}{5}w_{\min}^\star \geq 960\varsigma$, where the value 960 is due to under the condition $2C_b + C_\gamma \leq \frac{1}{80}, 4C_b \leq C_\gamma$, we set $C_b = \frac{1}{480}, C_\gamma = \frac{1}{120}$, we can use run DGD $T_1'$ iterations to obtain the estimator $\max\left\{\left\|\left(\overline{\boldsymbol{u}}_{\mathcal{S}^+}^{\overline{T}_{k-1}}\right)^2 - \boldsymbol{w}_{\mathcal{S}^+}^\star\right\|_\infty, \left\|\left(\overline{\boldsymbol{v}}_{\mathcal{S}^-}^{\overline{T}_{k-1}}\right)^2 - \boldsymbol{w}_{\mathcal{S}^-}^\star\right\|_\infty\right\} \leq \frac{w_{\max}^\star}{2^K} \leq \frac{1}{5}w_{\min}^\star$, then based on Proposition 2 in [36], we can also run DGD with another $\frac{45}{32\eta w_{\min}^\star}\log\frac{w_{\min}^\star}{\epsilon}$ iterations to obtain the dimension-independent bound $|\overline{w}_j^t - w_j^\star| \leq \max\left\{\sqrt{s}\delta\max_{i \in \mathcal{S}}B_i, B_j, \epsilon\right\}, \forall j \in$

$\mathcal{S}$ where the $B_j := \left\| \frac{\boldsymbol{X}^T \boldsymbol{\xi}}{N} \odot \boldsymbol{1}_j \right\|_\infty$. The total iterations are $\overline{T}_{K-1} + \frac{45}{32\eta w_{\min}^\star} \log \frac{w_{\min}^\star}{\epsilon} \leq \mathcal{O}\left( \frac{1}{\eta w_{\min}^\star} \log \frac{1}{\alpha} \right) = \mathcal{O}\left( \frac{1}{\eta \zeta} \log \frac{1}{\alpha} \right)$ where the first inequality is due to the condition of $\alpha$.

The second case is $\zeta = 960\varsigma \geq \frac{1}{5} w_{\min}^\star$, then running of DGD with $\mathcal{O}\left( \frac{1}{\eta\zeta} \right)$ iterations would obtain the result in Theorem 1.

### A.5.2 Proof of Corollary 1

Because the entries of global design matrix generated from i.i.d. 1-Sub-Gaussian distribution, we have the upper bound of global RIP parameter as $\delta \lesssim \frac{1}{\sqrt{s}}$ probability at least $1 - \frac{1}{8d^3}$, which is based on the Lemma 1 and sample size lower bound in Corollary 1 matches the condition of sample complexity in Lemma 1. For the local design matrix $\{\boldsymbol{X}_i/\sqrt{n}\}_{i=1}^m$, local sample size satisfies $n = \frac{N}{m} \gtrsim \left( \sqrt{\frac{m}{\delta}} \right)^{-2} \left( s \ln \frac{ed}{s} + \ln(dn) \right)$, we can bound the local RIP parameter as $\delta_{\max} \leq \sqrt{\frac{m}{\delta}}$ with probability as least $1 - \frac{1}{8d^3}$ based on the union bound and Lemma 1. Based on condition in (6), the $\rho \lesssim \frac{1}{m^4}$ holds with probability at least $1 - \frac{1}{4d^3}$. Based on the Lemma 4, we have the upper bound for the noise level as $\left\| \frac{\boldsymbol{X}^T \boldsymbol{\xi}}{N} \right\|_\infty \lesssim \sigma \sqrt{\frac{\log d}{N}}$ with probability at least $1 - \frac{1}{8d^3}$. Thus, with probability at least $1 - \frac{3}{8d^3}$, we have $\left\| \overline{\boldsymbol{w}}^t - \boldsymbol{w}^\star \right\|_2^2 \lesssim s\epsilon^2 + \frac{d\epsilon^2}{(2d+1)^2} \lesssim \frac{s\sigma^2 \log d}{N}$ where in the last inequality we select $\epsilon = 4\sigma\sqrt{\frac{2\log(2d)}{N}}$ in Theorem 1.

### A.5.3 Proof of Proposition 1

Because each agent has same initialization $\boldsymbol{u}^{0,i} = \boldsymbol{v}^{0,i} = \alpha\boldsymbol{1}_d, \forall i \in [m]$, after one step of local gradient descent from (3) and (4), there are

$$\boldsymbol{u}^{1,i} = \alpha \odot \left( 1 + 4\eta \left( \boldsymbol{w}^\star + \left( \frac{\boldsymbol{X}_i^T \boldsymbol{X}_i}{n} - \boldsymbol{I} \right) \boldsymbol{w}^\star + \frac{\boldsymbol{X}_i^T \boldsymbol{\xi}_i}{n} \right) \right)$$

$$\boldsymbol{v}^{1,i} = \alpha \odot \left( 1 - 4\eta \left( \boldsymbol{w}^\star + \left( \frac{\boldsymbol{X}_i^T \boldsymbol{X}_i}{n} - \boldsymbol{I} \right) \boldsymbol{w}^\star + \frac{\boldsymbol{X}_i^T \boldsymbol{\xi}_i}{n} \right) \right). \tag{93}$$

Because the perturbed error bound

$$\left\| \left( \frac{\boldsymbol{X}_i^T \boldsymbol{X}_i}{n} - \boldsymbol{I} \right) \boldsymbol{w}^\star \right\|_\infty + \left\| \frac{\boldsymbol{X}_i^T \boldsymbol{\xi}_i}{n} \right\|_\infty \lesssim \phi := \sqrt{s}\delta_{\max} w_{\max} + \sigma\sqrt{\frac{\log d}{n}} \tag{94}$$

holds with probability at least $1 - \frac{3}{8d^3}$ based on Lemma 2 and Lemma 4, we denote $\boldsymbol{\nu}^i = \left( \frac{\boldsymbol{X}_i^T \boldsymbol{X}_i}{n} - \boldsymbol{I} \right) \boldsymbol{w}^\star + \frac{\boldsymbol{X}_i^T \boldsymbol{\xi}_i}{n}$.

Consider $\boldsymbol{u}^{1,i}$, for $\forall p \in \mathcal{S}^+, \forall q \in \mathcal{S}^c$, there is $w_p^\star - |\nu_p^i| > \phi > |\nu_q^i| > 0$ based on the condition in Proposition 1. In addition, $\forall j \in \mathcal{S}^-$, there is $-|w_j^\star| + |\nu_j| < 0$ based on the the condition on Proposition 1. Thus, the growth of elements on positive support $\mathcal{S}^+$ would be larger than these of $\mathcal{S}^-, \mathcal{S}^c$, and the $\text{Trun}_k$ operator would identify the $\mathcal{S}^+$. The analogous analysis could also applied to $\boldsymbol{v}^{1,i}$ that the $\text{Trun}_k$ operator would identify the $\mathcal{S}^-$. Because each agent can identify the $\mathcal{S}^+$ and $\mathcal{S}^-$ and based on results in Proposition 3, the $\text{Trun}_k$ would also obtain the optimal statistical error with probability at least $1 - \frac{3}{8d^3}$. $\qquad\square$

## B  Appendix.B

### B.1  Kernel to rich regime transition

Previous works show a transition from the kernel regime to the rich regime by varying the initialization scale in the gradient descent method [38]. Experimental results in Fig. 7 show that the transition phenomenon also appears in DGD. We can observe that when we increase the initialization scale $\alpha$ gradually, DGD would converge to the minimal $\ell_2$ norm solution $\boldsymbol{w}_{\ell_2}^\star$. On the contrary, when we

decrease the $\alpha$, the DGD would converge to the minimal $\ell_1$ norm solution (sparse solution) $\boldsymbol{w}_{\ell_1}^{\star}$. Thus, Fig. 7 demonstrates the existence of phase transition from kernel to rich regime for DGD when decreasing initialization $\alpha$. Since we focus on sparse recovery, the small initialization would achieve this aim with better generalization performance.

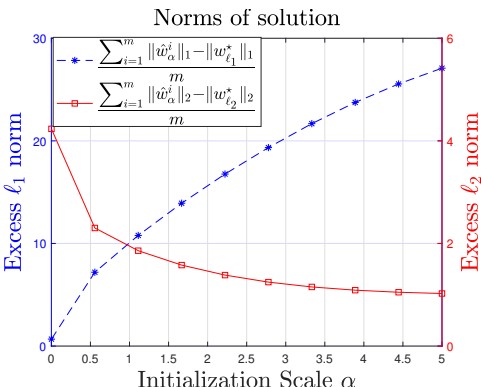

Figure 7: We plot $\frac{\sum_{i=1}^{m}\|\hat{\boldsymbol{w}}_{\alpha}^{i}\|_1 - \|\boldsymbol{w}_{\ell_1}^{\star}\|_1}{m}$ in the blue and $\frac{\sum_{i=1}^{m}\|\hat{\boldsymbol{w}}_{\alpha}^{i}\|_2 - \|\boldsymbol{w}_{\ell_1}^{\star}\|_2}{m}$ in red vs.$\alpha$, where $\hat{\boldsymbol{w}}_{\alpha}^{i}$ denote the convergent solution by DGD in $i$-th agent. The setting is $d = 2000, k = 2, N = 20, m = 20, \rho = 0.2135$ and magnitude of sparse signal is 10.

## B.2  More implicit regularizations of DSGD

To verify the widespread existence of implicit regularization in decentralized optimization, we add two extra experiments on general overparameterized neural network architecture trained by DSGD, which are motivated from [38].

(1) The first one is that we use vanilla decentralized SGD(DSGD) to train a depth-2, 5000 hidden ReLU network with the cross-entropy loss on the MNIST dataset until each agent model reaches almost 100% training accuracy. The number of agents is 10 and network connectivity is $\rho = 0.178$. 60000 total training samples and 10000 test samples are uniformly allocated to agents. To evaluate the implicit regularization of SGD under varying initialization scales, the network weights are initialized as $\alpha\boldsymbol{w}_0, \boldsymbol{w}_0 \sim \mathcal{N}(0, \sqrt{\frac{2}{n_{\text{in}}}})$, which is suggested by [14] and $n_{\text{in}}$ denotes the number of units in the last year. Each agent uses the same batch size 256 to train in DSGD. The step sizes were optimally tuned for each $\alpha$ individually to achieve the best validation error. We plot the average test error (which is defined as the summation of the test error of each agent's model, then divided by the number of agents) vs. $\alpha$ in Fig. 8(a). The figure shows a visible phase transition for generalization ($\approx 98\%$ for $\alpha \leq 6$, and $\approx 96.6\%$ error for $\alpha \geq 100$). Fig. 8(a) shows that the transition from the kernel regime to the rich regime by varying the initialization scale may also exist in complex fully connected neural networks.

(2) The second one is that we use vanilla DSGD with batch size 128 to train the VGG11-like deep convectional neural network on CIFAR10 with small step size $10^{-4}$ for 2000 epochs such that each local model achieves almost 100% training accuracy. The network setting is the same as the first experiment. The VGG11-like architecture is the same as [38]. Weights were initialized using Uniform He initialization multiplied by $\alpha$. The Fig. 8(b) plots the average test accuracy vs. $\alpha$. In addition, we adopt the sparse feature learning measure [4] to monitor the sparsity of learned features in all agents along the epoch in Fig. 8(c). From Fig. 8(c) and Fig. 8(d), we can observe that the implicit regularization of DGD under small initialization may promote neural networks to learn sparse features that can have better generalization performance, which can be regarded as the complementary observation in [4] under decentralized learning setting. How to prove this kind of implicit regularization of DSGD theoretically will be an interesting future work.

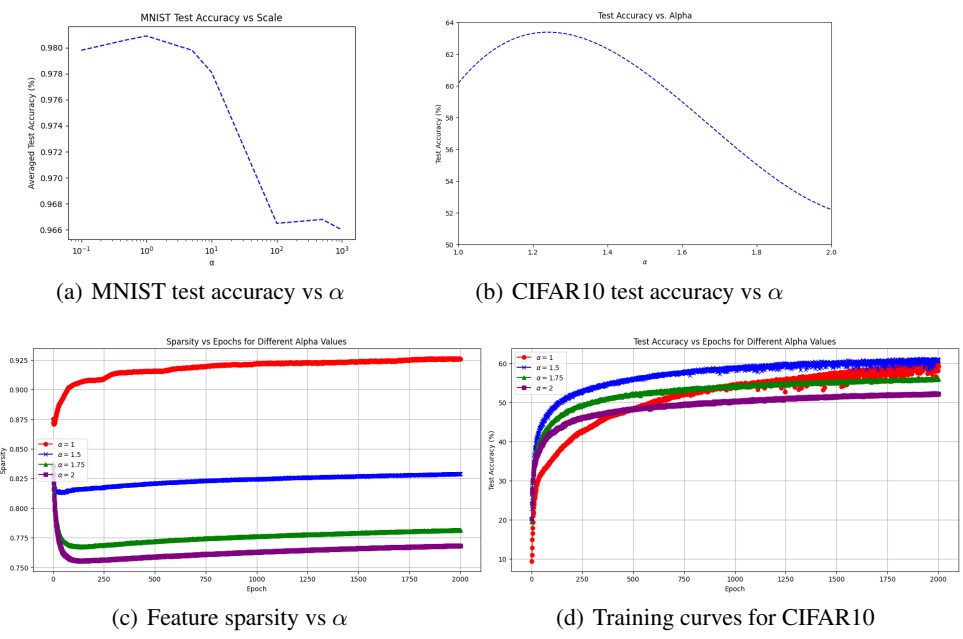

Figure 8: (a) DSGD trained ReLU network on **MNIST**. (b), (c), (d) DSGD trained VGG11-like network on **CIFAR10**.

