# OpenReview forum: "Implicit Regularization of Decentralized Gradient Descent for Sparse Regression"
_NeurIPS.cc/2024/Conference — NeurIPS 2024 poster_

### Official Review · Reviewer_Agu7 · 2024-07-07

**Soundness:** 4
**Presentation:** 4
**Contribution:** 4
**Rating:** 6
**Confidence:** 2

**Summary:**

This paper proves the convergence of the decentralized gradient descent (DGD) algorithm under RIP condition when the initialization scale is small. The authors also propose a truncated version of the algorithm with a cheaper cost but comparable performance (in certain situations) to the original DGD.

**Strengths:**

The theory seems sound and nontrivial. The authors prove the convergence for DGD, while previously it seems that only centralized GD results exist.

**Weaknesses:**

This paper is a little bit technical.

**Questions:**

There are already many technical explanations and comparisons in the paper. However, it would be better if more intuitive understanding of the convergence of DGD, and the comparison with centralized GD can be added.

**Limitations:**

Yes

---

> ### Author Rebuttal · Authors · 2024-08-07
>
> We sincerely appreciate your time and effort in reviewing our manuscript and providing valuable feedback. We have addressed your comments and questions as detailed below.\
> $\textbf{High-level explanation of promoting sparsity for centralized GD.}$ For gradient descent (GD), the diagonal linear reparameterization turns the additive updates into multiplicative updates as $w_+^{t+1} = w_+^t \odot \left(  \boldsymbol 1_d - 4\eta \left( \boldsymbol s^t - \boldsymbol w^\star + \boldsymbol p^t + \boldsymbol b^t\right) \right)^2$ and $w_-^{t+1} = w_-^t \odot \left(  \boldsymbol 1_d + 4\eta \left( \boldsymbol s^t - \boldsymbol w^\star + \boldsymbol p^t + \boldsymbol b^t\right) \right)^2$ based on signal and error decomposition, where $\boldsymbol s^t$ denotes the signal part and $\boldsymbol p^t$ and $\boldsymbol b^t$ are bounded error parts. For the element $i \in \mathcal S$, the small initialization can make $|1 \pm 4\eta( s^t_i - w^\star_i + p^t_i + b^t_i )|>1$ roughly, which can amplify the $s^{t}_i$ to $s^{t+1}_i$ and let $s^t_i$ tend to converge to $w^{\star}_i$. On the contrary, if $i \in \mathcal S^c$, we can select the appropriate step size to bound the cumulative product  $\prod^t_j { |1 \pm 4\eta(s^j_i + p^j_i + b^j_i)| }$  at the $t$-th iteration before  early stopping to keep $s^t_i$ is still small enough. The difference dynamics on elements in support set $\mathcal S$ and non-support set $\mathcal S^c$ result in sparsity. Sparsity is induced with reparameterization together with small initialization size (one without the other doesn’t work).\
> $\textbf{High-level explanation of the success of DGD and comparison.}$ The consensus errors induced from decentralized network complicate the multiplicative updates, which becomes inexact multiplicative updates as $\overline{\boldsymbol u}^{t+1} = \overline{\boldsymbol u}^t \odot \left(1 - 4\eta \left( \overline{\boldsymbol u}^t \odot \overline{\boldsymbol u}^t - \boldsymbol w^\star + \hat{\boldsymbol p}^t + \hat{\boldsymbol b}^t \right)\right) + \boldsymbol e^t$. Compared with the exact multiplicative updates, the challenge is that the extra error term $\boldsymbol e^t$ outside of the multiplication prevent applying the centralized analysis trivially. In addition, the perturbation error terms $\hat{\boldsymbol p}^t, \hat{\boldsymbol b}^t$ within the multiplication are much more complicated than the ${\boldsymbol p}^t, {\boldsymbol b}^t$ in the centralized setting due to additionally multiple consensus errors and loss of global RIP condition for each agent. This requires carefully bounding the consensus error terms, which can control the complicated perturbation errors $\hat{\boldsymbol p}^t, \hat{\boldsymbol b}^t, \boldsymbol e^t$ not to be large. Specially, we can utilize the signal and error decomposition and diagonal linear reparameterization to reparameterize $\boldsymbol e^t$ into the formula as $\boldsymbol e^t= \overline{\boldsymbol{u}}^t \odot 4\eta \boldsymbol f^t$, where the $\boldsymbol f^t$ is the bounded perturbation error that can be merged into $\hat{\boldsymbol p}^t, \hat{\boldsymbol b}^t$. Now we can rewrite the inexact multiplicative updates $\overline{\boldsymbol u}^{t+1} = \overline{\boldsymbol u}^t \odot \left(1 - 4\eta \left( \overline{\boldsymbol u}^t \odot \overline{\boldsymbol u}^t - \boldsymbol w^\star + \hat{\boldsymbol p}^t + \hat{\boldsymbol b}^t + \boldsymbol f^t \right)\right)$. Trivially applying existing consensus error analysis would give crude bounds for  $\hat{\boldsymbol p}^t, \hat{\boldsymbol b}^t, \boldsymbol f^t$, which hinder achieving statistical optimal recovery. Thus, we propose a fine-grained analysis for consensus errors to bound $\hat{\boldsymbol p}^t, \hat{\boldsymbol b}^t, \boldsymbol f^t$ carefully. In addition, the existence of $\boldsymbol e^t$ makes extending the proof of the simplified non-negative $\boldsymbol w^\star$ case to the general $\boldsymbol w^\star$ setting in the decentralized framework non-trivial compared to the centralized setting, which only needs to conduct induction hypothesis for non-negative $\boldsymbol w^\star$ case. Therefore, we conduct a comprehensive induction process to both $\boldsymbol u$ and $\boldsymbol v$ simultaneously for general $\boldsymbol w^\star$. Under our integrated induction hypothesis, we can use network connectivity to control the fine-grained consensus errors to bound these three perturbation errors $\hat{\boldsymbol p}^t, \hat{\boldsymbol b}^t, \boldsymbol f^t$ small enough to make the distance between two trajectories obtained by inexact and exact multiplicative updates within statistical accuracy, which can promote sparsity in the decentralized setting. The detailed theoretical mechanism of promoting sparsity has been demystified in Proposition 3. Please refer to the Appendix on pages 28-29.\
> $\textbf{Empirical Results Comparison between GD and DGD.}$ The comparison results are shown in Figure 4 of $\textbf{supply.pdf}$, which indicates that the trajectory of DGD  mimics that of GD. DGD can achieve optimal statistical estimation as GD. The delayed convergence of DGD is due to the decentralized network. These experimental results corroborate our high-level explanation of the success of DGD.

---

> > ### Comment · Reviewer_Agu7 · 2024-08-11
> >
> > Thank you for the detailed explanation. It helps but still seems a little technical to me. It would be better to present higher-level ideas instead of technical illustrations. Please consider adding them to the paper. I will maintain my score.

---

> > > ### Author Response · Authors · 2024-08-12
> > > **Authors' feedback**
> > >
> > > Thank you for the feedback. We will add higher-level explanations to make the ideas clearer and more accessible.

---

### Official Review · Reviewer_rjVx · 2024-07-10

**Soundness:** 3
**Presentation:** 3
**Contribution:** 3
**Rating:** 6
**Confidence:** 2

**Summary:**

This manuscript studies a decentralized optimization method for training linear sparse models using a network of agents that collect linear measurements. Unlike decentralized methods relying on L1 regularization, this approach leverages implicit regularization inherited in the gradient descent process. The authors propose a communication-efficient variant referred to as Truncated Decentralized Gradient Descent T-DGD. The authors analyze their decentralized version of gradient descent applied to a non-convex least squares formulation. The manuscript concludes with numerical results that validate the effectiveness of both DGD and T-DGD for sparse learning tasks.

**Strengths:**

The manuscript is well-written and provides an interesting perspective on decentralized optimization of a linear least squares system.

**Weaknesses:**

It remains unclear to me how the approach promotes sparsity. This has neither been discussed nor numerically explored. Here, I would expect a comparison to l1-type regularization approaches promoting sparsity in w* but such comparison and sparsity levels are not provided.

**Questions:**

See weaknesses

**Limitations:**

yes.

---

> ### Author Rebuttal · Authors · 2024-08-07
>
> Thank you for your thoughtful feedback and for highlighting the areas that need further clarification.\
> $\textbf{High-level explanation of promoting sparsity for centralized GD.}$ For gradient descent (GD), the diagonal linear reparameterization turns the additive updates into multiplicative updates as $w_+^{t+1} = w_+^t \odot \left(  \boldsymbol 1_d - 4\eta \left( \boldsymbol s^t - \boldsymbol w^\star + \boldsymbol p^t + \boldsymbol b^t\right) \right)^2$ and $w_-^{t+1} = w_-^t \odot \left(  \boldsymbol 1_d + 4\eta \left( \boldsymbol s^t - \boldsymbol w^\star + \boldsymbol p^t + \boldsymbol b^t\right) \right)^2$ based on signal and error decomposition, where $\boldsymbol s^t$ denotes the signal part and $\boldsymbol p^t$ and $\boldsymbol b^t$ are bounded error parts. For the element $i \in \mathcal S$, the small initialization can make $|1 \pm 4\eta( s^t_i - w^\star_i + p^t_i + b^t_i )|>1$ roughly, which can amplify the $s^{t}_i$ to $s^{t+1}_i$ and let $s^t_i$ tend to converge to $w^{\star}_i$. On the contrary, if $i \in \mathcal S^c$, we can select the appropriate step size to bound the cumulative product  $\prod^t_j { |1 \pm 4\eta(s^j_i + p^j_i + b^j_i)| }$  at the $t$-th iteration before early stopping to keep $s^t_i$ is still small enough. The difference dynamics on elements in support set $\mathcal S$ and non-support set $\mathcal S^c$ result in sparsity. Sparsity is induced with reparameterization together with small initialization size (one without the other doesn’t work).\
> $\textbf{High-level explanation of promoting sparsity for DGD.}$ The consensus errors induced from decentralized network complicate the multiplicative updates, which becomes inexact multiplicative updates as $\overline{\boldsymbol u}^{t+1} = \overline{\boldsymbol u}^t \odot \left(1 - 4\eta \left( \overline{\boldsymbol u}^t \odot \overline{\boldsymbol u}^t - \boldsymbol w^\star + \hat{\boldsymbol p}^t + \hat{\boldsymbol b}^t \right)\right) + \boldsymbol e^t$. Compared with the exact multiplicative updates, the challenge is that the extra error term $\boldsymbol e^t$ outside of the multiplication prevent applying the centralized analysis trivially. In addition, the perturbation error terms $\hat{\boldsymbol p}^t, \hat{\boldsymbol b}^t$ within the multiplication are much complicated than the ${\boldsymbol p}^t, {\boldsymbol b}^t$ in the centralized setting due to additionally multiple consensus errors. This requires carefully bounding the consensus error terms, which can control the complicated perturbation errors $\hat{\boldsymbol p}^t, \hat{\boldsymbol b}^t, \boldsymbol e^t$ not to be large. Thus, we can use network connectivity to control the consensus errors to bound these three perturbation errors small enough to make the distance between two trajectories obtained by inexact and exact multiplicative updates within statistical accuracy, which can promote sparsity in the decentralized setting. The detailed theoretical mechanism of promoting sparsity has been demystified in Proposition 3. Please refer to the Appendix on pages 28-29.\
> $\textbf{ Validating promoting sparsity numerically.}$  We direct the reviewer to Section 6.1 of our main paper. The simulations presented in Figure 1 provide empirical evidence supporting Proposition 3's theoretical mechanism for promoting sparsity. These results show that DGD effectively distinguishes between non-zero and zero support elements, aligning with our theoretical findings. Reviewer can also refer to Figure 4 in $\textbf{supply.pdf}$, which also demystifies the promoting sparsity for GD and DGD.\
> $\textbf{(4)Comparison with decentralized sparse solvers.}$ \
> $\textbf{(1) Vanilla Comparison.}$  \
> We have compared with existing decentralized sparse solvers, namely: CTA-DGD (LASSO) [2], ATC-DGD (LASSO)[3], and DGT (NetLASSO) [4]. These methods are all based on the LASSO formulation with explicit regularization. The results are presented in Figure 1 of $\textbf{supply.pdf}$. For each method, we tuned the step size to achieve the best performance. Our proposed method demonstrated the best recovery performance in all network settings with minimum iterations.\
> $\textbf{ (2) Truncated version comparison.}$ \
> We further compared T-DGD with truncated versions of existing methods: Trun-CTA-DGD (LASSO), Trun-ATC-DGD (LASSO), and Trun-DGT (NetLASSO) which use the same Top-$s$ truncation operator. As shown in Figure 2.(a) of $\textbf{supply.pdf}$, our proposed method is the only one to achieve successful recovery, while all other  methods failed. This demonstrates that naively combining sparsification with decentralized algorithms is not granted to converge. This is precisely one of the motivations of this work: to provide communication-efficient algorithms with both provably statistical and computational guarantees. \
> [1] Yuan, K., Ling, Q., & Yin, W. (2016). On the convergence of decentralized gradient descent. SIAM Journal on Optimization, 26(3), 1835-1854.
> [2] Ji, Yao, et al. "Distributed sparse regression via penalization." Journal of Machine Learning Research 24.272 (2023): 1-62.\
> [3] Ji, Yao, et al. "Distributed (ATC) gradient descent for high dimension sparse regression." IEEE Transactions on Information Theory 69.8 (2023): 5253-5276.\
> [4] Sun, Ying, et al. "High-dimensional inference over networks: Linear convergence and statistical guarantees." arXiv preprint arXiv:2201.08507 (2022).

---

> > ### Comment · Reviewer_rjVx · 2024-08-12
> >
> > Thank you for your response to my concerns. I have slightly increased my ratings.

---

> > > ### Author Response · Authors · 2024-08-12
> > > **Authors' feedback**
> > >
> > > Thank you for highlighting the weaknesses and for considering a slight increase in the rating.

---

> > > ### Author Response · Authors · 2024-08-13
> > > **Increased ratings**
> > >
> > > Dear Reviewer,
> > >
> > > We apologize for the inconvenience, but we would greatly appreciate it if you could remind us where you have increased the rating in your review. Upon reviewing the feedback, we realized that we no longer have the details of the previous ratings and would like to ensure we have an accurate understanding.
> > >
> > > Thank you for your time and assistance.
> > >
> > > Bests,
> > > Authors

---

### Official Review · Reviewer_E5yh · 2024-07-11

**Soundness:** 3
**Presentation:** 2
**Contribution:** 3
**Rating:** 5
**Confidence:** 4

**Summary:**

This paper focuses on deriving the implicit regularization effects of decentralized gradient descent (DGD) for minimizing an objective function over undirected mesh networks. In particular, this paper establishes the fact that the solution returned by DGD with early stopping is statistically optimal under certain conditions. In addition, this paper also proposes a new method, the T-DGD, that can have better performance than vanilla DGD.

**Strengths:**

1.	The studying of the DGD setting is novel, as most works in the direction of implicit regularization of algorithms typically do not pay attention to DGD. The main contributions are clearly reflected in Theorem 1, showing the statistical guarantee and computational complexity of DGD.
2.	The proposed TDGD can be seen as an interesting application of the theoretical observation, which additionally makes the theoretical claims inspiring.

**Weaknesses:**

1.	The simulations are only for diagonal linear networks, which limits the generalizability of the proposed methods and theoretical conclusions.
2.	Though results for the setting of this paper are novel, the motivation for studying such a setting is unclear. It would be better to specify clearly why and how the setting could be useful in practice. Currently it is hard for to see how the theoretical results could be important.

**Questions:**

1.	Could you demonstrate your theoretical conclusions in more general architectures to further elaborate the implications of the theoretical claims?
2.	Previous works shows a transition from kernel regime to rich regime by varying the initialization scale, does such phenomenon exist in the DGD setting?

**Limitations:**

1.	The motivation and implications of the theoretical claims are not clear.
2.	Numerical experiments are only conducted in simple architectures.

---

> ### Author Rebuttal · Authors · 2024-08-07
>
> Thank you for your detailed and insightful comments. We appreciate the opportunity to address your concerns.\
> $\textbf{Experimental results for general architectures}.$\
> We add two experiments to validate the implicit regularization of DGD on general overparameterized neural network architecture, the details of two experiments is  in the caption of Figure 3 of $\textbf{supply.pdf}$.\
> (1) The first  is that we use vanilla decentralized SGD(DSGD) to train a depth-2, 5000 hidden ReLU network with the cross-entropy loss on the MNIST dataset. We plot the average test error (which is defined as the summation of the test error of each agent's model, then divided by the number of agents) vs. $\alpha$ in Figure 3.(a), which shows a visible phase transition for generalization ($\approx 98\%$ for $\alpha\leq 6$, and $\approx 96.6\%$ error for $\alpha \geq 100$). Figure 3.(a) shows that the transition from the kernel regime to the rich regime by varying the initialization scale may also exist in complex fully connected neural networks.\
> (2) The second is we use vanilla DSGD to train the VGG11-like deep convectional neural network on CIFAR10. The Figure 3.(b) plots the average test accuracy vs. $\alpha$.  In addition, we adopt the sparse feature learning measure [1] to monitor the sparsity of learned features in all agents along the epoch in Figure 3.(c). From two figures, we can observe that the implicit regularization of DGD under small initialization prefers choosing solutions with better generalization performance and makes DSGD implicitly learn the models that can induce sparser features.\
> $\textbf{Motivation of the study.}$\
> The purpose of this work is twofold. \
> $\textbf {1).}$  The sparse recovery problem (1) itself has important applications in sensing, signal processing, and learning where data is limited and distributed into different locations [2]. Our algorithm provides decentralized solutions to these problems and outperforms state-of-the-art methods, as shown by Figure 1 and Figure 2.(a) in $\textbf{supply.pdf}$. Furthermore, Problem (2) does not require choosing the regularization parameter before executing the algorithm. Rather, sparsity regularization is controlled through early termination. In practice, it is much easier to implement since one can stop the algorithm by monitoring the test error on the validation dataset.\
> $\textbf {2).}$ The over-parameterized formulation (2) can also be regarded as training a two-layer diagonal linear neural network and thus connects to the literature of deep learning theory. Existing works in this area primarily focus on studying the implicit bias of gradient methods in the centralized setting, with the decentralized regime less explored. When the data is split over multiple agents, a new component, the interaction of the agents through the network, comes into play and will affect the algorithms' trajectory. For a gradient algorithm that is biased toward benign solutions in the centralized setting, it is not clear if the new perturbation introduced by the network will drive it towards somewhere else. This work provides an understanding of the problem and shows how initialization, step size, and network connectivity should be coordinated to retain a small error. \
> $\textbf{Formal study of more general NN architecture.}$ Analyzing the implicit regularization for NNs, especially in the rich regime, relies heavily on the specific problem structures. As such, existing works, even in the centralized setting, are largely case-by-case studies.  The analysis in this paper for DGD could be potentially generalized to other cases, but providing formal studies is beyond the scope of the current work. We also mention there are some recent efforts trying to provide a unified perspective on the inductive bias of gradient descent [3,4]. These results characterize where the algorithm converges to, but not how fast it reaches such a solution. The technique along this line is less close to that developed in this paper. Yet it would be interesting to investigate the decentralized/FL counterpart of it. The results of Figure 3 of $\textbf{supply.pdf}$ on fully connected and deep CNN can demonstrate that our current theorem 1  (DGD with smaller initialization implicitly chooses the sparser and better generalization solutions in solving overparameterized models) is not limited to the diagonal linear network, but might be valid for general architecture.\
> $\textbf{Kernel to rich regime transition.}$\
> Previous works show a transition from the kernel regime to the rich regime by varying the initialization scale; this also happens for DGD. Experimental results showing the phenomenon are shown in Figure 2.(b) of  $\textbf{supply.pdf}$. We  observe that when we increase the initialization scale $\alpha$ gradually, DGD would converge to the minimal $\ell_2$ norm solution $\boldsymbol w^\star_{\ell_2}$. On the contrary, when we decrease the $\alpha$, the DGD would converge to the minimal ${\ell_1}$ norm solution (sparse solution) $\boldsymbol w^\star_{\ell_1}$. Thus, Figure 2.(b) demonstrates the existence of phase transition from kernel to rich regime for DGD when decreasing initialization $\alpha$. Since we focus on sparse recovery, small initialization would achieve this aim with better generalization.\
> [1] Andriushchenko, Maksym, et al. "Sgd with large step sizes learns sparse features." International Conference on Machine Learning. PMLR, 202.\
> [2] Mateos, Gonzalo, Juan Andrés Bazerque, and Georgios B. Giannakis. "Distributed sparse linear regression." IEEE Transactions on Signal Processing 58.10 (2010): 5262-5276.\
> [3] Azulay, Shahar, et al. "On the implicit bias of initialization shape: Beyond infinitesimal mirror descent." International Conference on Machine Learning. PMLR, 2021.\
> [4] Moroshko, Edward, et al. "Implicit bias in deep linear classification: Initialization scale vs training accuracy." Advances in neural information processing systems 33 (2020): 22182-22193.

---

### Official Review · Reviewer_BLgr · 2024-07-13

**Soundness:** 4
**Presentation:** 3
**Contribution:** 2
**Rating:** 5
**Confidence:** 4

**Summary:**

The paper shows that the implicit regularization enjoyed by a well-known reparameterizion of least squares extends to the decentralized setting.  Convergence guarantees are provided, and it is also shown that communication can be limited by thresholding vectors before they are communicated to neighbors.

**Strengths:**

The convergence guarantess end up having the form that one would hope.

**Weaknesses:**

I don't quite understand how this paper is contributing to our understanding of machine learning.  That running gradient descent on the nonlinear least-squares problem in (2) gives you sparse solutions that are statistically near-oprimal is interesting, but is of course known and written about a lot already.  That you can make gradient descent for non-convex problems decentralized using consesus averaging has also been written about a lot already.  That communications can be reduced using truncation when we are solving sparse regression problems is also an idea that has received plenty of attention.  This paper puts all of these things together, but we don't seem to learn anything new about implicit regularization or decentralized optimization.  The algorithm is not compared against existing decentralized sparse solvers.  If it happened to perform better even on stylized examples, that would at least be something new.

**Questions:**

none

---

> ### Author Rebuttal · Authors · 2024-08-07
>
> Thank Reviewer E5yh  for your valuable comments. We appreciate the opportunity to clarify the contributions and address the concerns in your review.
>
> $\textbf{(1) Contribution and significance}$ \
> While implicit bias has been studied for centralized gradient methods for various models, the decentralized setting is relatively less with several key questions largely under-explored: $\textbf{1)}:$ Will decentralized algorithms also experience implicit bias as in the centralized setting? $\textbf{2)}:$ How will the initialization and the interaction of the agents through the network affect the implicit regularization? $\textbf{3)}:$ How to choose the algorithm and set hyper-parameters to obtain solutions with small errors? \
> Even under the overparameterized sparse model, answering these questions is non-trivial due to the additional error terms induced by the decentralized network, which can hinder DGD from inducing sparse regularization. For example, DGD in poorly connected large-scale networks may not induce optimal regularization, as validated in Figure. 3(a) of the paper.
>
> Our main contributions are as follows:\
> $\textbf{Statistical and computational guarantee of DGD}.$ We provided sufficient conditions—namely, if the global design matrix satisfies the RIP condition, the initialization is sufficiently small, and the network is sufficiently connected—the early-stopped DGD can implicitly obtain statistically optimal solutions. These results, formally stated in Theorem 1, provide a potential answer to the general questions listed above.\
> $\textbf{New communication-efficient algorithm by truncation.}$ The analysis of DGD shows that the magnitude of the elements on the support will progressively increase while those off support remain small as initialization, which motivates the development of the T-DGD algorithm that keeps the top $s$ largest elements and truncates others. Proposition 1 shows in the high SNR regime, T-DGD enjoys the same iteration complexity as DGD, but the communication cost is significantly reduced from $\mathcal O(d)$ to $\mathcal O(s)$. There is currently no work to prove that the truncated explicitly decentralized LASSO can achieve the $\mathcal O(s)$ communication complexity. Our experiments in Figure 2.(a) of $\textbf{supply.pdf}$ show that truncated explicitly decentralized LASSO solvers do not work. This indicates that the implicit regularization induced by DGD in solving the overparameterized model can be better than existing explicitly decentralized LASSO solvers. This expands the new understanding of the benefits of optimization-induced implicit regularizations in overparameterized learning models.
>
> $\textbf{(2)Simple combination of existing results will not work.}$ \
> We fully agree that the implicit bias of GD on problem (2) has been analyzed in centralized setting. However, existing understandings of DGD indicate that it cannot converge to exact minimizers or stationary points if the problem is nonconvex [1]. As problem (2) is nonconvex, none of these results could easily predict that DGD will compute statistically optimal solutions.\
> T-DGD reduces the transmitted elements from $d$ to $s$ without affecting the iteration complexity, which is achieved by carefully exploiting the problem's sparsity structure induced by implicit regularization of DGD, which is new to our knowledge. \
> $\textbf{(3) Insights of broader interest.}$\
> $\textbf{ Algorithm selection}.$ In decentralized optimization literature, methods with gradient correction are often preferred due to  heterogeneity of local loss functions. However, our results suggest that for certain machine learning models, the simpler DGD  suffices to achieve satisfactory or even optimal outcomes.\
> $\textbf{ Generalization to more complex machine learning models}.$ The study potentially initiates further investigations into the implicit regularization induced by decentralized methods for more complex models. Our fine-grained analysis of the interaction between the optimization dynamics and network effect can be developed and generalized to explore the implicit regularization of other decentralized methods in more complex learning models, such as DNN.\
> $\textbf{(4)Comparison with decentralized sparse solvers.}$ \
> $\textbf{Vanilla Comparison.}$  \
> We have compared with existing decentralized sparse solvers, namely: CTA-DGD (LASSO) [2], ATC-DGD (LASSO)[3], and DGT (NetLASSO) [4]. These methods are all based on the LASSO formulation with explicit regularization. The results are presented in Figure 1 of $\textbf{supply.pdf}$. For each method, we tuned the step size to achieve the best performance. Our proposed method demonstrated the best recovery performance in all network settings with minimum iterations.\
> $\textbf{ Truncated version comparison.}$ \
> We further compared T-DGD with truncated versions of existing methods: Trun-CTA-DGD (LASSO), Trun-ATC-DGD (LASSO), and Trun-DGT (NetLASSO) which use the same Top-$s$ truncation operator. Figure 2.(a) of $\textbf{supply.pdf}$ shows our method is the only one to achieve successful recovery, while all other  methods failed. This demonstrates that naively combining sparsification with decentralized algorithms is not granted to converge. This is precisely one of the motivations of this work: to provide communication-efficient algorithms with both provably statistical and computational guarantees. \
> [1] Yuan, K., Ling, Q., & Yin, W. (2016). On the convergence of decentralized gradient descent. SIAM Journal on Optimization, 26(3), 1835-1854.
> [2] Ji, Yao, et al. "Distributed sparse regression via penalization." Journal of Machine Learning Research 24.272 (2023): 1-62.\
> [3] Ji, Yao, et al. "Distributed (ATC) gradient descent for high dimension sparse regression." IEEE Transactions on Information Theory 69.8 (2023): 5253-5276.\
> [4] Sun, Ying, et al. "High-dimensional inference over networks: Linear convergence and statistical guarantees." arXiv preprint arXiv:2201.08507 (2022).

---

> > ### Comment · Reviewer_BLgr · 2024-08-13
> >
> > Thanks for this detailed response and for putting your numerical experiments in more context.  I can bump my score up given this.

---

> > > ### Author Response · Authors · 2024-08-14
> > > **Authors feedback**
> > >
> > > We sincerely appreciate your thoughtful review and raising your score. Thank you for providing us the opportunity to clarify our contributions. We will ensure that the numerical comparison results are included in the revised version of the paper.

---

### Author Rebuttal · Authors · 2024-08-07

We sincerely thank all the reviewers for their careful review and insightful comments. We have addressed each of the questions raised by all  reviewers in a point-by-point manner as detailed below. We hope our responses can address the concerns raised. Additionally, we have included supplementary experimental results in the attached file $\textbf{supply.pdf}$. We invite the reviewers to read these results and welcome any further suggestions they might have.

---

### Decision · Program_Chairs · 2024-09-25

**Decision:**

Accept (poster)

**Comment:**

This paper delves into the implicit regularization of decentralized gradient descent applied to diagonal linear networks for sparse regression. While implicit regularization in centralized settings has been extensively studied, this work offers valuable theoretical insights into its behavior within decentralized frameworks (which are under-explored). The authors provide a solid theoretical analysis and, based on that, propose a novel truncated decentralized gradient descent algorithm.

In the response to reviewer E5yh, the authors has provided a comprehensive summary of the novelty and key contributions. Some theoretical insights regarding the interplay between implicit regularization and decentralized optimization are new and interesting.

Although some reviewers (rjVx and Agu7) initially expressed concerns about the intuitive explanations behind the technical results, the authors addressed them in the rebuttal and will revise the paper to enhance clarity.

Ultimately, all reviewers reached a positive consensus, leading me to recommend acceptance based on the rebuttal and discussion.